# TimeAutoDiff: A Unified Framework for Generation, Imputation, Forecasting, and Time-Varying Metadata Conditioning of Heterogeneous Time Series Tabular Data

**Namjoon Suh[1]**[*] **Yuning Yang[2], Din-Yin Hsieh[2], Qitong Luan[2],**
**Shirong Xu[3]**[*] **Shixiang Zhu[4], Guang Cheng[2]**

*Microsoft[1], UCLA[2], Xiamen Univ.[3], CMU[4]*

**Reviewed on OpenReview:** *https://openreview.net/forum?id=bkUd1Dg46c*

## Abstract

We present `TimeAutoDiff`, a unified latent-diffusion framework that addresses four fundamental time-series tasks—unconditional generation, missing-data imputation, forecasting, and time-varying-metadata conditional generation—within a single model that natively handles heterogeneous features (continuous, binary, and categorical). We unify these tasks through a simple masked-modeling strategy: a binary mask specifies which time feature cells are observed and which must be generated. To make this work on mixed data types, we pair a lightweight variational autoencoder (i.e., VAE)—which maps continuous, categorical, and binary variables into a continuous latent sequence—with a diffusion model that learns dynamics in that latent space, avoiding separate likelihoods for each data type while still capturing temporal and cross-feature structure. Two design choices give `TimeAutoDiff` clear speed and scalability advantages. First, the diffusion process samples a single latent trajectory for the full time horizon rather than denoising one timestep at a time; this whole-sequence sampling drastically reduces reverse-diffusion calls and yields an order-of-magnitude throughput gain. Second, the VAE compresses along the feature axis, so very wide tables are modeled in a lower-dimensional latent space, further reducing computational load. Empirical evaluation demonstrates that `TimeAutoDiff` matches or surpasses strong baselines in synthetic sequence fidelity (discriminative, temporal-correlation, and predictive metrics) and consistently lowers MAE/MSE for imputation and forecasting tasks. Time-varying metadata conditioning unlocks real-world scenario exploration: by editing metadata sequences, practitioners can generate coherent families of counterfactual trajectories that track intended directional changes, preserve cross-feature dependencies, and remain conditionally calibrated—making "what-if" analysis practical. Our ablation studies confirm that performance is impacted by key architectural choices, such as the VAE's continuous feature encoding and specific components of the DDPM denoiser. Furthermore, a distance-to-closest-record (DCR) audit demonstrates that the model achieves generalization with limited memorization given enough dataset. Code implementations of `TimeAutoDiff` are provided in https://github.com/namjoonsuh/TimeAutoDiff.

## 1 Introduction

Time series with heterogeneous features are ubiquitous in both engineering and scientific domains, such as financial markets combining continuous stock prices with categorical market sentiment indicators (Zou et al., 2022), and healthcare systems (Theodorou et al., 2023) integrating continuous vital signs with discrete treatment variables.

---

[1]Work done while at UCLA.

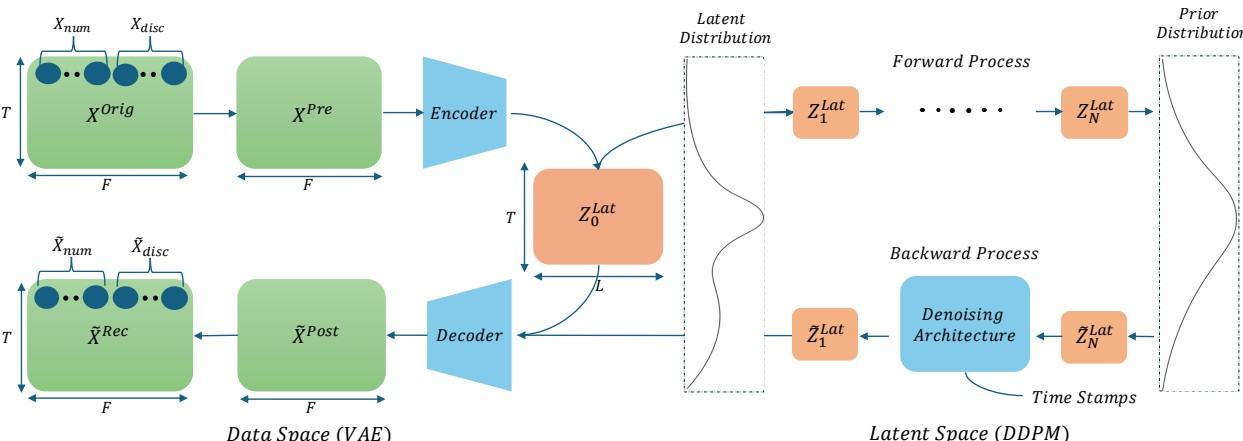

Figure 1: **The overview of `TimeAutoDiff` (Unconditional Generation):** the model has three components: (1) pre- and post-processing steps for the original (i.e., $\mathbf{X}^{\text{Orig}}$) and synthesized data (i.e., $\tilde{\mathbf{X}}^{\text{Post}}$); (2) VAE for training encoder and decoder, and for projecting the pre-processed data to the latent space; (3) Diffusion model for learning the distribution of projected data in latent space and generating new latent data. Notably, the feature dimension can be compressed in the latent space such that $L \leq F$.

The recent emergence of diffusion models (DMs) (Ho et al., 2020; Song et al., 2020b) has garnered significant attention across scientific disciplines, achieving remarkable success in diverse domains, including image synthesis (Rombach et al., 2022), tabular data generation (Zhang et al., 2023b), molecular modeling (Xu et al., 2023), and audio synthesis (Zhang et al., 2023a). The diffusion model consists of two processes. In the *forward* process, data are progressively perturbed until they resemble pure Gaussian noise, while a neural network is trained to estimate the noise added at each diffusion step. In the *reverse* process, the trained denoising network generates new samples by iteratively removing noise, starting from random Gaussian noises. Time series community has recently adopted DMs for downstream tasks (Yang et al., 2024b) for their abilities of generating *high quality, complex sequences*. Nonetheless, existing models primarily focus on continuous time series data, which is suboptimal for tasks involving heterogeneous features. (The modeling efforts for time series with heterogeneous features are surprisingly sparse.) Additionally, most models lack versatility, typically being capable of performing only a single task.

In this paper, we propose a novel approach designed to overcome the aforementioned constraints by utilizing a smooth continuous latent space, named `TimeAutoDiff`. Our framework is structured as a Variational Autoencoder (VAE) (Kingma & Welling, 2013), with DMs functioning on the latent space. The model is designed for 'multi-task' functionalities, capable of performing four distinct tasks on time series with heterogeneous features: (1) **unconditional time series generation** (Yoon et al., 2019; Naiman et al., 2023), (2) **missing data imputation** (Tashiro et al., 2021; Wang et al., 2023b). (3) **forecasting** (Nie et al., 2022; Naiman et al., 2024), and (4) **time-varying metadata conditional generation** (hereafter referred to as TV-MCG) (Narasimhan et al., 2024). Technically, these four tasks can be framed within the context of conditional distribution modeling, where the goal is to generate target data given observed data, inspired by the idea used in masked language modeling (Devlin, 2018). To the best of our knowledge, we are the first work to incorporate heterogeneous features into latent space modeling for multi-task time series generations.

A unique advantage of `TimeAutoDiff` is that unlike previous DM methods operating in the feature domain (Tian et al., 2023), we explicitly incorporate a latent space to capture the complex structures of time series with heterogeneous features. This framework exhibits several strengths. First, projecting the raw data to continuous latent space avoids direct modeling of complicated likelihood of heterogeneous features in the original data space (VAE's role), and leverages the power of DMs for modeling distributions in the continuous space, and is therefore more expressive. Second, working on the latent space allows the model to capture complicated dependent structures along feature and temporal dimensions of data naturally without sophisticated modeling of heterogeneous issues. Third, the latent space allows for training and sampling in a lower dimensionality, reducing generative modeling complexity by compressing data along feature dimensions and facilitating efficient modeling of high-dimensional features. Lastly, the latent space

| Models | Hetero. | Single-Seq. | Multi-Seq. | Cond. Gen. | Domain-Agnostic | Code | Sampling Time |
|---|---|---|---|---|---|---|---|
| `TimeAutoDiff` | ✓ | ✓ | ✓ | ✓ | ✓ | ✓ | 3 |
| `TimeDiff` (Tian et al., 2023) | ✓ | ✓ | ✗ | ✗ | ✗ | ✗ | – |
| `Diffusion-ts` (Yuan & Qiao, 2023) | ✗ | ✓ | ✗ | ✗ | ✓ | ✓ | 5 |
| `TSGM` (Lim et al., 2023) | ✗ | ✓ | ✗ | ✗ | ✓ | ✓ | 6 |
| `TimeGAN` (Yoon et al., 2019) | ✗ | ✓ | ✗ | ✗ | ✓ | ✓ | 2 |
| `DoppelGANger` (Lin et al., 2020) | ✗ | ✓ | ✗ | ✗ | ✓ | ✓ | 1 |
| `EHR-M-GAN` (Li et al., 2023) | ✓ | ✓ | ✗ | ✗ | ✗ | ✓ | – |
| `CPAR` (Zhang et al., 2022) | ✓ | ✗ | ✓ | ✗ | ✓ | ✓ | 4 |
| `TabGPT` (Padhi et al., 2021b) | ✓ | ✗ | ✓ | ✗ | ✗ | ✓ | – |

Table 1: A comparison table that summarizes `TimeAutoDiff` against baseline methods, evaluating metrics like heterogeneity, single- and multi-sequence data generation, conditional generation, domain-agnostic (i.e., whether the model is not designed for specific domains), code availability, and sampling time. Baseline models without domain specificity and with available code are used for numerical comparisons. The sampling time column ranks models by their speed, with lower numbers indicating faster sampling. (Only models with generative capabilities for time series data are included in the comparison.)

approach simplifies modeling by transforming $P(\text{heterogeneous} \mid \text{heterogeneous})$ into the more tractable $P(\text{continuous} \mid \text{heterogeneous})$. This enables efficient handling of diverse heterogeneous features as conditions, crucial for tasks like imputation, forecasting, and TV-MCG. Our framework, inspired by successes in text-guided image generation (Rombach et al., 2022), offers a unified solution for complex, heterogeneous data that would be challenging to model otherwise.

## 2 Relevant Literatures

In this section, we focus on topics most pertinent to our study: heterogeneous tabular modeling, time series data generation, and downstream time series tasks such as forecasting, imputation, and conditional generation.

**Heterogeneous Non-time series-tabular Modeling:** For mixed-type tables, GAN methods such as `CTGAN` (Xu et al., 2019) and its extensions `CTABGAN` (Zhao et al., 2021) and `CTABGAN+` (Zhao et al., 2022) popularized adversarial synthesis for categorical–continuous data (Goodfellow et al., 2020). Diffusion-based tabular synthesizers soon showed advantages, with `Stasy` (Kim et al., 2022) outperforming GANs on several tasks, although classical diffusion processes (Ho et al., 2020; Song et al., 2020b) are not inherently tailored to heterogeneity. Subsequent work addressed this via tailored stochastic processes and objectives—e.g., Doob's $h$-transform for categorical variables (Liu et al., 2022), `TabDDPM` (Kotelnikov et al., 2022) and `CoDi` (Lee et al., 2023), which combine discrete/continuous diffusion (Song et al., 2020b; Hoogeboom et al., 2022) and employ contrastive co-evolution (Schroff et al., 2015). Most recently, latent diffusion has emerged as a simple and effective unifying strategy: `AutoDiff` (Suh et al., 2023) and `TabSyn` (Zhang et al., 2023b) first compress heterogeneous tables with an autoencoder and then learn a diffusion model in the continuous latent space, demonstrating strong fidelity and utility across diverse tabular benchmarks. However, these models are not explicitly designed to capture the temporal inductive biases inherent in time series tabular data.

**Unconditional Time Series Data Generation:** Early work adopts GANs: `TimeGAN` (Yoon et al., 2019) learns a latent sequence representation with an autoencoder–adversarial scheme, followed by healthcare-oriented variants such as `EHR-Safe` (Yoon et al., 2023), which integrates an encoder–decoder with a GAN, and `EHR-M-GAN` (Li et al., 2023), which employs type-specific encoders; however, GANs remain sensitive to non-convergence and mode collapse (Goodfellow et al., 2020). Diffusion models then gain traction: `TimeDiff` (Tian et al., 2023) couples multinomial diffusion for discrete variables with Gaussian diffusion for continuous ones to synthesize mixed-type EHR sequences; `TSGM` (Lim et al., 2023) uses a latent conditional score-based diffusion for continuous series; and `Diffusion-TS` (Yuan & Qiao, 2023) combines a transformer autoencoder with latent diffusion to capture temporal dynamics. Beyond these, GPT-style and parametric approaches include `TabGPT` (Padhi et al., 2021b), a GPT-2 based synthesizer for mixed-type time-series tables, and `CPAR` (Zhang et al., 2022), which assigns different parametric models to heterogeneous variables in multi-sequence tabular settings. Despite progress, accurately modeling cross-feature correlations while handling complex temporal dependencies across data types remains challenging.

*Remark* 2.1. For a more in-depth understanding of our model's positioning, we provide detailed comparisons between `TimeAutoDiff` and related methods—including `TimeDiff`, `Diffusion-TS`, `TabSyn`, `AutoDiff`, and `TabDDPM`—in Appendix D.

**Imputation** & **Forecasting for Time Series Data:** Several recent methods addressed time series tasks like filling missing data and predicting future values. For filling in missing data, CSDI (Tashiro et al., 2021) studied the imputations of continuous time series tabular data through a diffusion-based framework. The main idea was to employ specially designed masks; masking the observed data, and letting the model predict the masked values in the observations, i.e., self-supervised learning. Then, the trained model could impute the real missing parts of the table by treating them as masked observations. Building on this diffusion paradigm, `MG-TSD` (Fan et al., 2024) was introduced to stabilize the stochastic nature of diffusion models by using the inherent multi-granularity levels within the data as intermediate targets to guide the learning process. `TimeDiT` (Cao et al., 2024) further extends this by combining a diffusion transformer with a comprehensive masking strategy for imperfect data and a novel mechanism for injecting physics knowledge during inference. `FiLM` (Zhou et al., 2022) used a memory system and noise-filtering techniques to capture long-term patterns efficiently. Meanwhile, `DLinear` (Zeng et al., 2023) broke time series into seasonal and trend parts, using simple linear models to match or outperform complex Transformer models, offering a fast and lightweight option. `TimesNet` (Wu et al., 2022) organized time series into a 2D format to highlight repeating patterns, using advanced convolution techniques to capture both short-term and long-term trends, which helped with both predicting future values and filling in missing data. `MICN` (Wang et al., 2023a) blended local and global patterns in time series data using a mix of standard and specialized convolution methods across different scales, boosting long-term prediction accuracy while keeping computations fast. `Time-LLM` (Jin et al., 2023) adapted a pre-trained language model using text-based guides and a special input format to predict time series values, achieving strong results with little or no extra training. `MOMENT` (Goswami et al., 2024) trained a large Transformer model on a wide range of time series data, teaching it to fill in missing parts, creating a versatile model that worked for predicting future values, filling gaps, classifying patterns, and detecting anomalies with minimal extra training. Following this trend toward large-scale foundation models, `Time-MoE` (Xiaoming et al., 2025) introduced a sparse Mixture-of-Experts (MoE) architecture, scaling a decoder-only model to billion-scale parameters on the massive "Time-300B" dataset to enhance forecasting precision while maintaining computational efficiency. `TEFN` (Zhan et al., 2024) used a method to measure both feature-specific and time-based patterns in data, combining them to make accurate and reliable long-term predictions. Other Transformer-based innovations include `PatchTST` (Nie et al., 2022), which segments time series into 'patches' and applies attention across these segments rather than individual points, and `iTransformer` (Liu et al., 2023), which inverts this by applying attention across entire variates (channels) as tokens to explicitly model inter-channel dependencies. Finally, `TimeMixer` (Wang et al., 2024b) blended patterns across different time scales using a simple, fast model (MLP-only architecture), combining specialized predictors to achieve top results for both short-term and long-term forecasting. This concept was extended in `TimeMixer++` (Wang et al., 2024a), a general Time Series Pattern Machine (TSPM) that transforms multi-scale time series into multi-resolution "time images" and uses dual-axis attention to decompose seasonal and trend patterns, achieving state-of-the-art results across eight distinct time series tasks.

*Remark* 2.2. The development of models like `MOMENT` within this literature highlights a significant trend toward large-scale, pre-trained Time-Series Foundation Models (TSFMs), which aim to generalize across diverse tasks and domains. However, many of these TSFMs, including the aforementioned examples, are often limited to continuous-only data (assuming channel independence) or specialized tasks. `TimeAutoDiff` complements this emerging field by introducing a unified latent-diffusion framework that (1) natively handles heterogeneous features (continuous, categorical, and binary), and (2) unifies four key tasks—generation, imputation, forecasting, and conditional generation—using a single conditional masking strategy. We provide more in-depth comparisons in this regard in the Appendix D.

**Time-Varying Metadata Conditional Generation:** The only work on *TV-MCG* we are aware of is `Timeweaver` (Narasimhan et al., 2024), which formulates conditional generation as diffusion conditioned on heterogeneous, time-varying metadata $\mathbf{X}^{\mathrm{con}}$: categorical and continuous metadata are tokenized, fused via self-attention into a time-aligned conditioning sequence, and injected into a CSDI-style denoiser to learn $P(\mathbf{X}^{\mathrm{tar}} \mid \mathbf{X}^{\mathrm{con}})$, enabling metadata-controllable synthesis. Unlike `Timeweaver`, which restricts $\mathbf{X}^{\mathrm{tar}}$ to continuous variables, `TimeAutoDiff` supports heterogeneous $\mathbf{X}^{\mathrm{tar}}$ (continuous, binary, categorical).

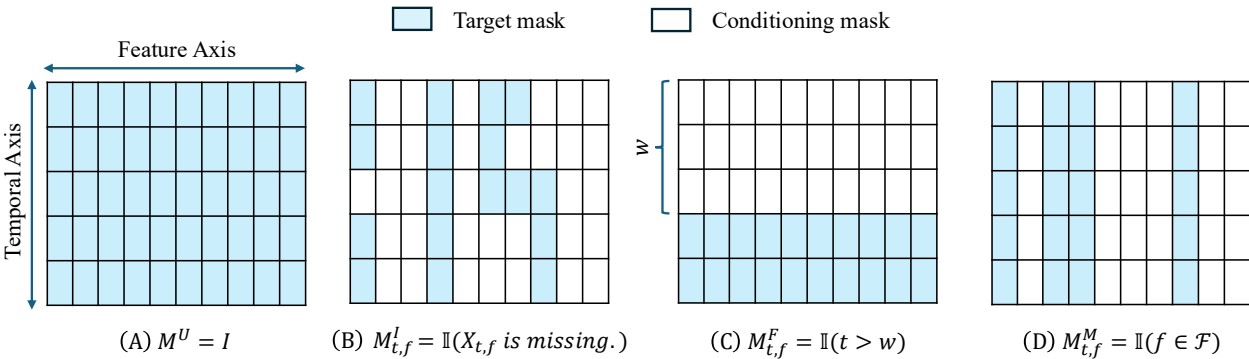

Figure 2: **Illustration of four binary-mask M (shaded cells = 1) on a $T \times F$ timeseries grid:** (A) Unconditional Generation ($\mathbf{M}^U$): all $T \times F$ entries shaded. (B) Missing-Data Imputation ($\mathbf{M}^I_{t,f} = \mathbb{1}(X_{t,f}$ is missing)): shaded cells mark missing entries, and unshaded cells indicate observed values. (C) Forecasting ($\mathbf{M}^F_{t,f} = \mathbb{1}(t > w)$): only rows with $t > w$ shaded, the first $w$ rows serve as conditioning. (D) Metadata-Conditional ($\mathbf{M}^M_{t,f} = \mathbb{1}(f \in \mathcal{F})$): only columns corresponding to features in $\mathcal{F}$ are shaded.

## 3 Problem Setting

Let $\mathbf{X} := [\mathbf{x}_1, \ldots, \mathbf{x}_T]^{\mathsf{T}} \in \mathbb{R}^{T \times F}$ be a $T$-sequence time series tabular data with heterogeneous features. Each observation $\mathbf{x}_j := [\mathbf{x}_{\text{Disc},j}, \mathbf{x}_{\text{Cont},j}]$ is an $F$ dimensional feature vector that includes both discrete ($\mathbf{x}_{\text{Disc},j}$) and continuous ($\mathbf{x}_{\text{Cont},j}$) variables. Throughout this paper, we assume that there are $B$ i.i.d observed sequences (i.e., $\{\mathbf{X}_i\}_{i=1}^B$) sampled from $P(\mathbf{X})$. Every record is timestamped in the 'YEAR-MONTH-DATE-HOUR' format. Notably, these timestamps (i.e $\mathbf{ts}_i \in \mathbb{R}^{T \times 4}, i \in \{1, \ldots, B\}$) may differ across sequences within the same batch (i.e, $\mathbf{ts}_i \neq \mathbf{ts}_j, \forall i \neq j$) and are treated as auxiliary variables that can be leveraged during training and inference. (This will be more detailed in Subsection 4.4.)

Our goal is to model the conditional distribution $P_\theta(\mathbf{X}^{\text{tar}} \mid \mathbf{X}^{\text{con}}, \mathbf{M})$, where $\mathbf{M} \in \mathcal{M}$ is a binary mask that specifies which entries belong to target output (i.e., $\mathbf{X}^{\text{tar}}$). In other words, the target data $\mathbf{X}^{\text{tar}} := \mathbf{M} \odot \mathbf{X}$ and the conditioning data $\mathbf{X}^{\text{con}} := (\mathbf{I} - \mathbf{M}) \odot \mathbf{X}$ with $\mathbf{I}$ being a matrix all entries are 1s. Here, $\odot$ denotes element-wise matrix multiplication. A family of the binary masks $\mathbf{M} \in \{0, 1\}^{T \times F}$ provides a unified framework that supports four different time series tasks framed under conditional distributional modeling:

1. **Unconditional Generation**: Setting $\mathbf{M}^U = \mathbf{I}$ (all entries are 1) makes $\mathbf{X}^{\text{con}} = \mathbf{0}$ (all entries are 0) inducing unconditional generation.
2. **Missing Data Imputation**: The mask identifies missing entries in the data as $\mathbf{M}^I_{t,f} = \mathbb{1}(\mathbf{X}_{t,f}$ is missing.$), \forall t \in \{1, \ldots, T\}, f \in \{1, \ldots, F\}$.
3. **Forecasting**: Given a lookback window of size $w \in \{1, 2, \ldots, T-1\}$, the mask is defined as $\mathbf{M}^F_{t,f} = \mathbb{1}(t > w)$ for all $f \in \{1, \ldots, F\}$, where $\mathbb{1}(t > w)$ is the indicator function with values 1 for $t > w$, and 0 otherwise.
4. **Metadata Conditional Generation**: For a fixed set of features $\mathcal{F} := \{f_i : i \in \{1, \ldots, F\}\}$ and all time steps $t \in \{1, \ldots, T\}$, the mask is set to $\mathbf{M}^M_{t,f} = \mathbb{1}(f \in \mathcal{F})$.

This formulation imposes no parametric assumptions on $w$, feature indices $f$, or missingness patterns. Experimental settings for each case are detailed in Section 5.

## 4 Method

In this section, we formally introduce our model, `TimeAutoDiff`, which extends recent advances in latent diffusion for image (Rombach et al., 2022) and tabular data generation (Zhang et al., 2023b; Suh et al., 2023). Unlike these domains, heterogeneous time series require tailored architectural choices: a customized encoder-decoder in the VAE (Section 4.3) and a domain-aware denoising module in the diffusion model (Section 4.4) to effectively capture inductive biases specific to temporal data. We begin with the overall training objective in Section 4.1, followed by pre- and post-processing strategies in Section 4.2. Sections 4.3

and 4.4 detail the VAE and DDPM components, respectively. The training and sampling procedures are summarized in Section 4.5.

## 4.1 Objective function

Let $\Theta = (\phi, \psi, \theta)$ denote the parameters involved in `TimeAutoDiff` where each parameter characterizes the encoder (i.e., $q_\phi$), decoder (i.e., $P_\psi$), and diffusion prior (i.e., $P_\theta$) distributions, respectively. The model (i.e., $P_\Theta$) estimates the conditional distribution $P(\mathbf{X}^{\text{tar}} \mid \mathbf{X}^{\text{con}}, \mathbf{M}^{\text{task}})$ with given samples $\{(\mathbf{X}^{\text{tar},(i)}, \mathbf{X}^{\text{con},(i)}, \mathbf{M}^{\text{task},(i)})\}_{i=1}^{B} \sim P(\mathbf{X}^{\text{tar}}, \mathbf{X}^{\text{con}}, \mathbf{M}^{\text{task}})$ by minimizing the ELBO loss (i.e., $\mathcal{L}_{\text{ELBO}} := \mathcal{L}_{\text{VAE}} + \mathcal{L}_{\text{DM}}$) of the following negative log-likelihood:

$$
\begin{aligned}
& -\log P_\Theta(\mathbf{X}^{\text{tar}} \mid \mathbf{X}^{\text{con}}, \mathbf{M}^{\text{task}}) \\
& \leq \underbrace{\mathbb{E}_{q_\phi}\big[-\log P_\psi(\mathbf{X}^{\text{tar}} \mid \mathbf{X}^{\text{con}}, \mathbf{Z}_0^{\text{Lat}}, \mathbf{M}^{\text{task}})\big] + \mathcal{D}_{\mathbf{KL}}\big(q_\phi(\mathbf{Z}_0^{\text{Lat}} \mid \mathbf{X}^{\text{tar}}, \mathbf{X}^{\text{con}}) \,\|\, \mathcal{N}(0, \mathcal{I}_{TF \times TF})\big)}_{:=\mathcal{L}_{\text{VAE}}} \\
& \quad + \underbrace{\mathbb{E}_{q_\phi}\big[-\log P_\theta(\mathbf{Z}_0^{\text{Lat}} \mid \mathbf{X}^{\text{con}})\big]}_{\leq \mathcal{L}_{\text{DM}}} + \text{Constant}.
\end{aligned}
$$

The $\mathcal{D}_{\mathbf{KL}}(P \,\|\, Q)$ is a KL-divergence between two probability measures $P$ and $Q$, and $\mathcal{I}_{TF \times TF}$ is an identity matrix of dimension $TF \times TF$. A simple proof of the above ELBO loss is provided in Appendix A.

As reflected in the reconstruction term of $\mathcal{L}_{\text{VAE}}$, the VAE component takes both $\mathbf{X}^{\text{tar}}$ and $\mathbf{X}^{\text{con}}$ as input to the encoder (i.e., $q_\phi$), which outputs the latent representation $\mathbf{Z}_0^{\text{Lat}}$. The decoder then reconstructs $\mathbf{X}^{\text{tar}}$ conditioned on $\mathbf{Z}_0^{\text{Lat}}$, $\mathbf{X}^{\text{con}}$, and the task-specific mask $\mathbf{M}^{\text{task}}$. This design enables the latent representation to capture richer context from the full input $\mathbf{X} = \mathbf{X}^{\text{tar}} + \mathbf{X}^{\text{con}}$, and empirically leads to more stable training compared to using $\mathbf{X}^{\text{tar}}$ alone as input to the encoder. The diffusion prior, $P_\theta$, models the conditional distribution $P(\mathbf{Z}_0^{\text{Lat}} \mid \mathbf{X}^{\text{con}})$. The model parameter $\theta$ can be estimated through minimizing the ELBO loss of $-\log P_\theta(\mathbf{Z}_0^{\text{Lat}} \mid \mathbf{X}^{\text{con}})$, denoted as $\mathcal{L}_{DM}$. The further descriptions on $\mathcal{L}_{\text{VAE}}$ and $\mathcal{L}_{\text{DM}}$ will be specified in the following sections.

## 4.2 Pre- and post-processing steps in `TimeAutoDiff`

It is essential to pre-process the real tabular data in a form that the machine learning model can extract the desired information from the data properly. We divide the heterogeneous features into two categories; (1) continuous, and (2) discrete. Following is how we categorize the variables and process each feature type. Let $\mathbf{x}$ be the column of a table to be processed.

1. ***Continuous Features***: A column $\mathbf{x}$ is treated as a continuous (numerical) feature if its entries are real-valued, or if they are integers with more than 25 unique values (e.g., "Age"). The threshold of 25 is a user-defined hyperparameter to distinguish high-cardinality integer features from discrete ones. All continuous features are normalized to the range $[0, 1]$ using min-max scaling (Yoon et al., 2019). The processed output is denoted by $\mathbf{x}_{\text{Num}}^{\text{Proc}}$.

2. ***Discrete / Categorical Features***: A column $\mathbf{x}$ is categorized as discrete if its entries are of string type (e.g., "Gender"), or if it consists of integers with fewer than 25 distinct values. During preprocessing, each unique value in $\mathbf{x}$ is mapped to a non-negative integer index. We further divide these into binary and categorical types: $\mathbf{x}_{\text{Bin}}^{\text{Proc}}$ for variables with exactly two categories, and $\mathbf{x}_{\text{Cat}}^{\text{Proc}}$ for those with more than three categories.

3. ***Post-processing***: Once the `TimeAutoDiff` model generates synthetic data, we apply inverse transformations to recover the original format. For continuous features, this involves reversing the min-max scaling. For discrete features, the integer-encoded values are mapped back to their original categorical or string representations.

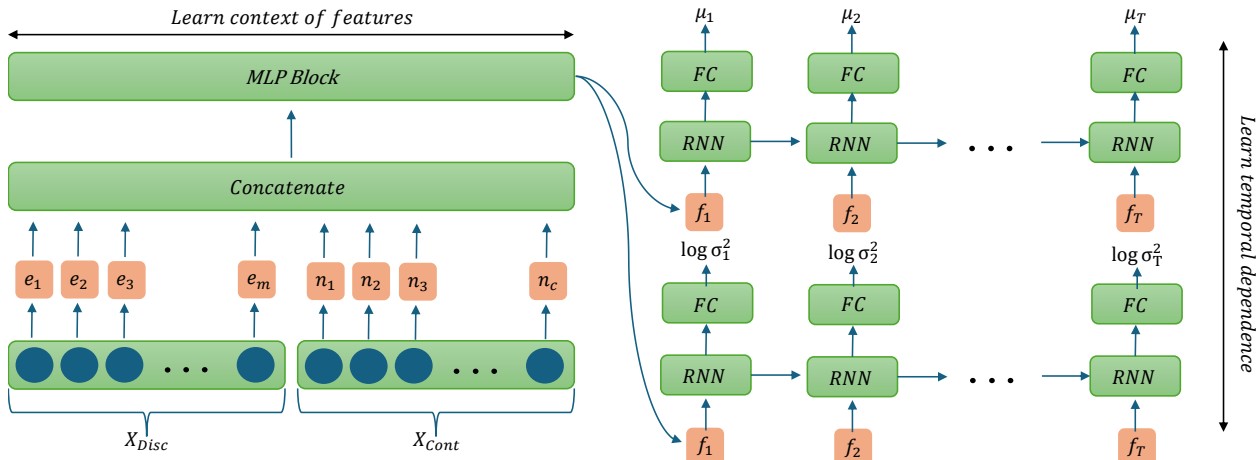

Figure 3: **Schematic architecture of the encoder in the variational autoencoder (VAE).** The encoder takes pre-processed multivariate time series input $\mathbf{X} = [\mathbf{x}_{\text{Disc}}^{\text{Proc}}; \mathbf{x}_{\text{Cont}}^{\text{Proc}}] \in \mathbb{R}^{T \times F}$ composed of $m$ discrete and $c$ continuous features, where $F = m + c$. Discrete features are embedded via a lookup table $\mathbf{e}(\cdot) \in \mathbb{R}^d$, while continuous features are transformed using frequency-based representations equation 1 to capture spectral information. At each time step $t$, the embeddings are concatenated into $\mathbf{E}(t) \in \mathbb{R}^{(m+c)d}$ and passed through an MLP to yield feature embeddings $\mathbf{f}_t \in \mathbb{R}^F$. The full sequence $\{\mathbf{f}_t\}_{t=1}^T$ is then processed independently by two RNNs to model temporal dependencies: one RNN estimates the mean vector $\boldsymbol{\mu} \in \mathbb{R}^{T \times L}$, and the other the log-variance $\log \boldsymbol{\sigma}^2 \in \mathbb{R}^{T \times L}$ of the approximate posterior. The latent trajectory $\mathbf{Z}_0^{\text{Lat}} \in \mathbb{R}^{T \times L}$ is obtained by sampling via the reparameterization trick: $\mathbf{Z}_0^{\text{Lat}} = \boldsymbol{\mu} + \mathbf{E} \odot \boldsymbol{\Sigma}$, where $\boldsymbol{\Sigma} = \exp(0.5 \log \boldsymbol{\sigma}^2)$ and $\mathbf{E} \sim \mathcal{N}(0, \mathbf{I})$. This encoder compresses the feature dimension from $F$ to $L$ while preserving temporal resolution.

### 4.3 Variational Autoencoder in `TimeAutoDiff`

In this section, encoder, decoder, objective function and training of VAE will be introduced.

**Encoder in VAE:** We begin with the pre-processed input data

$$\mathbf{X} = \mathbf{X}^{\text{tar}} + \mathbf{X}^{\text{con}} := [\mathbf{x}_{\text{Disc}}^{\text{Proc}}; \mathbf{x}_{\text{Cont}}^{\text{Proc}}] \in \mathbb{R}^{T \times F},$$

where $\mathbf{x}_{\text{Disc}}^{\text{Proc}} \in \mathbb{R}^{T \times m}$ contains discrete (binary and categorical) features, and $\mathbf{x}_{\text{Cont}}^{\text{Proc}} \in \mathbb{R}^{T \times c}$ contains continuous features, with $F = m + c$.

For discrete features $\mathbf{x}_j$ ($j \in \{1, \ldots, m\}$), we map their values into a $d$-dimensional embedding space $\mathbf{e}(x_j(t)) \in \mathbb{R}^d$ using a lookup table $\mathbf{e}(\cdot) \in \mathbb{R}^d$, where $d = 128$. This embedding approach, motivated by TabTransformer (Huang et al., 2020), allows the model to easily differentiate discrete classes across columns. For a continuous feature value $\nu = x_{\text{Cont}}^{\text{Proc}}(t, i)$, we employ a frequency-based representation (in short 'FR' for later use in Figure 5):

$$n_i(\nu) := \text{Linear}\big(\text{SiLU}\big(\text{Linear}\big([\sin(2^0 \pi \nu), \cos(2^0 \pi \nu), \ldots, \sin(2^7 \pi \nu), \cos(2^7 \pi \nu)]\big)\big)\big) \in \mathbb{R}^d, \quad (1)$$

which captures high-frequency variations in continuous signals. This leverages the spectral bias of deep networks towards low-frequency functions (Rahaman et al., 2019), ensuring better reconstruction fidelity for continuous features.

At each time step $t$, we concatenate the discrete and continuous embeddings:

$$\mathbf{E}(t) := [\mathbf{e}(x_1(t)); \ldots; \mathbf{e}(x_m(t)); n_1(x_{\text{Cont}}^{\text{Proc}}(t, 1)); \ldots; n_c(x_{\text{Cont}}^{\text{Proc}}(t, c))] \in \mathbb{R}^{(m+c)d}.$$

Applying an MLP block at each timestep and stacking the outputs gives

$$\mathbf{Emb}(\mathbf{X}^{\text{Proc}}) = [\text{MLP}(\mathbf{E}(1)); \ldots; \text{MLP}(\mathbf{E}(T))]^\top \quad (2)$$
$$= [\mathbf{f}_1, \mathbf{f}_2, \ldots, \mathbf{f}_T]^\top \in \mathbb{R}^{T \times F}.$$

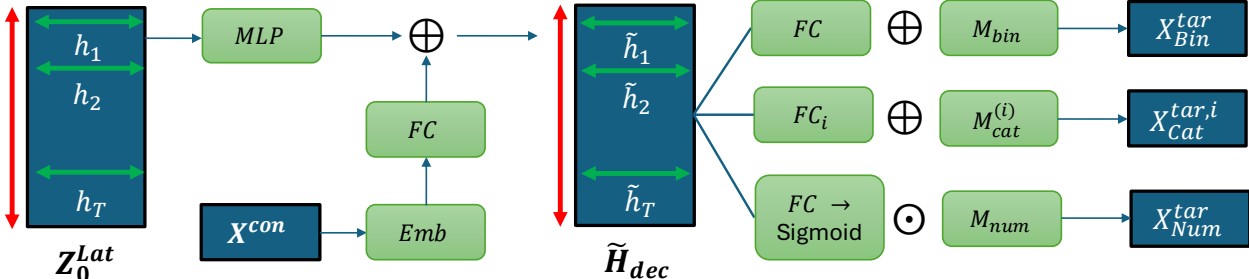

Figure 4: **VAE decoder with conditioning and modality-specific masking.** A latent trajectory $\mathbf{Z}_0^{\text{Lat}}$ is mapped by an MLP to a hidden sequence $\mathbf{H}_{dec}$. Conditioning information $\mathbf{X}^{\text{con}}$ is embedded (Emb) and projected with a Linear layer, then added to the hidden state to form $\tilde{\mathbf{H}}_{dec} = \mathbf{H}_{dec} + \text{Linear}(\text{Emb}(\mathbf{X}^{\text{con}}))$, injecting context at every time step. From $\tilde{\mathbf{H}}_{dec}$, three modality-specific heads produce targets: **Binary** — $\text{Linear}(\tilde{\mathbf{H}}_{dec}) + \mathbf{M}_{\text{bin}} \to \mathbf{x}_{\text{Bin}}^{\text{tar}}$ (logits with an additive mask that biases masked positions); **Categorical** — for each categorical variable $i$, $\text{Linear}[i](\tilde{\mathbf{H}}_{dec}) + \mathbf{M}_{\text{cat}}^{(i)} \to \mathbf{x}_{\text{Cat}}^{\text{tar},i}$ (class logits with an additive mask that routes masked entries to a designated class, e.g., index 0); **Numerical** — $\sigma(\text{Linear}(\tilde{\mathbf{H}}_{dec})) \odot \mathbf{M}_{\text{num}} \to \mathbf{x}_{\text{Num}}^{\text{tar}}$ (values with a multiplicative elementwise gate). Type-specific masks $\{\mathbf{M}_{\text{bin}}, \mathbf{M}_{\text{cat}}^{(i)}, \mathbf{M}_{\text{num}}\}$ are derived from the task mask $\mathbf{M}^{\text{task}}$ by splitting and reshaping according to feature types, so that binary/categorical channels use additive logit biasing while numerical channels use elementwise gating.

The output from the MLP block, $[\mathbf{f}_1, \mathbf{f}_2, \ldots, \mathbf{f}_T]^\top \in \mathbb{R}^{T \times F}$, is fed into two separate RNNs, each unfolded over $T$ steps, to model the mean and covariance of the latent distribution. Specifically, these RNNs process $\{\mathbf{f}_j\}_{j=1}^T$ to capture temporal dependencies. The final hidden states are then passed through fully connected (FC) layers (see Figure 3) to produce:

$$\mu := [\mu_1, \ldots, \mu_T]^\top \in \mathbb{R}^{T \times L},$$
$$\log \sigma^2 := [\log \sigma_1^2, \ldots, \log \sigma_T^2]^\top \in \mathbb{R}^{T \times L}.$$

Sampling from these parameters via the reparameterization trick yields:

$$\mathbf{Z}_0^{\text{Lat}} := \mu + \mathbf{E} \odot \mathbf{\Sigma} \in \mathbb{R}^{T \times L},$$

where $\mathbf{\Sigma} = \exp\left(\frac{1}{2}\log \sigma^2\right)$ and $\mathbf{E} \in \mathbb{R}^{T \times L}$ whose entries are drawn from $\mathcal{N}(0, 1)$. Note that the compression into this lower-dimensional latent space occurs along the feature dimension, reducing it from $F$ to $L (\leq F)$.

**Decoder in VAE:** The decoder reconstructs the binary, categorical, and numerical components of the target data $\mathbf{X}^{\text{tar}}$ from latent representations conditioned with $\mathbf{X}^{\text{con}}$. Given a latent feature tensor $\mathbf{Z}_0^{\text{Lat}} \in \mathbb{R}^{T \times L}$ through a MLP block:

$$\mathbf{H}_{dec} = \text{MLP}(\mathbf{Z}_0^{\text{Lat}}) \in \mathbb{R}^{T \times h},$$

where $h = \texttt{hidden\_size}$.

The decoder incorporate conditioning information $\mathbf{X}^{\text{con}} \in \mathbb{R}^{T \times F}$. This conditioning data is embedded through the operation equation 2 (i.e., $\mathbf{Emb}(\mathbf{X}^{\text{con}})$) and projected into the hidden space. The resulting conditional embedding is added to $\mathbf{H}_{dec}$, enabling context-dependent reconstruction:

$$\tilde{\mathbf{H}}_{dec} = \mathbf{H}_{dec} + \text{Linear}(\mathbf{Emb}(\mathbf{X}^{\text{con}})).$$

Additionally, to handle masked values and produce modality-specific reconstructions, the decoder uses the provided mask $\mathbf{M}^{\text{task}} \in \{0,1\}^{T \times F}$ to *create* separate masks for binary ($\mathbf{M}_{\text{bin}}$), categorical ($\mathbf{M}_{\text{cat}}^{(i)}$), and numerical ($\mathbf{M}_{\text{num}}$) features. These newly formed masks are adjusted such that masked inputs are effectively forced to zero for binary, categorical and numerical features, ensuring coherent reconstructions even in the presence of unobserved entries.

$$\mathbf{M}_{\text{num}} \in \mathbb{R}^{T \times n_{\text{nums}}}, \quad \mathbf{M}_{\text{bin}} \in \mathbb{R}^{T \times n_{\text{bins}}},$$
$$\{\mathbf{M}_{\text{cat}}^{(i)}\}_{i=1}^{n_{\text{cats}}}, \quad \mathbf{M}_{\text{cat}}^{(i)} \in \mathbb{R}^{T \times \texttt{cat}_i}.$$

Here, $n_{\text{bins}}$, $n_{\text{cats}}$, and $n_{\text{nums}}$ are the counts of binary, categorical, and numerical features, respectively, and $\texttt{cat}_i$ is the cardinality of $i$-th categorical feature. These modality-specific masks are constructed as follows:

- **Numerical Mask:** $\mathbf{M}_{\text{num}}$ is identical in shape and values to the corresponding numerical portion of $\mathbf{M}$. Thus, masked values are directly mapped to zeros, making no predictions are produced for masked numerical entries.

- **Binary Mask:** $\mathbf{M}_{\text{bin}}$ retains the same shape as the binary portion of $\mathbf{M}$, but any zero-valued entries are replaced with a large negative constant. This effectively suppresses any positive predictions and forces the model to output zeros for masked binary features.

- **Categorical Mask:** For each categorical feature $i$, $\mathbf{M}_{\text{cat}}^{(i)}$ is expanded along the category dimension, and any zero-valued entries of the original mask are assigned a large positive constant at the first category index. This ensures that masked categorical inputs are consistently mapped to a designated "masked" category (assigned index 0), concentrating the predicted probability mass on that category.

Equipped with the well-designed masks $\mathbf{M}_{\text{bin}}$, $\{\mathbf{M}_{\text{Cat}}^{(i)}\}_{i=1}^{n_{\text{cats}}}$, $\mathbf{M}_{\text{num}}$, the final outputs of decoder is $\widehat{\mathbf{X}}^{\text{tar}} := [\mathbf{x}_{\text{Bin}}^{\text{tar}}, \{\mathbf{x}_{\text{Cat}}^{\text{tar},i}\}_{i=1}^{n_{\text{cats}}}, \mathbf{x}_{\text{Num}}^{\text{tar}}]$:

$$\mathbf{x}_{\text{Bin}}^{\text{tar}} = \text{Linear}(\tilde{\mathbf{H}}_{dec}) + \mathbf{M}_{\text{bin}} \in \mathbb{R}^{T \times n_{\text{bins}}}.$$

$$\mathbf{x}_{\text{Cat}}^{\text{tar},i} = \text{Linear}[i](\tilde{\mathbf{H}}_{dec}) + \mathbf{M}_{\text{cat}}^{(i)} \in \mathbb{R}^{T \times \texttt{cat}_i}.$$

$$\mathbf{x}_{\text{Num}}^{\text{tar}} = \sigma(\text{Linear}(\tilde{\mathbf{H}}_{dec})) \odot \mathbf{M}_{\text{num}} \in \mathbb{R}^{T \times \texttt{num}},$$

where $\sigma(\cdot)$ denotes a sigmoid function.

**Obj. function & Training of VAE:** The reconstruction error $\ell_{\text{recons}}(\mathbf{X}^{\text{tar}}, \widehat{\mathbf{X}}^{\text{tar}})$ in the VAE is defined as the sum of mean-squared error (MSE), binary cross entropy (BCE), and cross-entropy (CE) between the target tuple $\mathbf{X}^{\text{tar}} := [\mathbf{x}_{\text{Bin}}^{\text{tar}}, \mathbf{x}_{\text{Cat}}^{\text{tar}}, \mathbf{x}_{\text{Num}}^{\text{tar}}]$ and the output tuple from decoder $\widehat{\mathbf{X}}^{\text{tar}} := [\mathbf{x}_{\text{Bin}}^{\text{Out}}, \mathbf{x}_{\text{Cat}}^{\text{Out}}, \mathbf{x}_{\text{Num}}^{\text{Out}}]$:

$$\text{BCE}(\mathbf{x}_{\text{Bin}}^{\text{tar}}, \mathbf{x}_{\text{Bin}}^{\text{Out}}) + \text{CE}(\mathbf{x}_{\text{Cat}}^{\text{tar}}, \mathbf{x}_{\text{Cat}}^{\text{Out}}) + \text{MSE}(\mathbf{x}_{\text{Num}}^{\text{tar}}, \mathbf{x}_{\text{Num}}^{\text{Out}}).$$

We use $\beta$-VAE (Higgins et al., 2017), where a coefficient $\beta(\geq 0)$ balances between the reconstruction error and KL-divergence of $\mathcal{N}(0, \mathcal{I}_{TF \times TF})$ and $\mathbf{Z}_0^{\text{Lat}} \sim \mathcal{N}(\text{vec}(\mu), \text{diag}(\text{vec}(\sigma^2)))$. The notations $\text{vec}(\cdot)$ and $\text{diag}(\cdot)$ are vectorization of input matrix and diagonalization of input vector, respectively. Finally, we minimize the following objective function $\mathcal{L}_{\text{VAE}}$ for training:

$$\mathcal{L}_{\text{VAE}} := \ell_{\text{recons}}(\mathbf{X}^{\text{tar}}, \widehat{\mathbf{X}}^{\text{tar}}) + \beta \mathcal{D}_{\mathbf{KL}}\big(\mathcal{N}(\text{vec}(\mu), \text{diag}(\text{vec}(\sigma^2))) \,||\, \mathcal{N}(0, \mathcal{I}_{TF \times TF})\big). \tag{3}$$

Our model does not require the distribution of embeddings $\mathbf{Z}_0^{\text{Lat}}$ to strictly follow a standard normal distribution, as the diffusion model additionally handles the distributional modeling in the latent space. Therefore, following Zhang et al. (2023b), we adopt the adaptive schedules of $\beta$ with its maximum value set as 0.1 and minimum as $10^{-5}$, decreasing the $\beta$ by a factor of 0.7 (i.e., $\beta^{\text{new}} = 0.7\beta^{\text{old}}$) from maximum to minimum whenever $\ell_{\text{recons}}$ fails to decrease for a predefined number of epochs. The effects of $\beta$-scheduling will be more detailed in Section 5.

### 4.4 Diffusion Model in `TimeAutoDiff`

In this section, our customized DDPM and a newly designed denoiser $\epsilon_\theta$ in `TimeAutoDiff` are introduced.

Let $\mathbf{Z}_0^{\text{Lat}} \in \mathbb{R}^{T \times L}$ be the initial latent matrix derived from the VAE, and let $\mathbf{Z}_n^{\text{Lat}} := [\mathbf{z}_{n,1}^{\text{Lat}}, \mathbf{z}_{n,2}^{\text{Lat}}, \ldots, \mathbf{z}_{n,L}^{\text{Lat}}]$ represent the latent matrix corrupted by noises after $n \in \{1, 2, \ldots, N\}$ diffusion steps. Here, $\mathbf{z}_{n,j}^{\text{Lat}} \in \mathbb{R}^T$ is the $j$-th column of $\mathbf{Z}_n^{\text{Lat}}$. The noising process is applied column-wise, independently for each $j \in [L]$, according to:

$$q(\mathbf{z}_{n,j}^{\text{Lat}} | \mathbf{z}_{0,j}^{\text{Lat}}) = \mathcal{N}\big(\sqrt{\bar{\alpha}_n} \mathbf{z}_{0,j}^{\text{Lat}}, (1 - \bar{\alpha}_n) \cdot \mathcal{I}_{T \times T}\big),$$

where $\bar{\alpha}_n = \prod_{i=1}^n \alpha_i$ and $\{\alpha_i\}_{i=1}^n \in [0,1]^n$ is a decreasing sequence. We employ a linear noise schedule (Ho et al., 2020). Further details on DDPM are provided in Appendix F.

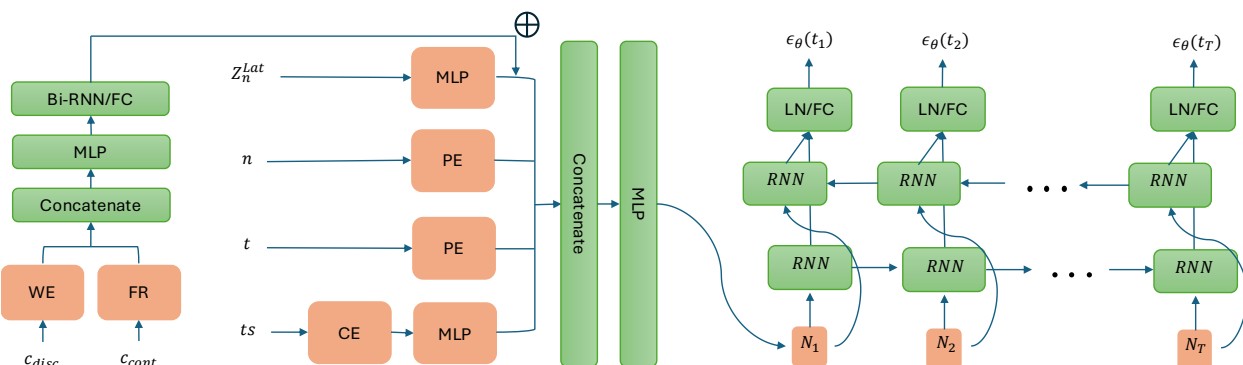

Figure 5: **The schematic architecture of the denoising model** $\epsilon_\theta(\mathbf{Z}_n^{\mathbf{Lat}}, n, \mathbf{t}, \mathbf{ts})$ **in the diffusion framework.** The inputs to the model $\epsilon_\theta$ include the noisy latent matrix $\mathbf{Z}_n^{\mathrm{Lat}}$ at the $n$th diffusion step, the diffusion step index $n$, the normalized time points $\mathbf{t}$, and the periodic timestamp embeddings $\mathbf{ts}$, projected through a multilayer perceptron (MLP). When conditional data $\mathbf{X}^{\mathrm{con}} = [c_{\mathrm{disc}}, c_{\mathrm{cont}}]$ is available, it is embedded using a word embedding (WE) for discrete variables and frequency-based representations (FR) for continuous variables, producing $\mathbf{Z}^{\mathrm{con}} := \mathbf{Emb}(\mathbf{X}^{\mathrm{con}})$. These embeddings are processed through a Bi-directional RNN (Bi-RNN) to capture temporal correlations, and the output is linearly projected and fused with $\mathbf{Z}_n^{\mathrm{Lat}}$ to produce $\mathbf{Z}_n'$. All components—$\mathbf{Z}_n'$, $n$, $\mathbf{t}$, and $\mathbf{ts}$—are passed through positional encodings and MLPs before concatenation. Here, $\bigoplus$ denotes matrix summation. The final block of RNNs, followed by layer normalization or fully connected layers (LN/FC), produces the diffusion stepwise prediction of noise $\epsilon_\theta(\mathbf{Z}_n^{\mathrm{Lat}}, n, \mathbf{t}, \mathbf{ts})$.

Conceptually, each column of $\mathbf{Z}_0^{\mathrm{Lat}}$ is treated as a discretized univariate time series in the latent space, and noise is added independently to each column. However, this does not preclude modeling cross-feature dependencies in $\mathbf{Z}_0^{\mathrm{Lat}}$ (Biloš et al., 2023). Although the forward diffusion corrupts columns independently, the reverse (denoising) process operates on the entire latent matrix simultaneously, capturing inter-feature correlations. A similar approach has been adopted in `TabDDPM` (Kotelnikov et al., 2022) for categorical data modeling. Under this formulation, we can write: $\mathbf{Z}_n^{\mathrm{Lat}} = \sqrt{\bar{\alpha}_n}\mathbf{Z}_0^{\mathrm{Lat}} + \sqrt{1 - \bar{\alpha}_n}\mathbf{E}^n$, where $\mathbf{E}^n := [\epsilon_1^n, \dots, \epsilon_F^n] \in \mathbb{R}^{T \times F}$ and each $\epsilon_j^n \sim \mathcal{N}(0, \mathcal{I}_{T \times T})$.

The training objective is the evidence lower bound (ELBO) defined as:

$$\mathcal{L}_{\mathrm{DM}} := \mathbb{E}_{n, \mathbf{E}^n}\left[\|\epsilon_\theta(\mathbf{Z}_n^{\mathrm{Lat}}, \mathbf{X}^{\mathrm{con}}, n, \mathbf{t}, \mathbf{ts}) - \mathbf{E}^n\|_2^2\right], \tag{4}$$

where $\epsilon_\theta$ is a neural network that predicts the noise $\mathbf{E}^n$ at a uniformly sampled diffusion step $n \sim \mathrm{Unif}\{1, \dots, N\}$.

Given the noisy latent matrix $\mathbf{Z}_n^{\mathrm{Lat}} \in \mathbb{R}^{T \times F}$, conditional data $\mathbf{X}^{\mathrm{con}} \in \mathbb{R}^{T \times F}$, the diffusion step $n \in \{1, \dots, N\}$, a normalized time vector $\mathbf{t} = \{t_1, \dots, t_T\}$ where $t_i = \frac{i}{T}$, and additional timestamp information $\mathbf{ts}$, goal of the denoiser $\epsilon_\theta$ is to predict the added noise $\mathbf{E}^n \in \mathbb{R}^{T \times F}$ at step $n$.

**Design of $\epsilon_\theta$:** First, we encode the diffusion step $n$ and the normalized time vector $\mathbf{t}$ using positional encodings (PE), as described in the Vaswani et al. (2017). These encodings make the model aware of the current diffusion progression and the temporal ordering of rows in $\mathbf{Z}_n^{\mathrm{Lat}}$.

Since $\mathbf{t}$ alone provides limited temporal context, we enhance it with cyclic date-time encodings for 'YEAR-MONTH-DATE-HOURS'. These cyclical encodings are computed via sine and cosine transformations:

$$\mathbf{CE}(\mathbf{ts}) := \left\{\left(\sin(\tfrac{\mathbf{x}}{\mathrm{Period} \times 2\pi}), \cos(\tfrac{\mathbf{x}}{\mathrm{Period} \times 2\pi})\right)\right\} : \mathrm{Period} \in \{\mathrm{YEAR}, \mathrm{MONTH}, \mathrm{DATE}, \mathrm{HOURS}\}\right\}.$$

and yield an 8-dimensional representation per timestamp. Here, Period is chosen based on the temporal granularity of the dataset (e.g., total number of years till today (for instance, 2025), 12 for months, 365 for days, and 24 for hours). This augmented timestamp information $\mathbf{ts}$ is projected through an MLP to match the dimension of the other encoded inputs. See Figure 5.

Similarly, if conditional data $\mathbf{X}^{\text{con}}$ is provided, it is processed through the same embedding procedures described in the VAE encoder (i.e., discrete features via lookup embeddings and continuous features via frequency-based encodings in equation 2), denoting $\mathbf{Z}^{\text{con}} := \mathbf{Emb}(\mathbf{X}^{\text{con}})$.

Next, we integrate the conditional embeddings $\mathbf{Z}^{\text{con}}$ with the noisy latent matrix $\mathbf{Z}_n^{\text{Lat}}$. The relation

$$\mathbf{Z}_n' = \text{Linear}(\mathbf{Z}_n^{\text{Lat}}) + \text{Linear}(\text{Bi-RNN}(\mathbf{Z}^{\text{con}}))$$

defines a fused latent representation $\mathbf{Z}_n'$. In this construction, $\mathbf{Z}_n^{\text{Lat}}$ encodes the corrupted latent factors at diffusion step $n$, while $\mathbf{Z}^{\text{con}}$ contains temporally aligned conditional embeddings that have been further processed through a Bi-directional RNN (Bi-RNN) (Schuster & Paliwal, 1997) to capture sequence-level dependencies among the conditional variables. Each of these two components—$\mathbf{Z}_n^{\text{Lat}}$ and Bi-RNN($\mathbf{Z}^{\text{con}}$)—is passed through its own linear mapping, ensuring both latent representations share a compatible feature space.

Next, we concatenate all encodings—$\mathbf{Z}_n'$, the positional encodings of $n$ and $\mathbf{t}$, and the augmented timestamp encodings from $\mathbf{ts}$—into a single tensor. This concatenated representation is passed through another MLP:

$$\mathbf{N} = \text{MLP}([\mathbf{Z}_n', \mathbf{PE}(n), \mathbf{PE}(\mathbf{t}), \mathbf{CE}(\mathbf{ts})]) \in \mathbb{R}^{T \times F}.$$

To capture temporal dependencies along the sequence length $T$, we feed $\mathbf{N}$ into a Bi-RNN, similar to the approach taken by Tian et al. (2023). The Bi-RNN processes the entire sequence $\{\mathbf{N}(t_1), \ldots, \mathbf{N}(t_T)\}$ forward and backward in time, producing a context-aware representation for each time step.

After applying layer normalization and a final fully-connected layer to the Bi-RNN outputs, $\epsilon_\theta$ generates:

$$[\epsilon_\theta(t_1), \ldots, \epsilon_\theta(t_T)]^\top \in \mathbb{R}^{T \times F},$$

which serves as the estimate of the noise $\mathbf{E}^n$ at the given diffusion step $n$.

### 4.5 Training, Inference & Computational Efficiency of `TimeAutoDiff`

In this section, we give the detailed procedures on training and sampling of `TimeAutoDiff`.

**Training:** While the training objectives for the VAE (i.e., $\mathcal{L}_{\text{VAE}}$) and the DDPM (i.e., $\mathcal{L}_{\text{DM}}$) are defined in equation 3 and equation 4, it remains unclear whether to train these models sequentially (Rombach et al., 2022) or in a joint manner (Vahdat et al., 2021). Recent research on latent DMs for image (Rombach et al., 2022), molecule (Xu et al., 2023), and tabular (Suh et al., 2023; Zhang et al., 2023b) generation indicates that a two-stage training approach usually achieves superior results, which we also observe in our experiments. Specifically, we first train the VAE with regularization and then train the latent DMs on the latent representations produced by the pre-trained encoder. A formal description of this procedure is provided in Algorithm 1.

**Inference:** A generative process, defined as $P_{\theta,\psi}(\mathbf{X}^{\text{tar}}, \mathbf{Z}_0^{\text{Lat}} \mid \mathbf{X}^{\text{con}}, \mathbf{M}^{\text{task}}) := P_\theta(\mathbf{Z}_0^{\text{Lat}} \mid \mathbf{X}^{\text{con}}) P_\psi(\mathbf{X}^{\text{tar}} \mid \mathbf{Z}_0^{\text{Lat}}, \mathbf{X}^{\text{con}}, \mathbf{M}^{\text{task}})$ operates in two steps. First, given $\mathbf{X}^{\text{con}}$, the trained diffusion prior $p_\theta$ samples a new latent matrix $\mathbf{Z}_0^{\text{Lat}}$. Next, the decoder takes $(\mathbf{X}^{\text{con}}, \mathbf{Z}_0^{\text{Lat}}, \mathbf{M}^{\text{task}})$ as input and generates a new sample, where $\mathbf{M}^{\text{task}} \in \mathcal{M}$ is a user-defined mask for specific time-series tasks. Finally, after post-processing, the final sample $\mathbf{X}^{\text{New}}$ is obtained. The sampling procedure is detailed in Algorithm 2.

**Computational Efficiency:** Its dominant cost arises from the $N$-step diffusion sampling, $O(NTH_{\text{diff}}^2)$, where $H_{\text{diff}}$ denotes the hidden size of the denoiser. By operating in a compressed latent space ($L \ll F$) and requiring only a moderate number of sampling steps ($N \approx 50$–$100$, in contrast to the $1000+$ steps used in data-space diffusion), our method attains a $10$–$100\times$ inference speed-up. In contrast, diffusion-based time-series synthesizers that rely on autoregressive sampling incur substantially higher computational cost—approximately $O(NT^2H_{\text{diff}}^2)$, since each timestamp must be generated sequentially with full prefix context. This efficiency advantage is obtained while preserving higher generative fidelity compared to single-step GAN approaches. See Appendix G for a complete breakdown.

---

**Algorithm 1** Training Algorithm of `TimeAutoDiff`

---

1: **Input:** $(\mathbf{X}^{\text{tar}}, \mathbf{X}^{\text{con}})$, $\mathbf{M}^{\text{task}}$, $\mathbf{ts}$, $\mathbf{t} = \left\{\frac{i}{T}\right\}_{i=1}^{T}$
2: **Initialize:** encoder network $\mathcal{E}_\phi$, decoder network $\mathcal{D}_\psi$, denoiser network $\epsilon_\theta$
3: **First Stage: VAE Training**
4: **while** $\phi$, $\psi$ have not converged **do**
5:     $\mu, \log \sigma^2 \leftarrow \mathcal{E}_\phi(\mathbf{X}^{\text{tar}}, \mathbf{X}^{\text{con}})$                                                 ▷ Encoding
6:     $\mathbf{E} \sim \mathcal{N}(0, \mathbf{I})$
7:     $\mathbf{Z}_0^{\text{Lat}} \leftarrow \boldsymbol{\mu} + \mathbf{E} \odot \boldsymbol{\Sigma}$                                          ▷ Reparameterization
8:     $\widehat{\mathbf{X}}^{\text{tar}} \leftarrow \mathcal{D}_\psi(\mathbf{X}^{\text{con}}, \mathbf{Z}_0^{\text{Lat}}, \mathbf{M}^{\text{task}})$                               ▷ Decoding
9:     $\mathcal{L}_{\text{VAE}} \leftarrow \ell_{\text{recons}}(\mathbf{X}^{\text{tar}}, \widehat{\mathbf{X}}^{\text{tar}}) + \beta \, \mathcal{D}_{\text{KL}}(q_\phi \,\|\, \mathcal{N}(0, \mathbf{I}))$
10:    $\phi, \psi \leftarrow \text{Optimizer}(\mathcal{L}_{\text{VAE}}; \phi, \psi)$
11:    **if** $\ell_{\text{recons}}$ has not improved for 10 epochs **then**
12:       $\beta \leftarrow \max(\beta_{\min}, 0.7\beta)$                                   ▷ Adaptive $\beta$ update
13:    **end if**
14: **end while**
15: **Second Stage: DDPM Training**
16: Fix $\phi, \psi$ and latent data $\mathbf{Z}_0^{\text{Lat}}$
17: **while** $\theta$ has not converged **do**
18:    $n \sim \text{Unif}\{1, \dots, N\}$, $\mathbf{E}^n \sim \mathcal{N}(0, \mathbf{I})$
19:    $\mathbf{Z}_n^{\text{Lat}} = \sqrt{\bar{\alpha}_n} \mathbf{Z}_0^{\text{Lat}} + \sqrt{1 - \bar{\alpha}_n} \mathbf{E}^n$
20:    $\mathcal{L}_{\text{DM}} = \|\epsilon_\theta(\mathbf{Z}_n^{\text{Lat}}, \mathbf{X}^{\text{con}}, n, \mathbf{t}, \mathbf{ts}) - \mathbf{E}^n\|_2^2$
21:    $\theta \leftarrow \text{Optimizer}(\mathcal{L}_{\text{DM}}; \theta)$
22: **end while**
23: **return** $(\mathcal{E}_\phi, \mathcal{D}_\psi, \epsilon_\theta)$

---

**Algorithm 2** Inference of `TimeAutoDiff`

---

1: **Input:** decoder $\mathcal{D}_\psi$, denoiser $\epsilon_\theta$, $\mathbf{X}^{\text{con}}$, $\mathbf{M}^{\text{task}}$, $\mathbf{ts}$, $\mathbf{t} = \{\frac{i}{T}\}_{i=1}^{T}$, $\mathbf{Z}_N^{\text{Lat}} \sim \mathcal{N}(0, \mathbf{I})$
2: **for** $n = N, N-1, \dots, 1$ **do**
3:     $\mathbf{z} \sim \mathcal{N}(0, \mathbf{I}).\text{reshape}(T, F)$                                     ▷ Latent denoising
4:     $\tilde{\epsilon}_\theta \leftarrow \epsilon_\theta(\mathbf{Z}_n^{\text{Lat}}, \mathbf{X}^{\text{con}}, n, \mathbf{t}, \mathbf{ts})$
5:     $\mathbf{Z}_{n-1}^{\text{Lat}} \leftarrow \frac{1}{\sqrt{\alpha_n}}\left(\mathbf{Z}_n^{\text{Lat}} - \frac{1-\alpha_n}{\sqrt{1-\bar{\alpha}_n}} \cdot \tilde{\epsilon}_\theta\right) + \beta_n \mathbf{z}$
6: **end for**
7: $\mathbf{X}^{\text{New}} \leftarrow \mathcal{D}_\psi(\mathbf{X}^{\text{con}}, \mathbf{Z}_0^{\text{Lat}}, \mathbf{M}^{\text{task}})$                                 ▷ Decoding
8: **return** $\tilde{\mathbf{X}}^{\text{New}} := \text{post-process}(\mathbf{X}^{\text{New}})$

---

# 5 Experiments

This section provides a thorough empirical evaluation of our proposed `TimeAutoDiff` model. We compare its performance against several state-of-the-art baselines across unconditional generation, imputation, forecasting, and TV-MCG tasks.

## 5.1 Unconditional Generation

This subsection evaluates the core capability of `TimeAutoDiff` to synthesize realistic time series data in an unconditioned setting. We assess the fidelity of generated samples using a comprehensive suite of quantitative metrics, including low-order statistics (e.g., feature and temporal correlations) and high-order statistics (e.g., statistical fidelity to the original data). Additionally, we evaluate the utility of the synthetic data for downstream predictive tasks, examine sampling efficiency. Next, we provide an ablation study which disentangles the contributions of individual model components to overall performance. Finally, we assess model generalizability using the Distance to the Closest Record (DCR).

### 5.1.1 Experimental Setting

**Datasets:** We use eight real-world time-series tabular datasets in this subsection. Among them, ETTh1 contains only numerical features, while the others include both numerical and categorical variables: Traffic, Pollution, Hurricane, AirQuality, Energy (single-sequence), NASDAQ100, and Card Fraud (multi-sequence). Appendix B provides details on how we preprocess both single- and multi-sequence tables for training, along with dataset-specific statistics. For all experiments in this section—except for qualitative and scalability analysis, where we use $T = 100 \sim 900$—we set the window size to $T = 48$ and stride to $S = 1$ (see Appendix B for details).

**Baselines:** To assess the quality of unconditionally generated time series data, we use 5 baseline models: (1) GAN based methods: `TimeGAN` (Yoon et al., 2019), `DoppelGANger` (Lin et al., 2020). (2) Diffusion based methods: `Diffusion-TS` (Yuan & Qiao, 2023), `TSGM` (Lim et al., 2023), (3) Parametric model: `CPAR` (Zhang et al., 2022).

**Evaluation Methods:** For the comprehensive quantitative evaluation of the synthesized data, we mainly focus on four criteria: (1) **Low-order statistic-** pair-wise column correlations and row-wise temporal dependences in the table are evaluated via *feature correlation score* (Kotelnikov et al., 2022) and *temporal discriminative score* (devised by us), respectively. (2) **High-order statistic-** the overall fidelities of the synthetic data in terms of joint distributional modeling are measured through *discriminative score* (Yoon et al., 2019). (3) The effectiveness of the synthetic data for **downstream** tasks is assessed through the predictive score (Yoon et al., 2019), where a predictive model (i.e., regressor or classifier) is trained using synthesized data and tested on real data (Mogren, 2016). (4) **Sampling times** (in sec.) are compared with other baseline methods. Detailed explanations for each metric are deferred in the Appendix E. (5) **generalizability** of the model is evaluated under "Distance to the Closest Record" (DCR; Park et al. (2018)) metric to ensure it draws samples from the distribution rather than memorizing the training data points.

**Model Parameter Configuration:** In Appendix F, we present the parameter configurations of VAE and DDPM in our model. Unless otherwise specified, they are universally applied to the entire dataset in the experiments conducted in this paper. Additionally, we study how the sizes of network architectures in DDPM and VAE, training epochs for both models, and noise schedulers (linear vs quadratic) in DDPM affect the performances of the model.

### 5.1.2 Unconditional Generation

**Experimental Results:** Table 2 shows that our `TimeAutoDiff` consistently outperforms other baseline models in almost all metrics both for single- and multi-sequence generation tasks. It significantly improves the (temporal) discriminative and feature correlation scores in all datasets over the baseline models. `TimeAutoDiff` also dominates the predictive score metric. (We train a classifier to predict a column in the dataset to measure the predictive score. The columns predicted in each dataset are listed in Table 4 in Appendix B.) But for some datasets, the performance gaps with the second-best model are negligible: e.g., `TimeAutoDiff` vs TSGM for *Hurricane* and *AirQuality* datasets. It is intriguing to note that the predictive scores can be good even when the data fidelity is low. The GAN-based models are faster in terms of sampling time compared to the diffusion-based models. These results are expected, as diffusion-based models require multiple denoising steps for sampling, whereas GAN-based models generate samples in a single step. Nonetheless, among diffusion-based models, our approach achieves the fastest sampling time—approximately 90 to 100 times faster than existing methods. In the Appendix H and I, we provide additional experiments on more metrics such as volatility, moving averages, Maximum Mean Discrepancy (MMD) and entropy for diversity (Nikitin et al., 2023).

**Qualitative Illustrations of Temporal & Feature Dependences:** Aside from the quantitative evaluations under the aforementioned metrics, we provide the qualitative analysis in Fig. 6 further demonstrating the effectiveness of `TimeAutoDiff`. The autocorrelation plots for both real datasets (AirQuality and ETTh1) and their synthetic counterparts reveal that `TimeAutoDiff` successfully captures complex and long-range temporal dependencies—extending up to $T = 500$ time steps. Additionally, the similar patterns observed for each feature in the real and synthetic data demonstrate the model's ability to capture the correlations along the feature dimension. We provide more visualizations across more various datasets in the Appendix L.

| Metric | Model | Single-Sequence | | | | Multi-Sequence | |
|---|---|---|---|---|---|---|---|
| | | Traffic | Pollution | Hurricane | AirQuality | Card Transaction | nasdaq100 |
| Discriminative Score (The lower, the better) | TimeAutoDiff | **0.026(0.014)** | **0.016(0.009)** | **0.047(0.016)** | **0.061(0.013)** | **0.215(0.058)** | **0.067(0.046)** |
| | Diffusion-ts | 0.202(0.021) | 0.133(0.015) | 0.181(0.018) | 0.134(0.016) | N.A. | N.A. |
| | TSGM | 0.500(0.000) | 0.488(0.010) | 0.482(0.020) | 0.452(0.009) | N.A. | N.A. |
| | TimeGAN | 0.413(0.057) | 0.351(0.053) | 0.254(0.062) | 0.460(0.020) | 0.482(0.037) | 0.267(0.115) |
| | DoppelGANger | 0.258(0.215) | 0.100(0.103) | 0.176(0.099) | 0.211(0.116) | 0.485(0.025) | 0.071(0.032) |
| | CPAR | 0.498(0.002) | 0.500(0.000) | 0.500(0.000) | 0.499(0.001) | 0.500(0.000) | 0.143(0.120) |
| | Real vs Real | 0.053(0.009) | 0.048(0.017) | 0.034(0.011) | 0.040(0.011) | 0.225(0.094) | 0.190(0.051) |
| Predictive Score (The lower, the better) | TimeAutoDiff | **0.203(0.014)** | **0.008(0.000)** | **0.098(0.026)** | **0.005(0.001)** | **0.001(0.000)** | 10.863(0.716) |
| | Diffusion-ts | 0.231(0.007) | 0.013(0.000) | 0.306(0.076) | 0.017(0.002) | N.A. | N.A. |
| | TSGM | 0.247(0.002) | 0.009(0.000) | 0.290(0.007) | 0.006(0.000) | N.A. | N.A. |
| | TimeGAN | 0.297(0.008) | 0.043(0.000) | 0.180(0.027) | 0.057(0.011) | 0.130(0.022) | 9.597(0.016) |
| | DoppelGANger | 0.300(0.005) | 0.282(0.028) | 0.214(0.000) | 0.060(0.009) | 0.004(0.006) | 11.556(1.093) |
| | CPAR | 0.263(0.003) | 0.032(0.009) | 0.420(0.055) | 0.030(0.007) | 0.132(0.035) | **8.270(0.019)** |
| | Real vs Real | 0.206(0.012) | 0.010(0.000) | 0.098(0.026) | 0.005(0.001) | 0.001(0.000) | 9.281(0.009) |
| Temporal Discriminative Score (The lower, the better) | TimeAutoDiff | **0.047(0.018)** | **0.014(0.013)** | **0.026(0.024)** | **0.033(0.014)** | **0.290(0.040)** | 0.159(0.140) |
| | Diffusion-ts | 0.199(0.028) | 0.165(0.084) | 0.247(0.093) | 0.183(0.064) | N.A. | N.A. |
| | TSGM | 0.499(0.001) | 0.499(0.001) | 0.497(0.002) | 0.499(0.000) | N.A. | N.A. |
| | TimeGAN | 0.429(0.050) | 0.397(0.060) | 0.465(0.025) | 0.457(0.014) | 0.497(0.007) | 0.419(0.140) |
| | DoppelGANger | 0.400(0.039) | 0.444(0.050) | 0.464(0.028) | 0.335(0.091) | 0.362(0.097) | 0.497(0.007) |
| | CPAR | 0.436(0.073) | 0.492(0.021) | 0.497(0.009) | 0.493(0.010) | 0.470(0.041) | 0.404(0.099) |
| | Real vs Real | 0.061(0.011) | 0.044(0.009) | 0.039(0.012) | 0.050(0.017) | 0.360(0.051) | 0.150(0.090) |
| Feature Correlation Score (The lower, the better) | TimeAutoDiff | **0.022(0.014)** | 1.244(0.844) | **0.074(0.013)** | **0.463(0.080)** | 0.078(0.137) | 0.243(0.012) |
| | Diffusion-ts | 2.148(1.439) | 1.716(1.096) | 1.881(1.208) | 0.716(0.141) | N.A. | N.A. |
| | TSGM | 2.092(1.485) | 1.710(0.705) | 0.424(0.249) | 0.543(0.077) | N.A. | N.A. |
| | TimeGAN | 1.243(0.535) | 2.068(1.093) | 2.151(1.113) | 0.865(0.123) | 2.301(0.723) | 1.488(1.069) |
| | DoppelGANger | 0.885(0.737) | 2.371(0.875) | 2.380(0.798) | 1.628(0.231) | 1.550(1.034) | 1.035(0.818) |
| | CPAR | 0.538(0.336) | 1.280(0.931) | 0.965(0.287) | 1.552(0.220) | 0.295(0.294) | 0.514(0.445) |
| | Real vs Real | 0.000(0.000) | 0.000(0.000) | 0.000(0.000) | 0.000(0.000) | 0.000(0.000) | 0.000(0.000) |
| Sampling Time (in Sec) (The lower, the better) | TimeAutoDiff | 3.512 (0.065) | 3.947 (0.070) | 3.740 (0.132) | 3.945 (0.103) | 3.384(0.064) | 3.133(0.129) |
| | Diffusion-ts | ≫ | ≫ | ≫ | ≫ | N.A. | N.A. |
| | TSGM | ≫ | ≫ | ≫ | ≫ | N.A. | N.A. |
| | TimeGAN | 0.127(0.056) | 0.113(0.058) | 0.125(0.060) | 0.131(0.060) | 0.051(0.051) | 0.047(0.039) |
| | DoppelGANger | **0.011(0.002)** | **0.014(0.001)** | **0.010(0.003)** | **0.017(0.003)** | **0.018(0.004)** | **0.041(0.001)** |
| | CPAR | 17.466(0.734) | 18.597(0.558) | 15.839(0.324) | 29.816(0.846) | 141.425(2.435) | 112.506(2.152) |

Table 2: Experimental results for single- and multi-sequence time series tabular data generation are reported under Discriminative, Predictive, Temporal Discriminative, and Feature Correlation metrics. Sampling times (in Secs) over 6 datasets are included; ≫ indicates times exceeding 300 seconds, and 'N.A.' denotes not applicable. Bold numbers indicate the best performance. Each metric shows the mean and standard deviation (in parentheses) over 10 synthetic datasets generated by the trained model. 'Real Data' serves as a baseline with each metric computed on Real vs. Real.

**Scalability of `TimeAutoDiff` along features axis:** To validate the scalability of our approach along feature axis, particularly the benefit of the VAE's latent feature compression, we conducted experiments on high-dimensional synthetic datasets (see Appendix J for full results). These tests demonstrate that `TimeAutoDiff` effectively models long sequences (up to T=900) with high fidelity when the feature count is low (F=5). Conversely, when increasing the feature dimension up to F=50 (at T=200), performance noticeably degrades without compression. However, by setting the latent feature dimension to $F/2$, the model maintained significantly better generation quality in this high-dimensional setting. This empirically confirms that the VAE's feature compression is a key component for efficiently modeling wide, high-dimensional time-series tables.

**Ablation:** The ablation test results are summarized in Table 3. A single model alone (i.e., only VAE or DDPM) cannot accurately capture the statistical properties of the distributions of tables, which strongly supports the motivation of our model. The components related to the diffusion model, such as timestamp encoding and Bi-RNN, impact the generative performance across most cases as models lacking these components do not exhibit optimal performance. The encodings for continuous features in the VAE notably enhance the fidelity and temporal dependences of the generated data.

Additionally, we consider the following scenarios:

1. Replacing the MLP with an RNN in the decoder of the VAE.

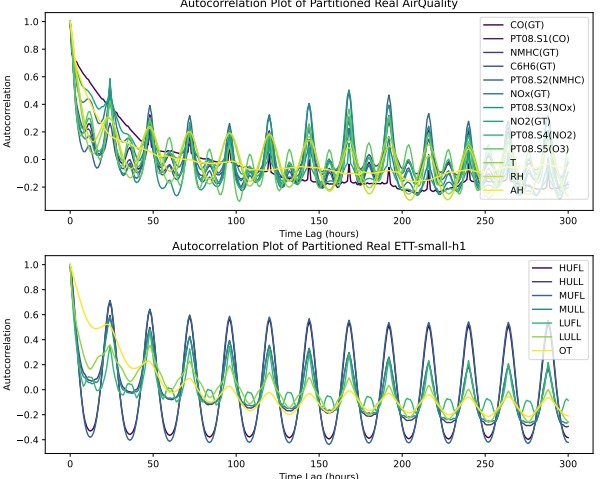
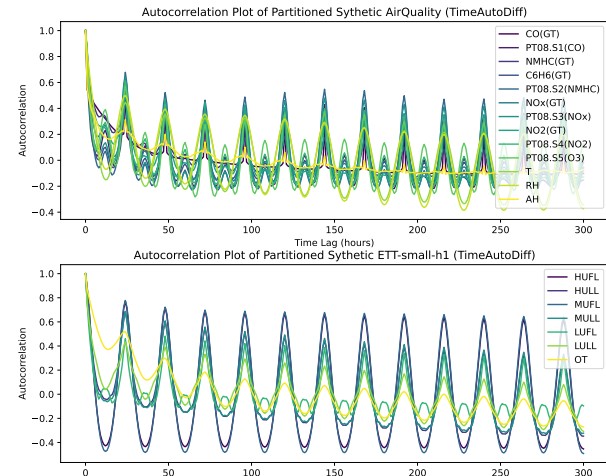

Figure 6: **Real (left)** vs. **Synthetic (right)**: Autocorrelation plots with a time lag of 300 (hours) for the AirQuality (top) and the ETTh1 (bottom) datasets. Sequence length is set as $T = 500$.

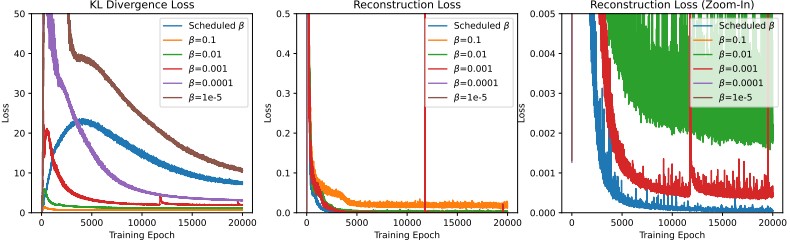

| $\beta$ | Disc. Score |
|---|---|
| $10^{-1}$ | 0.369(0.101) |
| $10^{-2}$ | 0.041(0.011) |
| $10^{-3}$ | 0.043(0.019) |
| $10^{-4}$ | 0.079(0.012) |
| $10^{-5}$ | 0.043(0.009) |
| Scheduled $\beta$ | **0.023(0.015)** |

Figure 7: KL-Divergence (left) and Reconstruction (middle) losses over 20000 training iterations of VAE on Traffic dataset. The zoomed-in panel (right) displays the scheduled-$\beta$ reaches the lowest reconstruction error stably without any spikes.

Table 4: The results of discriminative scores with varying $\beta$ values on the Traffic dataset.

2. Replacing the two RNNs with an MLP in the encoder of the VAE.

3. Inspired by Biloš et al. (2023), we explore injecting continuous noise from a stochastic process (Gaussian process) into the DDPM. Specifically, the perturbation kernel

$$q(\mathbf{z}_{n,j}^{\text{Lat}}|\mathbf{z}_{0,j}^{\text{Lat}}) = \mathcal{N}(\sqrt{\bar{\alpha}_n}\mathbf{z}_{0,j}^{\text{Lat}}, (1 - \bar{\alpha}_n)\mathbf{\Sigma})$$

is applied independently to each column of $\mathbf{Z}_0^{\text{Lat}} \in \mathbb{R}^{T \times F}$, where $\mathbf{\Sigma}_{ij} = \exp(-\gamma|\mathbf{t}_i - \mathbf{t}_j|)$ with $\gamma = 0.2$.

The experimental results indicate that none of the ablated models significantly outperformed the original configuration. In particular, the second configuration highlights the potential benefits of modeling temporal dependencies at two stages—within both the VAE and the DDPM. While the precise mechanisms remain to be further validated, we hypothesize the following contributing factors: (1) *Hierarchical Temporal Dependency Modeling:* The VAE encoder captures compact latent representations that preserve temporal structure, providing a well-organized foundation for the diffusion model. This hierarchical setup may allow the diffusion process to focus on refining fine-grained temporal patterns, rather than redundantly learning high-level structures, leading to more realistic outputs. (2) *Noise-Tolerant Latent Representation:* Incorporating temporal dependencies early in the VAE may yield a latent variable $\mathbf{Z}_0^{\text{Lat}}$ that is inherently more robust to noise. This resilience could help preserve critical temporal information during the diffusion process, ultimately improving the fidelity of the generated sequences.

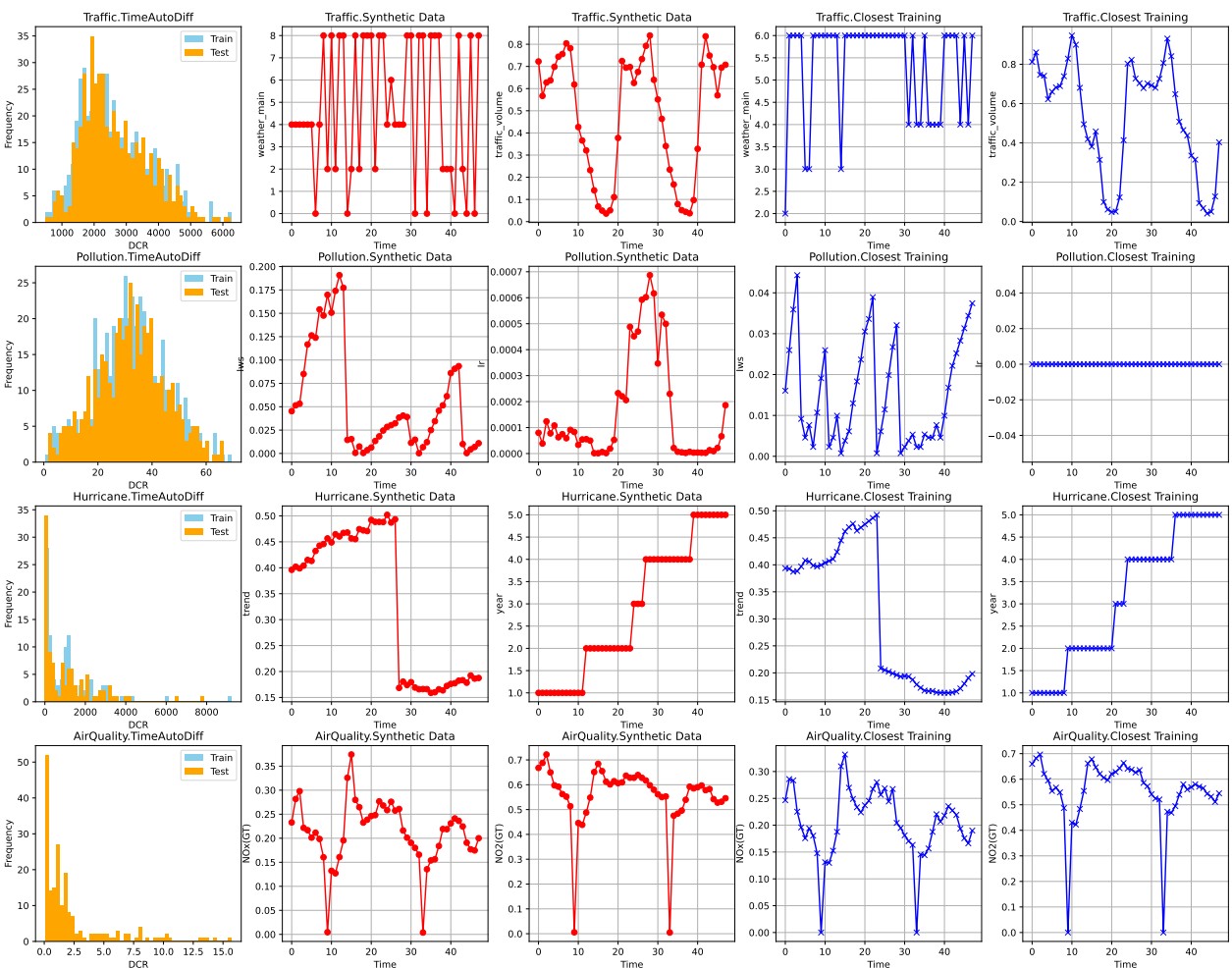

Figure 8: The leftmost column plots the empirical distributions of distance-to-closest-record (DCR) for the training and test splits across four datasets (top to bottom: *Traffic*, *Pollution*, *Hurricane*, *AirQuality*). For each dataset, we pick two representative variables: the second and third columns show these variables over time for one synthetic sample ($T = 48$), and the fourth and fifth columns show the same variables for its nearest training record (by DCR). On *Traffic* and *Pollution*, the DCR mass lies noticeably away from zero and the synthetic trajectories visibly diverge from their nearest neighbors, indicating new patterns rather than replicas. On *Hurricane* and *AirQuality*, the DCR concentrates near zero and the synthetic trajectories closely track the nearest training records, suggesting replication.

**Memorization versus generalization:** Consistent with the phenomenon reported by (Zhang et al., 2023c), `TimeAutoDiff` exhibits data-regime–dependent behavior: when effective model capacity exceeds the available data (data-poor regime), memorization is more likely; when the data budget dwarfs model capacity (memorization regime), generalization emerges. With model size fixed in our study, *Traffic* and *Pollution* provide over 20K training sequences (generalization regime), aligning with the nonzero DCR and novel trajectories, whereas *Hurricane* and *AirQuality* provide fewer than 5K (data-poor), aligning with near-zero DCR and training-like generations.

| Metric | Model | Traffic | Pollution | Hurricane | AirQuality |
|---|---|---|---|---|---|
| Discriminative Score 

 (The lower, the better) | TimeAutoDiff | 0.027(0.014) | **0.014(0.011)** | **0.035(0.010)** | 0.035(0.016) |
| | only VAE | 0.476(0.010) | 0.491(0.010) | 0.490(0.010) | 0.494(0.007) |
| | only DDPM | 0.283(0.131) | 0.313(0.163) | 0.252(0.034) | 0.266(0.048) |
| | w/o Encoding equation 1 | 0.029(0.017) | 0.062(0.015) | 0.063(0.018) | 0.072(0.020) |
| | w/o Timestamps | 0.095(0.016) | 0.105(0.012) | 0.171(0.085) | 0.074(0.013) |
| | w/o Bi-directional RNN | 0.049(0.015) | 0.021(0.020) | 0.300(0.036) | **0.019(0.015)** |
| | RNN in decoder (VAE) | 0.186(0.019) | 0.185(0.020) | 0.198(0.031) | 0.124(0.018) |
| | MLP in encoder (VAE) | 0.017(0.011) | 0.072(0.020) | 0.117(0.019) | 0.067(0.025) |
| | Smooth Noise (DDPM) | **0.015(0.009)** | 0.078(0.013) | 0.140(0.016) | 0.140(0.016) |
| Predictive Score 

 (The lower, the better) | TimeAutoDiff | 0.229(0.010) | **0.008(0.000)** | 3.490(0.097) | **0.004(0.000)** |
| | only VAE | 0.241(0.001) | 0.008(0.000) | 4.566(0.041) | 0.019(0.002) |
| | only DDPM | 0.241(0.012) | 0.016(0.000) | **0.034(0.007)** | 0.009(0.002) |
| | w/o Encoding equation 1 | **0.219(0.011)** | 0.008(0.000) | 3.611(0.216) | 0.005(0.000) |
| | w/o Timestamps | 0.241(0.003) | 0.008(0.000) | 4.228(0.248) | 0.004(0.000) |
| | w/o Bi-directional RNN | 0.231(0.008) | 0.008(0.000) | 3.549(0.047) | 0.004(0.000) |
| | RNN in decoder (VAE) | 0.232(0.008) | 0.008(0.000) | 3.598(0.095) | 0.012(0.004) |
| | MLP in encoder (VAE) | 0.220(0.011) | 0.008(0.000) | 3.365(0.072) | 0.061(0.002) |
| | Smooth Noise (DDPM) | 0.221(0.011) | 0.008(0.000) | 0.091(0.027) | 0.059(0.001) |
| Temporal Discriminative Score 

 (The lower, the better) | TimeAutoDiff | 0.047(0.017) | **0.008(0.005)** | **0.020(0.010)** | 0.035(0.024) |
| | only VAE | 0.368(0.107) | 0.484(0.043) | 0.490(0.014) | 0.493(0.006) |
| | only DDPM | 0.197(0.127) | 0.135(0.131) | 0.213(0.096) | 0.242(0.122) |
| | w/o Encoding equation 1 | 0.036(0.016) | 0.052(0.019) | 0.049(0.022) | **0.008(0.005)** |
| | w/o Timestamps | 0.084(0.047) | 0.053(0.018) | 0.117(0.065) | 0.064(0.019) |
| | w/o Bi-directional RNN | 0.031(0.021) | 0.047(0.057) | 0.404(0.013) | 0.023(0.015) |
| | RNN in decoder (VAE) | 0.130(0.025) | 0.133(0.019) | 0.324(0.072) | 0.331(0.130) |
| | MLP in encoder (VAE) | 0.037(0.017) | 0.060(0.018) | 0.094(0.019) | 0.045(0.032) |
| | Smooth Noise (DDPM) | **0.020(0.007)** | 0.059(0.029) | 0.090(0.027) | 0.091(0.027) |
| Feature Correlation Score 

 (The lower, the better) | TimeAutoDiff | **0.022(0.014)** | **1.104(0.900)** | **0.069(0.027)** | **0.147(0.230)** |
| | only VAE | 0.404(0.339) | 1.329(0.757) | 0.427(0.371) | 0.702(1.001) |
| | only DDPM | 2.238(1.530) | 2.020(1.460) | 2.380(1.513) | 0.198(0.298) |
| | w/o Encoding equation 1 | 0.029(0.021) | 1.148(0.850) | 0.077(0.034) | 0.266(0.405) |
| | w/o Timestamps | 0.247(0.521) | 1.303(0.793) | 0.097(0.044) | 0.231(0.349) |
| | w/o Bi-directional RNN | 0.048(0.024) | 1.227(0.863) | 0.090(0.043) | 0.155(0.256) |
| | RNN in decoder (VAE) | 0.413(0.544) | 1.187(0.820) | 0.247(0.123) | 0.913(1.302) |
| | MLP in encoder (VAE) | 0.025(0.015) | 1.240(0.853) | 0.122(0.058) | 1.217(1.745) |
| | Smooth Noise (DDPM) | 0.059(0.037) | 1.246(0.843) | 0.882(1.271) | 1.215(1.345) |

Table 3: The experimental results of ablation test in `TimeAutoDiff`. The bolded number indicates the best-performing model.

**The effect of adaptive $\beta$-VAE:** Motivated from (Zhang et al., 2023b), we evaluate the effects of scheduling on $\beta$ coefficients in VAE in terms of tradeoffs between reconstruction error and KL-divergence. In Fig 7, we observe that while large $\beta$ can ensure the close distance between the embedding and standard normal distributions, its reconstruction loss is relatively larger than that of smaller ones, and vice versa. The adaptive $\beta$-scheduling ensures both the lowest reconstruction error and relatively lower KL-divergence, preserving the shape of embedding distribution. The adaptive $\beta$-scheduling achieves the fastest and the most stable signal reconstructions among other $\beta$-choices. Table 4 shows the effectiveness of $\beta$-scheduling for quality of synthetic data in discriminative score.

**Generalizability of `TimeAutoDiff`:** In generative modeling, it is essential to check whether the learned model can generate the datasets not seen in the training set. If model memorizes and reproduces data points from the training dataset (Zhang et al., 2023c), this can undermine the primary motivation of data synthesizing, which is *'increasing dataset diversity'*. To investigate further in this regard, we design an experiment using the notion of Distance to the Closest Record (DCR) (Park et al., 2018), which computes the Euclidean distance between a data point $r \in \mathbb{R}^{T \times F}$ in the synthesized dataset and the closest record to $r$ in the original table. We split the data into training (50%) / testing (50%) sets.

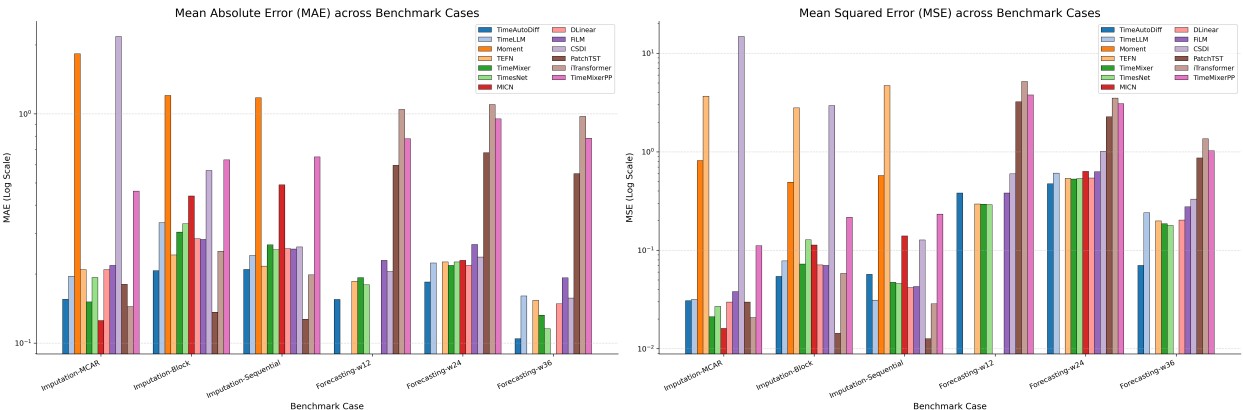

Figure 9: **Dataset-averaged benchmark performance across tasks.** Bars indicate the mean error across four datasets (*Bike Sharing*, *Traffic*, *Pollution*, and *AirQuality*) for each model and benchmark case, with the total sequence length fixed at $T = 48$. **Left:** Mean Absolute Error (MAE, logarithmic scale). **Right:** Mean Squared Error (MSE, logarithmic scale). The six benchmark cases (from left to right on the $x$-axis) are: *Imputation-MCAR*, *Imputation-Block* (`block_len`=3, `block_width`=2), *Imputation-Sequential* (`seq_len`=12), and forecasting with look-back windows $w \in \{12, 24, 36\}$. For baselines lacking reported results in the original tables (e.g., see Table K in the Appendix for missing `TimeLLM`, `Moment`, `MICN`, and `DLinear` entries over $w \in \{12, 24, 36\}$, bars are shown with zero height); means are computed while ignoring NaN values. A logarithmic y-axis is used for MAE and MSE to reduce the dominance of large errors and to reveal relative differences among the remaining cases.

DCR scores for both training and testing datasets can be used to evaluate the model's performance. Significant overlap between the DCR distributions of the training and testing datasets suggests that the model is drawing data from the data distribution. However, even with substantial overlap between the distributions, if the distances to the origin are small, this suggests that the patterns in the training and testing sets are alike, implying the model may have memorized specific training data points. If the DCR distribution of the training data is notably closer to zero compared to the testing data, it indicates that the model has memorized the training dataset. Last but not least, it's important to recognize that random noise can also produce similar DCR distributions. Therefore, the DCR score should be evaluated in conjunction with other measures of fidelity (i.e., discriminative score), and utility measures (i.e., predictive score), to provide a comprehensive assessment of the model's generalization capabilities. We provide the interpretations of DCR distributions of `TimeAutoDiff` for *Traffic*, *Pollution*, *Hurricane*, and *AirQuality* datasets in the caption of Fig. 8.

## 5.2 Imputation & Forecasting

In this subsection, we present experimental results on two conditional generation tasks: *Imputation* and *Forecasting*.

**Extended Evaluation Settings:** We evaluate `TimeAutoDiff` across a comprehensive suite of six benchmark cases, encompassing three imputation regimes and three forecasting horizons. All experiments are conducted on four real-world tabular time series datasets—*Bike Sharing*, *Traffic*, *Pollution*, and *AirQuality*—each of which contains both numerical and categorical variables. The sequence length is fixed to $T = 48$ for all tasks to ensure comparability. For imputation, we simulate three types of missingness using the binary mask $\mathbf{M}^{\mathrm{I}}_{t,f} = \mathbb{1}(\mathbf{X}_{t,f}$ is missing): (1) MCAR (Missing Completely At Random), where entries are masked uniformly; (2) Block Missingness, where rectangular regions over time and features simulate sensor dropouts; and (3) Sequential Missingness, where temporally contiguous subsequences emulate localized corruption. For the forecasting task, we vary the look-back window size $w \in \{12, 24, 36\}$, defining the forecasting mask as $\mathbf{M}^{\mathrm{F}}_{t,f} = \mathbb{1}(t > w)$. This setup evaluates the model's ability to extrapolate across both long- and short-range temporal horizons, while maintaining a fixed total sequence length $T = 48$.

**Baselines:** To evaluate the performance of `TimeAutoDiff` on imputation and forecasting tasks, we compare it against nine established baseline models: `TimeLLM` (Jin et al., 2023), `Moment` (Goswami et al., 2024), `TEFN` (Zhan et al., 2024), `TimeMixer` (Wang et al., 2024b), `TimesNet` (Wu et al., 2022), `MICN` (Wang et al., 2023a), `DLinear` (Zeng et al., 2023), `FiLM` (Zhou et al., 2022), `CSDI` (Tashiro et al., 2021), `PatchTST` (Nie et al., 2022), `iTransformer` (Liu et al., 2023), and `TimeMixer++` (Wang et al., 2024a). All selected models are designed to support both imputation and forecasting tasks. For consistency and reproducibility, we utilize the `PyPOTS` framework (Du, 2023), which provides unified implementations of all baseline models. Detailed descriptions of each baseline model are provided in the Appendix D.2. The hyperparameter settings for each model are specified in the `ImputeEval.py` file, which is available in our public code repository.

**Evaluation Methods:** For the experiments, we split the dataset into training (70%) / validation (20%) / testing (10%) sets. We independently select the best-performing checkpoint for the VAE and DDPM based on validation error. For both imputation and forecasting tasks, we use two standard metrics: Mean Absolute Error (MAE) and Mean Squared Error (MSE) for performance evaluations.

**Quantitative Summary:** Based on the quantitative results, `TimeAutoDiff` demonstrates strong overall performance, achieving a top-2 score in 31 out of the 48 total benchmark settings (4 datasets, 6 tasks, 2 metrics). This includes 23 state-of-the-art (SOTA) scores. The model's primary strength is in forecasting, where it secures 13 out of 24 SOTA metrics, plus an additional 6 second-best scores. In contrast, its performance in imputation is more specialized, achieving 10 out of 24 SOTA metrics (plus 2 second-best scores), primarily concentrated in the *Traffic* and *Pollution* datasets. The model's performance is strongest on the *Traffic* dataset, where it leads in 10 out of 12 total tasks. Conversely, it shows its weakest comparative performance in the *Bike Sharing* imputation tasks, where it secures no SOTA or second-best results, being outperformed by `PatchTST`, `iTransformer`, and `MICN`.

**Result Interpretations:** Our interpretation is that `TimeAutoDiff`'s architecture is highly effective at modeling complex, heterogeneous, and non-stationary temporal dynamics, which is critical for its strong performance in forecasting. This advantage is most pronounced on the volatile *Traffic* dataset. The imputation tasks, however, particularly on the highly structured *Bike Sharing* dataset, appear to reward different architectural specializations. Models like `PatchTST`, which utilize patching, excel at modeling local patterns, making them robust to structured (Block/Sequential) missingness. Meanwhile, models like `iTransformer` and `MICN`, which focus on cross-channel correlations, are better suited to impute missing values by referencing other related features (e.g., weather, season) present at the same timestamp.

### 5.3 Time Varying-Metadata Conditional Generation (TV-MCG)

We evaluate `TimeAutoDiff` on the Time-Varying Metadata Conditional Generation (TV–MCG) setting, where a metadata trajectory $\mathbf{X}^{\text{con}}$ is fixed and the target sequence $\mathbf{X}^{\text{tar}}$ is generated conditionally. Note that $\mathbf{X}^{\text{con}}$ corresponds to a subset of features from the full dataset $\mathbf{X}$, selected along the feature axis, characterized by a mask $\mathbf{M}^{\text{M}}$. Our study is organized into two complementary parts:

| Metric | Methods | Single-Sequence | | | | | |
|---|---|---|---|---|---|---|---|
| | | Traffic | Pollution | Hurricane | AirQuality | ETTh1 | Energy |
| Discriminative Score | TimeAutoDiff | **0.078(0.038)** | **0.056(0.017)** | **0.014(0.005)** | 0.090(0.007) | **0.036(0.008)** | **0.113(0.070)** |
| | Real vs Real | 0.091(0.021) | 0.067(0.020) | 0.081(0.009) | **0.085(0.027)** | 0.051(0.011) | 0.270(0.028) |
| Predictive Score | TimeAutoDiff | 0.113(0.007) | **0.008(0.000)** | 0.060(0.009) | **0.004(0.000)** | **0.048(0.002)** | **0.228(0.005)** |
| | Real vs Real | **0.107(0.001)** | 0.008(0.000) | **0.058(0.010)** | 0.004(0.000) | 0.051(0.001) | 0.230(0.003) |
| Temporal Discriminative Score | TimeAutoDiff | **0.123(0.034)** | **0.081(0.027)** | **0.048(0.025)** | **0.116(0.018)** | **0.045(0.015)** | **0.224(0.013)** |
| | Real vs Real | 0.134(0.015) | 0.083(0.019) | 0.072(0.019) | 0.138(0.014) | 0.074(0.014) | 0.300(0.031) |
| Feature Correlation Score | TimeAutoDiff | **0.012(0.003)** | **0.026(0.008)** | **0.175(0.032)** | **0.011(0.002)** | **0.014(0.002)** | **0.029(0.007)** |
| | Real vs Real | 0.000(0.000) | 0.000(0.000) | 0.000(0.000) | 0.000(0.000) | 0.000(0.000) | 0.000(0.000) |

Table 5: Time varying metadata conditional generations: the experiments conducted over 6 single-sequence datasets with sequence length set as $T = 96$. See the caption of Figure 15 (Appendix M) for output and condition pairs for each dataset used for the experiments. Overall, `TimeAutoDiff` performs well, achieving results comparable to the Real vs Real baseline over the test dataset.

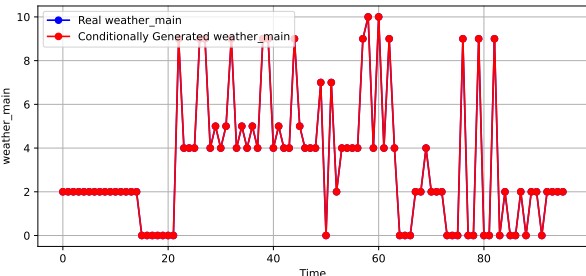 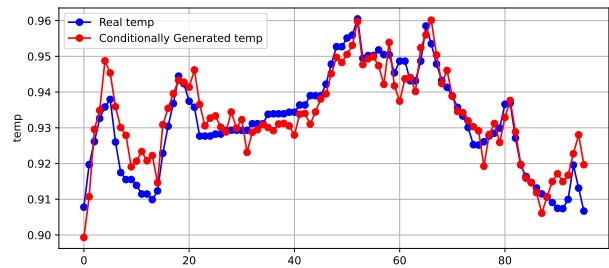

Figure 10: Datasets: (output variables) from top to bottom: ***Traffic***: ('Weather main', 'temp'), The output is chosen to be heterogeneous both having discrete and continuous variables. Conditional variables are set as remaining variables from the entire features. See the list of entire features of the dataset through the link in Appendix B.

**(i) Quantitative assessment:** We evaluate `TimeAutoDiff` on multiple public datasets using four fidelity and dynamics-based metrics, comparing against an oracle Real-vs-Real proxy. **(ii) Counterfactual scenario exploration:** To verify that the model learns meaningful conditional structure, we examine both synthetic settings with known rules and real-world Traffic data, generating counterfactual trajectories under alternative metadata conditions. **(iii) Robustness of generative model:** Finally, we study robustness by introducing controlled metadata corruption (noise for continuous variables, random flips for categorical ones) and measuring how fidelity metrics change as corruption increases. This reveals how reliably the model preserves conditional dynamics under noisy or imperfect metadata.

### 5.3.1 Quantitative assessment with public data

**Baselines.** Few prior models address conditional tabular time-series generation with fixed side information. To our knowledge, `TimeWeaver` (Narasimhan et al., 2024) is conceptually the closest, but it is limited to generating only continuous time series data. Furthermore, no public implementation of the model is available. Accordingly, we report a Real vs Real proxy baseline as an oracle reference and compare it against our conditional generator.

**Evaluation Methods:** To evaluate the generalization capability of `TimeAutoDiff` to unseen conditions, we randomly split each dataset into training and test sets using an 80%/20% ratio. Synthetic samples are then generated to match the size of the test set, with the sequence length fixed at $T = 96$. The quality of the generated data is assessed using the 4 evaluation metrics: *Discriminative Score*, *Predictive Score*, *Temporal Discriminative Score*, and *Feature correlation score*.

**Quantitative Results.** Table 5 shows that `TimeAutoDiff` matches or improves upon the Real vs Real oracle across datasets. Discriminative and temporal–discriminative scores are consistently lower, indicating high conditional fidelity and well-preserved dynamics. Predictive scores are on par with or slightly better than the oracle, confirming that synthetic data retain task-relevant signal. Feature-correlation deviations remain small, evidencing faithful cross-feature structure under fixed metadata. Overall, the model reliably captures both fidelity and conditional coupling across easy (periodic) and challenging (high-variance) settings.

**Dataset-level Observations.** Datasets with strong exogenous structure or seasonal drivers—*ETTh1*, *AirQuality*—show very low discriminative and predictive gaps, reflecting that `TimeAutoDiff` reliably reproduces condition-aligned periodic and cross-feature patterns. Challenging, high-variance settings such as *Hurricane* and *Energy* still yield markedly low discriminative and temporal discriminative scores, underscoring the model's capacity to respect conditional dynamics even when the marginal variability is large. On *Traffic* and *Pollution*, the method remains close to (or better than) Real vs Real across all metrics, evidencing accurate conditional coupling between environmental or load conditions $\mathbf{X}^{\mathrm{con}}$ and target responses $\mathbf{X}^{\mathrm{tar}}$.

### 5.3.2 Counterfactual scenario exploration

We further provide numerical validations that `TimeAutoDiff` indeed learn the conditional distribution $\mathbb{P}(\mathbf{X}^{\mathrm{tar}}|\mathbf{X}^{\mathrm{con}}, \mathbf{M}^{\mathrm{M}})$ of both $\mathbb{P}(\text{'Cont Var.'}|\text{'Disc Var.'})$ and $\mathbb{P}(\text{'Disc Var.'}|\text{'Cont Var.'})$ under synthetic data set-

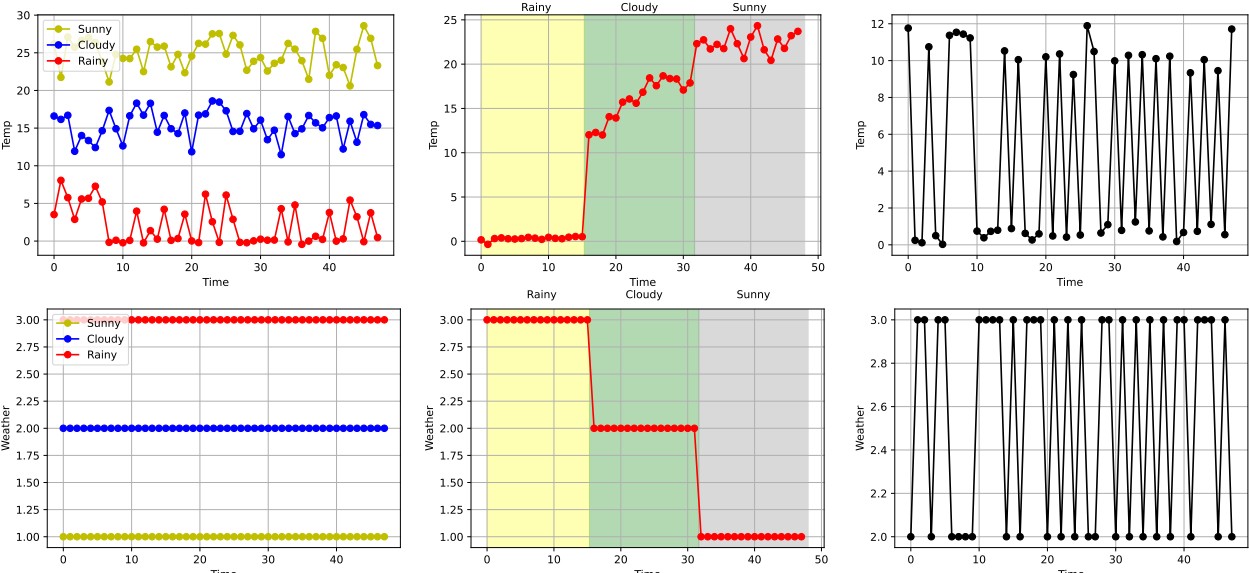

Figure 11: Empirical validations of the model's ability to learn the conditional probability distributions $\mathbb{P}\big(\text{'Temperature'}|\text{'Weather'}\big)$ (top 3 panels) and $\mathbb{P}\big(\text{'Weather'}|\text{'Temperature'}\big)$ (bottom 3 panels).

ting. Additionally, we explore its application in counterfactual scenario analysis with real-world Traffic data, investigating how weather sequences affect traffic volume.

**Synthetic Setting:** Real-world data often involves complex correlations and confounding factors, making it difficult to establish strict causal relationships. To validate that `TimeAutoDiff` can effectively learn conditional rules, we use a synthetic dataset with variables 'Temperature' and 'Weather'. The 'Temperature' is generated over 10,000 time points as:

$$\text{Temp}(t) = 15 + 10\sin\left(\frac{2\pi t}{365}\right) + \mathcal{N}(0, 2^2),$$

where Temp(t) follows a sinusoidal pattern with added Gaussian noise. Based on the generated 'Temperature', the categorical 'Weather' variable is derived as follows: 'Sunny' if Temp > 20, 'Cloudy' if $10 < \text{Temp} \le 20$, and 'Rainy' if $0 < \text{Temp} \le 10$. We set the time window as $T = 48$ (hours) and train the model to learn two conditional distributions: $\mathbb{P}(\text{Temp}|\text{Weather})$, which predicts temperature given weather, and $\mathbb{P}(\text{Weather}|\text{Temp})$, which predicts weather given temperature.

Fig 11 (top 3) demonstrates the model's ability to generate 'Temperature' sequences corresponding to specific weather conditions under three scenarios: (1) sequences with constant weather conditions across three consecutive 48-step windows—'Sunny', 'Cloudy', and 'Rainy', labeled as 1, 2, and 3 respectively; (2) a repeating pattern of weather labels (e.g., 16 'Rainy', 16 'Cloudy', 16 'Sunny'), and (3) random alternating patterns of 'Cloudy' and 'Rainy'. The results show distinct separations in the temperature sequences generated for each weather condition, validating the model's ability to learn $\mathbb{P}(\text{Cont Var.}|\text{Disc Var.})$. Similarly, Fig 11 (bottom 3) demonstrates the reverse case. When conditioned on 'Temperature' values generated in the previous scenarios, the model correctly predicts the corresponding 'Weather' labels at each time step (which is surprising), further validating its ability to learn $\mathbb{P}(\text{Disc Var.}|\text{Cont Var.})$.

**Traffic Data:** To evaluate `TimeAutoDiff` on real-world data, we use the Traffic dataset with 'Traffic Volume' (continuous) as the output and 'Weather-main' (categorical) as the conditional variable. 'Weather-main' includes labels such as {'Clear', 'Rain', 'Squall', 'Cloudy'}, among others. Intuitively, we expect lower traffic volumes during adverse weather conditions (e.g., 'Squall', 'Rain') and higher traffic volumes during good weather (e.g., 'Clear', 'Cloudy'). We test the model under three weather scenarios: 'Cloudy', 'Squall', and 'Clear', using six different timestamp sequences to observe patterns. As shown in the results, 'Traffic Volume' is consistently lower during 'Squall' compared to 'Cloudy' and 'Clear', while no significant differences are

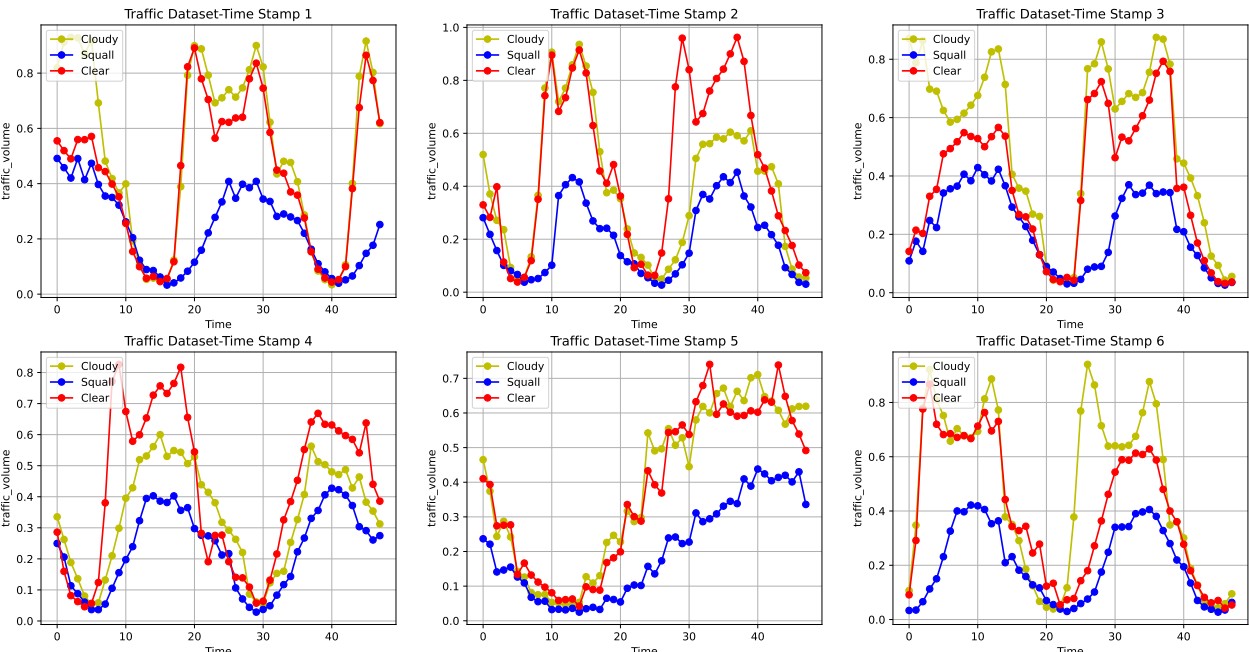

Figure 12: We choose arbitrary 6 timestamp sequences in dataset, and give the models labels of ['Cloudy', 'Squall', 'Clear'] weather-conditions. The traffic-volume axis is normalized.

observed between 'Clear' and 'Cloudy'. These findings confirm the model's ability to reflect expected traffic patterns under different weather conditions.

### 5.3.3 Impact of Metadata Corruption on Generative Robustness.

In real-world time-series generation tasks, metadata—such as contextual covariates, environmental indicators, or categorical conditions—often contain noise or missing values due to sensor drift, transmission errors, or imperfect data logging.

To this end, we introduce a *metadata corruption setting* that perturbs both continuous and categorical metadata according to a corruption level $\alpha \in \{0, 0.1, 0.2, 0.4, 0.6\}$. Continuous metadata are corrupted by additive Gaussian noise scaled to each feature's empirical variability:

$$\tilde{z} = z + \alpha \sigma_z \epsilon, \qquad \epsilon \sim \mathcal{N}(0, I),$$

where $\sigma_z$ denotes the per-feature standard deviation. Categorical metadata are corrupted by randomly flipping each category with probability $\alpha$, replacing it with a uniformly sampled alternative:

$$\tilde{c} = \begin{cases} c, & \text{with probability } 1 - \alpha, \\ \text{Unif}(\mathcal{C} \setminus \{c\}), & \text{with probability } \alpha. \end{cases}$$

This corruption mechanism provides a controlled degradation of contextual information, enabling evaluation of the model's robustness under increasingly unreliable metadata.

Under this noisy setting, we evaluate two fidelity-based metrics—*Discriminative Score* and *Temporal Discriminative Score*—which quantify how closely the generated sequences match the real data in terms of distributional similarity and temporal coherence, respectively. Lower values indicate higher fidelity. As shown in Fig. 13, all datasets (*Traffic*, *Pollution Data*, *Hurricane*, *AirQuality*, *ETTh1*, *Energy*) exhibit increasing scores as the corruption level grows, demonstrating that metadata noise degrades both distributional and temporal realism. (Recall that values approaching 0.5 correspond to real and synthetic data becoming fully distinguishable.) We adopt the same experimental protocol as in subsection 5.3.1.

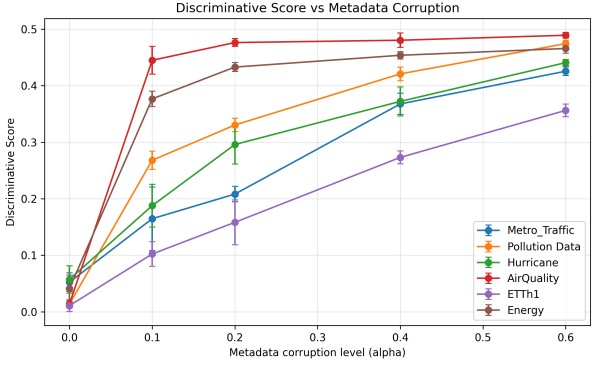
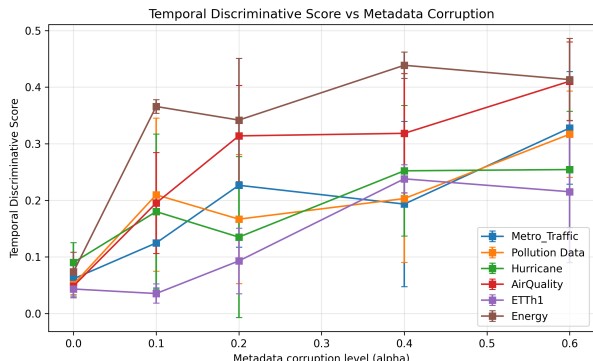

**(a)** Discr. Score vs. corruption level $\alpha$.        **(b)** Temporal Disc. Score vs. corruption level $\alpha$.

Figure 13: **Fidelity under metadata corruption.** Both metrics (lower = higher fidelity) degrade as metadata corruption $\alpha$ increases. For each $\alpha$, scores are computed over five independent runs and then averaged.

This overall trend matches our intuition. Interestingly, the *Discriminative Score* reveals two characteristic behaviors across datasets: some datasets (*Traffic*, *Pollution Data*, *Hurricane*, *ETTh1*) display a near-linear degradation with corruption, while others (*AirQuality*, *Energy*) show a sharp increase at small corruption levels (e.g., $\alpha = 0.1$) followed by a plateau. This suggests that the degree to which metadata shapes marginal feature distributions varies by dataset. In contrast, the *Temporal Score* exhibits less structured trends—no clear linear increase—yet the overall upward shift remains consistent across datasets. This indicates that temporal coherence is also negatively impacted, but in a manner that depends more intricately on each dataset's underlying dynamics than on a simple corruption–fidelity proportionality.

# 6 Discussions on future topics with relevant literature

In this subsection, we further discuss about the four possible extensions of `TimeAutoDiff` in sequel: (1) Privacy guarantees; (2) Interpretability of generated time series data; (3) Extension to foundational model; (4) Bias from conditional metadata generation.

**(1) Privacy Guarantees** is one of the main motivations of synthetic data. Specifically, in the time series domain, data from the healthcare and financial sectors is ubiquitous, but it often comes with significant privacy concerns. We hope the synthetic data does not leak any private information of the original data, while preserving good fidelities. `TimeAutoDiff` lays the foundation for guaranteeing such privacy concerns with the generated synthetic data. In the vision domain, differential privacy guarantees (Dwork, 2006) of synthetic images from diffusion-based models have been investigated by several researchers (Dockhorn et al., 2022; Ghalebikesabi et al., 2023; Lyu et al., 2023). Specifically, Lyu et al. (2023) studied DP-guarantees of latent diffusion model by fine-tuning the attention module of noise predictor in their diffusion model, and claim their synthetic images both have good fidelities and DP-guarantees.

Nonetheless, it is still not clear how the same idea can be applied to time series synthetic data (or regular tabular data), as differentially private time series data is frequently challenging to interpret (Yoon et al., 2020). In this regard, another privacy criterion, $\varepsilon$-identifiability (Yoon et al., 2020) (with $\varepsilon \in [0, 1]$) can be considered as another alternative. The distance between synthetic and original data is measured through Euclidean distance, and we want at least $(1 - \varepsilon)$-proportion of the synthetic data to be distinguishable (or different enough) from the original data. Under this criterion, we conjecture `TimeAutoDiff` can be extended to the synthesizer with a (theoretically-provable) privacy guarantees. The idea can be underpinned around several recent results on diffusion model (Zhang et al., 2023c; Bodin et al., 2024). Zhang et al. (2023c) showed that there exist closed-form solutions of noise predictors for every diffusion step of noisy training data points. This means that we can trace back the latent vectors (or matrix) where the original training data points are generated from. Recent findings (Bodin et al., 2024) suggest that a proper linear

combination of data in the latent space can produce a new semantically meaningful dataset in the original space. Combining the fact that the mapping from the latent space to the original space is Lipschitz continuous (Zhang et al., 2023c) through deterministic sampling (probability-flow), we might be able to have controls over the generations of time series synthetic data, whose Euclidean distances from training data points are away from the training data points. This idea is naturally related to the diversity of generated data as well.

**(2) Interpretability** of the generated time series data is another crucial aspect that time series synthesizer should possess. In many practical applications, for instance, in financial sector, stakeholders and domain experts may be hesitant to rely on synthesis models that are difficult to interpret, as they need to understand and trust the model's behavior, especially when dealing with critical or high-risk scenarios. The current version of `TimeAutoDiff` does not have the luxury of generating interpretable results, but this can be easily adopted by following the previous works. Specifically, we want to point out readers `TimeVAE` (Desai et al., 2021) and `Diffusion-TS` (Yuan & Qiao, 2023), which both focus on building a synthesizer with interpretability. Specifically, `TimeVAE` adopted a sophisticatedly designed decoder in VAE, which has *trend*, *seasonality*, and *residual* blocks for signal decompositions. Similarly, `Diffusion-TS` also design a sophisticated decoder for the decomposition of signals into trend, seasonality, and residual, where they employ the latent diffusion framework. Both of these ideas can be directly employed in `TimeAutoDiff`, where the current decoder is set as an MLP block for simplicity.

**(3) Extension to foundational model** is another promising route the `TimeAutoDiff` can take. Recently, we have been seeing a wave of foundational models research on time series domain (Cao et al., 2024; Liu et al., 2024; Das et al., 2023; Yang et al., 2024a). These models can accommodate multiple tables from cross domains, enabling multiple time series tasks in one model; for instance, forecasting, anomaly detection, imputation, and synthetic data generation (See Cao et al. (2024).) Among them, Cao et al. (2024) devised cleverly designed masks, which provide the unifying framework to do the four abovementioned tasks under diffusion based framework. Nonetheless, their methods are confined to the continuous data modality, and not clear how the model can be extended to heterogeneous features, leaving the great future opportunities for `TimeAutoDiff` to be extended. We also conjecture the synthetic data from `TimeAutoDiff` can be beneficial to improving quality of forecasting foundation model i.e., see Section 5 in (Das et al., 2023).

**(4) Bias from conditional metadata generation:** Generated data can indeed be biased with respect to conditional metadata, arising from various factors. Bias in the training data, such as inherent associations between metadata and outputs, may lead the model to replicate these biases, for instance, generating disproportionately high traffic volumes for "Clear" weather even when the true relationship is less deterministic. Imbalanced metadata distributions further exacerbate this issue, as underrepresented conditions in the training set often result in less reliable outputs for those conditions, such as biased outcomes for minority demographic groups in healthcare datasets. Simplified assumptions in the model, such as assuming linear relationships between metadata and outputs, can overlook complex dependencies, producing data that fails to reflect the true conditional distribution. Noise injection, a feature of models like diffusion models and VAEs, can introduce additional bias if the noise interacts with metadata in unexpected ways, particularly for rare metadata values. Furthermore, limitations in conditional architectures, such as inadequate metadata encoding, can prevent the model from capturing nuanced dependencies, leading to misaligned outputs. To mitigate such biases, ensuring balanced training data, employing robust metadata encoding techniques, applying regularization or fairness constraints, performing post-generation bias audits, and designing disentangled latent spaces are crucial steps. While conditional generative models aim to align generated data with metadata, addressing these biases is essential to ensure fairness and reliability.

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

# A  Derivation of ELBO loss for $-\log P_\Theta\big(\mathbf{X}^{\text{tar}} \mid \mathbf{X}^{\text{con}}, \mathbf{M}^{\text{task}}\big)$

We assume the conditional density $P_\Theta\big(\mathbf{X}^{\text{tar}}, \mathbf{Z}_0^{\text{Lat}} \mid \mathbf{X}^{\text{con}}, \mathbf{M}^{\text{task}}\big) := P_\psi\big(\mathbf{X}^{\text{tar}} \mid \mathbf{X}^{\text{con}}, \mathbf{Z}_0^{\text{Lat}}, \mathbf{M}^{\text{task}}\big) P_\theta\big(\mathbf{Z}_0^{\text{Lat}} \mid \mathbf{X}^{\text{con}}\big)$. Then, negative loglikelihood (i.e., $-\log P_\Theta\big(\mathbf{X}^{\text{tar}} \mid \mathbf{X}^{\text{con}}, \mathbf{M}^{\text{task}}\big)$) can be bounded by:

$$
\begin{aligned}
&-\log P_\Theta\big(\mathbf{X}^{\text{tar}} \mid \mathbf{X}^{\text{con}}, \mathbf{M}^{\text{task}}\big) \\
&\quad = -\log \int P_\psi\big(\mathbf{X}^{\text{tar}} \mid \mathbf{X}^{\text{con}}, \mathbf{Z}_0^{\text{Lat}}, \mathbf{M}^{\text{task}}\big) P_\theta\big(\mathbf{Z}_0^{\text{Lat}} \mid \mathbf{X}^{\text{con}}\big) d\mathbf{Z}_0^{\text{Lat}} \\
&\quad = -\log \int \frac{q_\phi\big(\mathbf{Z}_0^{\text{Lat}} \mid \mathbf{X}^{\text{tar}}, \mathbf{X}^{\text{con}}\big)}{q_\phi\big(\mathbf{Z}_0^{\text{Lat}} \mid \mathbf{X}^{\text{tar}}, \mathbf{X}^{\text{con}}\big)} P_\psi\big(\mathbf{X}^{\text{tar}} \mid \mathbf{X}^{\text{con}}, \mathbf{Z}_0^{\text{Lat}}, \mathbf{M}^{\text{task}}\big) P_\theta\big(\mathbf{Z}_0^{\text{Lat}} \mid \mathbf{X}^{\text{con}}\big) d\mathbf{Z}_0^{\text{Lat}} \\
&\quad \leq -\int q_\phi\big(\mathbf{Z}_0^{\text{Lat}} \mid \mathbf{X}^{\text{tar}}, \mathbf{X}^{\text{con}}\big) \log P_\psi\big(\mathbf{X}^{\text{tar}} \mid \mathbf{X}^{\text{con}}, \mathbf{Z}_0^{\text{Lat}}, \mathbf{M}^{\text{task}}\big) d\mathbf{Z}_0^{\text{Lat}} \\
&\qquad + \int q_\phi\big(\mathbf{Z}_0^{\text{Lat}} \mid \mathbf{X}^{\text{tar}}, \mathbf{X}^{\text{con}}\big) \log \frac{q_\phi\big(\mathbf{Z}_0^{\text{Lat}} \mid \mathbf{X}^{\text{tar}}, \mathbf{X}^{\text{con}}\big)}{P_\theta\big(\mathbf{Z}_0^{\text{Lat}} \mid \mathbf{X}^{\text{con}}\big)} d\mathbf{Z}_0^{\text{Lat}} \\
&\quad = \mathbb{E}_{q_\phi}\big[-\log P_\psi\big(\mathbf{X}^{\text{tar}} \mid \mathbf{X}^{\text{con}}, \mathbf{Z}_0^{\text{Lat}}, \mathbf{M}^{\text{task}}\big)\big] + \mathcal{D}_{\mathbf{KL}}\big(q_\phi\big(\mathbf{Z}_0^{\text{Lat}} \mid \mathbf{X}^{\text{tar}}, \mathbf{X}^{\text{con}}\big) \,\|\, P_\theta\big(\mathbf{Z}_0^{\text{Lat}} \mid \mathbf{X}^{\text{con}}\big)\big) \\
&\quad = \underbrace{\mathbb{E}_{q_\phi}\big[-\log P_\psi\big(\mathbf{X}^{\text{tar}} \mid \mathbf{X}^{\text{con}}, \mathbf{Z}_0^{\text{Lat}}, \mathbf{M}^{\text{task}}\big)\big] + \mathcal{D}_{\mathbf{KL}}\big(q_\phi\big(\mathbf{Z}_0^{\text{Lat}} \mid \mathbf{X}^{\text{tar}}, \mathbf{X}^{\text{con}}\big) \,\|\, \mathcal{N}\big(0, \mathcal{I}_{TF \times TF}\big)\big)}_{:=\mathcal{L}_{\text{VAE}}} \\
&\qquad + \underbrace{\mathbb{E}_{q_\phi}\big[-\log P_\theta\big(\mathbf{Z}_0^{\text{Lat}} \mid \mathbf{X}^{\text{con}}\big)\big]}_{:=\mathcal{L}_{\text{DM}}} + \underbrace{\mathbb{E}_{q_\phi}\big[\log \mathcal{N}\big(0, \mathcal{I}_{TF \times TF}\big)\big]}_{\text{Const.}}.
\end{aligned}
$$

# B  Datasets and Data Processing Steps

We used six single-sequence and two multi-sequence time-series datasets for our experiments. The statistical information of datasets used in our experiments is in Table 6.

**Single-sequence:** For single-sequence datasets, we segment the time series with $N$ total timestamps into overlapping windows of length $T$ and stride $S$, resulting in a tensor of shape $(N - T + S) \times T \times F$, where $F$ denotes the number of features in the table. The specific values of $T$ and $S$ vary across different experimental settings.

| Dataset | # of Rows | #-Cont. | #-Disc. | Seq. Type | Pred Score Col. |
|---------|-----------|---------|---------|-----------|-----------------|
| Traffic | 48205 | 3 | 5 | Single | `traffic volume` |
| Pollution | 43825 | 5 | 3 | Single | `lr` |
| Hurricane | 9937 | 4 | 4 | Single | `seasonal` |
| AirQuality | 9358 | 1 | 12 | Single | `AH` |
| ETTh1 | 17431 | 7 | 0 | Single | `OT` |
| Energy | 19736 | 27 | 1 | Single | `rv2` |
| Bike Sharing | 731 | 6 | 6 | Single | `N.a.N` |
| Card Transaction | 20000 | 2 | 6 | Multi | `Is Fraud?` |
| nasdaq100 | 18231 | 3 | 4 | Multi | `Industry` |

Table 6: Datasets used for our experiments. The date time column is considered as neither continuous nor categorical. The 'Seq.Type' denotes the time series data type: single- or multi-sequence data. The 'Pred Score Col' denotes columns in each dataset used for measuring predictive scores.

- **Traffic** (UCI) is a single-sequence, mixed-type time-series dataset describing the hourly Minneapolis-St Paul, MN traffic volume for Westbound I-94. The dataset includes weather features and holidays for evaluating their impacts on traffic volume. (URL: `https://archive.ics.uci.edu/dataset/492/metro+interstate+traffic+volume`)

- **Pollution** (UCI) is a single-sequence, mixed-type time-series dataset containing the PM2.5 data in Beijing between Jan 1st, 2010 to Dec 31st, 2014. (URL: `https://archive.ics.uci.edu/dataset/381/beijing+pm2+5+data`)

- **Hurricane** (NHC) is a single sequence, mixed-type time-series dataset of the monthly sales revenue (2003-2020) for the tourism industry for all 67 counties of Florida which are prone to annual hurricanes. This dataset is used as a spatio-temporal benchmark dataset for forecasting extreme events and anomalies (Farhangi et al., 2023). (URL: `https://www.nhc.noaa.gov/data/`)

- **AirQuality** (UCI) is a single sequence, mixed-type time-series dataset containing the hourly averaged responses from a gas multisensor device deployed on the field in an Italian city. (URL: `https://archive.ics.uci.edu/dataset/360/air+quality`)

- **ETTh1** (Github: Zhou et al. (2021)) is a single sequence, continuous only time-series dataset, recording hourly level ETT (i.e., Electricity Transformer Temperature), which is a crucial indicator in the electric power long-term deployment. Specifically, the dataset combines short-term and long-term periodical patterns, long-term trends, and many irregular patterns. (URL: `https://github.com/zhouhaoyi/ETDataset/tree/main`)

- **Energy** (Kaggle) is a single sequence time-series dataset. The dataset, spanning 4.5 months, includes 10-minute interval data on house temperature and humidity via a ZigBee sensor network, energy data from m-bus meters, and weather data from Chievres Airport, Belgium, with two random variables added for regression model testing. (URL: `https://www.kaggle.com/code/gaganmaahi224/appliances-energy-time-series-analysis`)

- **Bike Sharing** (UCI) is a single sequence time-series dataset collected from a bike rental service with automated stations. It contains hourly and daily rental records over two years, including information on rental counts, weather conditions, seasonality, and timestamps. Each entry includes normalized temperature, humidity, wind speed, and metadata like holiday/weekend indicators, making it suitable for mobility modeling and demand forecasting tasks. (URL: `http://archive.ics.uci.edu/ml/datasets/Bike+Sharing+Dataset`) *In our experiment, we drop 'instant', 'yr', and 'mnth' columns.*

**Multi-sequence:** The sequences in the multi-sequence data vary in length from one entity to another, so we selected entities with sequences longer than $T = 200$ and $T = 177$ and truncated them to a uniform length of $T$ for the "card transaction" and "nasdaq100" datasets.

- **Card Transaction** is a multi-sequence, synthetic mixed-type time-series dataset created by Padhi et al. (2021a) using a rule-based generator to simulate real-world credit card transactions. We selected 100 users (i.e., entities) for our experiment. In the dataset, we choose {*'Card', 'Amount', 'Use Chip', 'Merchant', 'MCC', 'Errors?', 'Is Fraud?'*} as features for the experiment. (URL: `https://github.com/IBM/TabFormer/tree/main`)

- **nasdaq100** is a multi-sequence, mixed-type time-series dataset consisting of stock prices of 103 corporations (i.e., entities) under nasdaq 100 and the index value of nasdaq 100. This data covers the period from July 26, 2016 to April 28, 2017, in total 191 days. (URL: `https://cseweb.ucsd.edu/~yaq007/NASDAQ100_stock_data.html`)

## C   Denoising Diffusion Probabilistic Model

Ho et al. (2020) proposes the denoising diffusion probabilistic model (DDPM) which gradually adds *fixed* Gaussian noise to the observed data point $\mathbf{x}_0$ via known variance scales $\beta_n \in (0, 1)$, $n \in \{1, \ldots, N\}$ at the diffusion step $n$. This process is referred as *forward process* in the diffusion model, perturbing the data point and defining a sequence of noisy data $\mathbf{x}_1, \mathbf{x}_2, \ldots, \mathbf{x}_N$:

$$q(\mathbf{x}_n \mid \mathbf{x}_{n-1}) = \mathcal{N}(\mathbf{x}_n; \sqrt{1 - \beta_n}\mathbf{x}_{n-1}, \beta_n\mathcal{I}), \quad q(\mathbf{x}_{1:N} \mid \mathbf{x}_0) := \prod_{n=1}^{N} q(\mathbf{x}_n \mid \mathbf{x}_{n-1}).$$

Since the transition kernel is Gaussian, the conditional probability of the $\mathbf{x}_n$ given its original observation $\mathbf{x}_0$ can be succinctly written as:

$$q(\mathbf{x}_n \mid \mathbf{x}_0) = \mathcal{N}(\mathbf{x}_n \mid \sqrt{\bar{\alpha}_n}\mathbf{x}_0, (1 - \bar{\alpha}_n)\mathcal{I}),$$

where $\alpha_n = 1 - \beta_n$ and $\bar{\alpha}_n = \Pi_{k=1}^{n}\alpha_k$. Setting $\beta_n$ to be an increasing sequence, for large enough $N$, leads $\mathbf{x}_N$ to the isotropic Gaussian.

Training objective of DDPM is to maximize the evidence lower bound (in short ELBO) of the log-likelihood $\mathbb{E}_{\mathbf{x}_0}[\log p_\theta(\mathbf{x}_0)]$ as follows;

$$\mathbb{E}_q\left[\log p_\theta(\mathbf{x}_0 \mid \mathbf{x}_1) - \mathcal{D}_{\mathbf{KL}}(q(\mathbf{x}_N \mid \mathbf{x}_0) \,\|\, p(\mathbf{x}_N)) - \sum_{n=1}^{N} \mathcal{D}_{\mathbf{KL}}(q(\mathbf{x}_{n-1} \mid \mathbf{x}_n, \mathbf{x}_0) \,\|\, p_\theta(\mathbf{x}_{n-1} \mid \mathbf{x}_n))\right].$$

The first two terms in the expectation are constants, and the third KL-divergence term needs to be controlled. Interestingly, the conditional probability $q(\mathbf{x}_{n-1} \mid \mathbf{x}_n, \mathbf{x}_0)$ can be driven in the closed-form solution:

$$q(\mathbf{x}_{n-1} \mid \mathbf{x}_n, \mathbf{x}_0) = \mathcal{N}\left(\mathbf{x}_{n-1} \mid \frac{\sqrt{\bar{\alpha}_{n-1}}\beta_n}{1 - \bar{\alpha}_n}\mathbf{x}_0 + \frac{\sqrt{\alpha_n}(1 - \bar{\alpha}_{n-1})}{1 - \bar{\alpha}_n}\mathbf{x}_n, \frac{1 - \bar{\alpha}_{n-1}}{1 - \bar{\alpha}_n}\beta_n\mathcal{I}\right).$$

Noticing the covariance is a constant matrix and KL-divergence between two Gaussians has closed-form solution; DDPM models $p_\theta(\mathbf{x}_{n-1} \mid \mathbf{x}_n) := \mathcal{N}(\mathbf{x}_{n-1} \mid \mu_\theta(\mathbf{x}_n, n), \frac{1 - \bar{\alpha}_{n-1}}{1 - \bar{\alpha}_n}\beta_n\mathcal{I})$. The mean vector $\mu_\theta(\mathbf{x}_n, n)$ is parameterized by a neural network.

The trick used in (Ho et al., 2020) is to reparameterize $\mu_\theta(\mathbf{x}_n, n)$ in terms of $\epsilon_\theta(\mathbf{x}_n, n)$ where it predicts the noise $\epsilon$ added to $\mathbf{x}_n$ from $\mathbf{x}_0$. (Note that $\mathbf{x}_n = \sqrt{\bar{\alpha}_n}\mathbf{x}_0 + \sqrt{1 - \bar{\alpha}_n}\epsilon$ with $\epsilon \sim \mathcal{N}(0, \mathcal{I})$.)

Given this, the final loss function DDPM wants to minimize is:

$$\mathcal{L}_{\text{diff}} := \mathbb{E}_{n,\epsilon}\left[\|\epsilon_\theta(\sqrt{\bar{\alpha}_n}\mathbf{x}_0 + \sqrt{1 - \bar{\alpha}_n}\epsilon, n) - \epsilon\|_2^2\right],$$

where the expectation is taken over $\epsilon \sim \mathcal{N}(0, \mathcal{I})$ and $n \sim \text{Unif}(\{0, \ldots, N\})$.

The generative model learns the *reverse process*. To generate new data from the learned distribution, the first step is to sample a point from the easy-to-sample distribution $\mathbf{x}_N \sim \mathcal{N}(0, \mathcal{I})$ and then iteratively denoise $(\mathbf{x}_N \to \mathbf{x}_{N-1} \to \cdots \to \mathbf{x}_0)$ it using the above model.

# D  Comparison Table of `TimeAutoDiff` with current literature

## D.1  Baseline models for Unconditional Generation

Table 1 compares `TimeAutoDiff` with other time series synthesizers in the literature under seven different aspects. Additionally, we provide further detailed comparisons between our model and `Diffusion-TS` (Yuan & Qiao, 2023) / `TimeDiff` (Tian et al., 2023), and `TabSyn` (Zhang et al., 2023b), `AutoDiff` (Suh et al., 2023), `TabDDPM` (Kotelnikov et al., 2022).

**Diffusion-TS** (Yuan & Qiao, 2023) employs an Autoencoder + DDPM framework for interpretable time-series generation. The primary difference from our work lies in the problem setting. Diffusion-TS assumes the signal consists only of continuous time series, based on the premise that the signal is decomposable into 'trend, seasonality, and noise'. While this decomposition-based approach is effective for continuous data, it is not well-defined for the heterogeneous features (especially discrete variables) that `TimeAutoDiff` addresses, making it difficult to apply directly to our problem.

**TimeDiff** (Tian et al., 2023) also handles heterogeneous features in EHR datasets, but it does so by integrating two distinct diffusion processes: applying DDPM for continuous variables and multinomial diffusion for discrete variables, respectively. This stands in sharp contrast to our architectural philosophy. `TimeAutoDiff` leverages a VAE to first project the entire heterogeneous time series into a single, continuous latent space. We then simplify the modeling by using only a single DDPM framework within this unified latent representation.

Our work is also partially inspired by recent successes in tabular data generation models like **TabSyn** Zhang et al. (2023b), **AutoDiff** Suh et al. (2023), and **TabDDPM** Kotelnikov et al. (2022). However, these models are fundamentally designed for static, 1D tabular data, treating each row as an independent and identically distributed (i.i.d.) data point. This i.i.d. assumption is invalid for time-series data, which is inherently 2D (time and feature axes) and defined by temporal dependencies. That is, these tabular methodologies do not model the sequential relationships between rows (time steps), which is the key distinction `TimeAutoDiff` aims to address.

## D.2  Baseline models for Imputation & Forecasting

**Time-LLM (Jin et al., 2023):** Time-LLM reframes numerical forecasting as a language-modeling problem. Numeric sequences are first segmented into short patches and mapped to a bank of text prototypes, creating a token stream that a frozen LLM can ingest. A natural-language Prompt-as-Prefix (PaP) explains how these synthetic tokens relate to time-series patterns and what forecast is required. The LLM reasons over this hybrid input without any weight updates; its token outputs are then linearly projected back to real values to obtain the prediction. With just the lightweight prototype encoder/decoder trained, Time-LLM matches or surpasses specialist forecasting architectures—and remains effective in zero-shot and few-shot settings.

**MOMENT (Goswami et al., 2024):** MOMENT is a family of large-scale foundation models for time-series analysis, designed to serve as general-purpose models across diverse tasks. Each MOMENT model is a high-capacity Transformer architecture trained on the Time-series-Pile—a massive repository of time-series data collected from many domains (e.g. healthcare, finance, climate). The training objective is a self-supervised masked time-series prediction task, wherein the model learns to reconstruct missing portions of time-series inputs, analogous to masked language modeling but for continuous sequences. Through this pre-training, MOMENT learns rich representations of temporal patterns that can be adapted to various tasks. A single pre-trained MOMENT model can be used out-of-the-box for forecasting, classification, anomaly detection, imputation and more, often in a zero-shot or few-shot manner, while also allowing fine-tuning for further gains. Experiments show that these foundation models perform effectively with minimal task-specific data, and the authors provide a benchmark to evaluate such models' performance across multiple time-series tasks.

**TEFN (Zhan et al., 2024):** The Time Evidence Fusion Network explicitly fuses two complementary evidential views of a multivariate series. First, a Basic Probability Assignment (BPA) module—grounded in Dempster–Shafer theory—assesses the contribution and uncertainty of patterns within each channel (spatial view) and across time (temporal view). A dedicated fusion layer then combines these sources of evidence into a unified latent representation, enabling the model to exploit cross-variable interactions and temporal

dynamics coherently. This design delivers state-of-the-art long-horizon accuracy while remaining lightweight and remarkably robust to hyper-parameter choices.

**TimeMixer (Wang et al., 2024b):** TimeMixer views forecasting through a multi-scale lens. Its Past-Decomposable-Mixing (PDM) block decomposes the history into frequency bands (e.g. seasonal vs. trend) and mixes them bidirectionally—from fine→coarse and coarse→fine scales—so that microscopic and macroscopic information reinforce each other. The Future-Multipredictor-Mixing (FMM) block then ensembles several scale-specialized heads, blending their predictions into the final forecast. Built entirely from MLP layers, TimeMixer attains SOTA accuracy on both short- and long-term tasks while keeping computational complexity linear in sequence length.

**TimesNet (Wu et al., 2022):** TimesNet converts a 1-D sequence into a 2-D tensor to isolate periodic structure. Dominant periods are detected, and the series is reshaped so that rows index successive periods and columns index positions within each period—turning intra-period variations into column patterns and inter-period variations into row patterns. A parameter-efficient TimesBlock (inception-style 2-D convolution) then captures these variations jointly. This single backbone achieves competitive or superior results in forecasting, imputation, classification, and anomaly detection, illustrating the power of modeling temporal data in the 2-D variation domain.

**MICN (Wang et al., 2023a):** The Multi-scale Isometric Convolution Network (i.e., MICN) combines the locality of CNNs with global context modeling at linear cost. Parallel branches process progressively down-sampled versions of the input, and each branch applies (i) standard convolutions for short-range patterns and (ii) isometric dilated convolutions whose receptive field spans the entire (down-sampled) sequence. Outputs from all branches are concatenated to form the forecast, yielding substantial improvements—17% (multivariate) and 22% (univariate) error reductions—over previous best models.

**DLinear (Zeng et al., 2023):** Decomposition-Linear replaces deep networks with a two-step linear pipeline. First, a simple moving-average filter separates each series into trend and seasonal components. Second, two independent one-layer linear projections extrapolate these parts and their outputs are summed to obtain the forecast. Despite its minimal complexity, this seasonal–trend decomposition plus linear extrapolation matches or surpasses state-of-the-art Transformer models on long-horizon benchmarks, showing that carefully chosen inductive bias can rival far more elaborate architectures.

**FiLM (Zhou et al., 2022):** The Frequency-Improved Legendre Memory module augments Legendre polynomial projections with signal-processing tricks to preserve long-range history. A Legendre basis encodes the entire past into a compact state; a learnable Fourier projector filters out high-frequency noise; and a low-rank approximation accelerates computation. FiLM can operate as a standalone predictor or as a plug-in memory block, boosting long-horizon accuracy by 20% when inserted into existing architectures.

**CSDI (Tashiro et al., 2021):** Conditional Score-based Diffusion Imputation frames gap-filling as conditional generation. A diffusion process iteratively perturbs the full sequence, while a neural score model—conditioned on the observed entries—guides reverse denoising to sample plausible completions. This yields a distribution over imputations, enabling uncertainty quantification; deterministic estimates are obtained by averaging or mode-seeking. Across healthcare and environmental datasets, CSDI reduces probabilistic-imputation errors by 40–65% relative to previous methods, and >10% even in deterministic settings.

**PatchTST (Nie et al., 2022):** PatchTST adapts Transformers for forecasting by treating non-overlapping time-series segments (patches) as analogous to NLP tokens. Input is segmented, flattened, and linearly embedded, with learnable positional embeddings added. This patching significantly reduces the input sequence length for a standard Transformer encoder, allowing attention to efficiently model dependencies between patches rather than individual points. This simple mechanism allows a "vanilla" Transformer to achieve state-of-the-art long-term forecasting performance by capturing local semantic information within each patch.

**TimeMixer++ (Wang et al., 2024a):** TimeMixer++ is a lightweight, attention-free MLP architecture based on 'Decomposable Multiscale Mixing'. It explicitly factorizes the time-series into components, using specialized MLPs to model 'Past' (recent variations), 'Seasonal' (periodicity), and 'Trend' (long-term tendencies) patterns separately. These representations are efficiently processed and mixed using shared MLPs,

achieving linear complexity. This interpretable decomposition matches or exceeds Transformer performance while being significantly more computationally efficient.

**iTransformer (Liu et al., 2023):** iTransformer inverts the standard Transformer's application for multivariate forecasting. Instead of applying attention across time, it treats each entire variate (channel) as a single token by embedding its whole time series. A standard Transformer encoder then applies self-attention across these variate-tokens, explicitly modeling the complex interdependencies between channels (e.g., how "temperature" relates to "humidity"). The resulting variate representations are then projected by simple linear layers to generate the forecast, setting a new, simple state-of-the-art baseline.

### D.3 Comparison with Time-Series Foundation Models (TSFMs)

Our work, `TimeAutoDiff`, is a generative framework for heterogeneous time series, and its design principles differ significantly from the emerging paradigm of Time-Series Foundation Models (TSFMs) such as `MOMENT` (Goswami et al., 2024), `Chronos` (Ansari et al., 2024), and `TimesFM` (Das et al., 2023). These large-scale models focus on pre-training, often using tokenization strategies on massive, mostly continuous datasets to learn powerful, general-purpose representations for zero-shot or few-shot forecasting.

The primary distinction lies in data and task specialization. `TimeAutoDiff` is fundamentally designed for the native handling of heterogeneous features. We employ a VAE to project a mix of continuous, discrete, and static data into a single continuous latent space. Our latent-diffusion process then models this unified representation directly. In contrast, many TSFMs are primarily demonstrated on (and architected for) large, homogenous datasets of continuous values, with the handling of mixed data types not being their central focus. Furthermore, our mask-based task unification represents a different objective. While TSFMs aim to create a single, powerful representation for downstream adaptation (primarily forecasting), `TimeAutoDiff` acts as a unified generative model for a diverse set of tasks—including imputation, forecasting, and synthetic data generation—all controlled via a flexible masking mechanism. These approaches may be viewed as complementary rather than competing. TSFMs excel at learning broad temporal patterns from large-scale data, whereas `TimeAutoDiff` provides a high-fidelity, specialized generative framework for complex, heterogeneous datasets where tasks beyond forecasting are critical.

## E   Evaluation Metric

For the quantitative evaluation of synthesized data, we mainly focus on three criteria (1) the distributional similarities of the two tables; (2) the usefulness for predictive purposes; (3) the temporal and feature dependencies; We employ the following evaluation metrics:

***Discriminative Score*** (Yoon et al., 2019) measures the fidelity of synthetic time series data to original data, by training a classification model (optimizing a 2-layer LSTM) to distinguish between sequences from the original and generated datasets.

***Predictive Score*** (Yoon et al., 2019) measures the utility of generated sequences by training a posthoc sequence prediction model (optimizing a 2-layer LSTM) to predict next-step temporal vectors under a *Train-on-Synthetic-Test-on-Real* (TSTR) framework.

***Temporal Discriminative Score*** measures the similarity of distributions of *inter-row differences* between generated and original sequential data. This metric is designed to see if the generated data preserves the temporal dependencies of the original data. For any fixed integer $t \in \{1, \ldots, T-1\}$, the difference of the $n$-th row and $(n+t)$-th row in the table over $n \in \{1, \ldots, T-t\}$ is computed for both generated and original data and discriminative score (Yoon et al., 2019) is computed over the differenced matrices from original and synthetic data. We average discriminative scores over 10 randomly selected $t \in \{1, \ldots, T-1\}$.

***Feature Correlation Score*** measures the averaged $L^2$-distance of correlation matrices computed on real and synthetic data. Following (Kotelnikov et al., 2022), to compute the correlation matrices, we use the Pearson correlation coefficient for numerical-numerical feature relationships, Theil's U statistics between categorical-categorical features, and the correlation ratio for categorical-numerical features. We use the following metrics to calculate the feature correlation score:

- **Pearson Correlation Coefficient**: Used for **Numerical** to **Numerical** feature relationship. Pearson's Correlation Coefficient $r$ is given by

$$r = \frac{\sum(x - \bar{x})(y - \bar{y})}{\sqrt{\sum(x - \bar{x})^2}\sqrt{\sum(y - \bar{y})^2}}$$

  where

  - $x$ and $y$ are samples in features $X$ and $Y$, respectively
  - $\bar{x}$ and $\bar{y}$ are the sample means in features $X$ and $Y$, respectively

- **Theil's U Coefficient**: Used for **Categorical** to **Categorical** feature relationship. Theil's U Coefficient $U$ is given by

$$U = \frac{H(X) - H(X|Y)}{H(X)}$$

  where

  - entropy of feature $X$ is defined as

$$H(X) = -\sum_x P_X(x) \log P_X(x)$$

  - entropy of feature $X$ conditioned on feature $Y$ is defined as

$$H(X|Y) = -\sum_{x,y} P_{X,Y}(x,y) \log \frac{P_{X,Y}(x,y)}{P_Y(y)}$$

  - $P_X$ and $P_Y$ are empirical PMF of $X$ and $Y$, respectively
  - $P_{X,Y}$ is the joint distribution of $X$ and $Y$

- **Correlation Ratio**: Used for **Categorical** to **Numerical** feature relationship. The correlation ratio $\eta$ is given by

$$\eta = \sqrt{\frac{\sum_x n_x(\bar{y}_x - \bar{y})^2}{\sum_{x,i}(y_{xi} - \bar{y})^2}}$$

  where

  - $n_x$ is the number of observations of label $x$ in the categorical feature
  - $y_{xi}$ is the $i$-th observation of the numerical feature with label $x$
  - $\bar{y}_x$ is the mean of observed samples $y_i \in Y$ with label $x$
  - $\bar{y}$ is the sample mean of $Y$

## F  Model parameter settings, Training & Hyper-parameter choices

Our model consists of two components: **VAE** and **DDPM**. We present the sizes of networks in both components that are applied entirely across the experiments in the paper.

VAE-Encoder $= \big\{$Dimension of first FC-layer in MLP-block for encoded features:

(Num of disc var.$\times$**128**+Num of cont var.$\times$**16**) $\times$ **128**,

Dimension of second FC-layer in MLP-block for encoded features:**128** $\times$ **F**,

Dimension of hidden layer for the 2-RNNs for $\mu$ and $\sigma$: **200**,

Number of layers for the 2-RNNs for $\mu$ and $\sigma$: **2**,

Dimension of fully-connected layer topped on 2-RNNs: **200** $\times$ **F** $\big\}$

| Method | Disc. Score | Pred. Score | Temp. Disc Score | Feat. Correl. |
|---|---|---|---|---|
| **TimeAutoDiff** | 0.015(0.012) | 0.229(0.010) | 0.034(0.020) | 0.043(0.000) |
| Latent Feature Dimension = F/2 | 0.009(0.004) | 0.227(0.009) | 0.096(0.061) | 0.055(0.000) |
| Latent Feature Dimension = F/4 | 0.038(0.021) | 0.233(0.007) | 0.099(0.171) | 0.048(0.000) |
| Diffusion Steps = 75 | 0.016(0.009) | 0.224(0.015) | 0.014(0.009) | 0.039(0.000) |
| Diffusion Steps = 50 | 0.118(0.019) | 0.241(0.003) | 0.092(0.046) | 0.109(0.000) |
| Diffusion Steps = 25 | 0.150(0.027) | 0.248(0.006) | 0.111(0.065) | 0.100(0.000) |
| VAE Training = 15000 | 0.075(0.009) | 0.243(0.005) | 0.035(0.007) | 0.091(0.000) |
| VAE Training = 10000 | 0.068(0.018) | 0.242(0.007) | 0.038(0.038) | 0.050(0.000) |
| VAE Training = 5000 | 0.195(0.025) | 0.245(0.002) | 0.039(0.019) | 0.077(0.000) |
| DDPM Training = 15000 | 0.098(0.014) | 0.237(0.015) | 0.062(0.038) | 0.086(0.000) |
| DDPM Training = 10000 | 0.220(0.025) | 0.246(0.004) | 0.165(0.045) | 0.195(0.000) |
| DDPM Training = 5000 | 0.267(0.021) | 0.255(0.001) | 0.216(0.031) | 0.190(0.000) |
| Hidden Dimension of RNNs (VAE) = 150 | 0.013(0.008) | 0.240(0.007) | 0.031(0.009) | 0.015(0.000) |
| Hidden Dimension of RNNs (VAE) = 100 | 0.030(0.009) | 0.236(0.017) | 0.017(0.011) | 0.039(0.000) |
| Hidden Dimension of RNNs (VAE) = 50 | 0.082(0.023) | 0.238(0.004) | 0.051(0.038) | 0.064(0.000) |
| Hidden Dimension of Bi-RNNs (DDPM) = 150 | 0.031(0.010) | 0.243(0.011) | 0.028(0.013) | 0.035(0.000) |
| Hidden Dimension of Bi-RNNs (DDPM) = 100 | 0.167(0.012) | 0.248(0.003) | 0.094(0.054) | 0.119(0.000) |
| Hidden Dimension of Bi-RNNs (DDPM) = 50 | 0.174(0.014) | 0.251(0.005) | 0.157(0.072) | 0.132(0.000) |
| Number of layers in RNNs (VAE) = 1 | 0.024(0.013) | 0.245(0.009) | 0.042(0.018) | 0.028(0.000) |
| Number of layers in Bi-RNNs (DDPM) = 1 | 0.097(0.009) | 0.250(0.002) | 0.245(0.009) | 0.086(0.000) |
| Quadratic Noise Scheduler | 0.109(0.017) | 0.234(0.013) | 0.072(0.025) | 0.106(0.000) |

Table 7: Performances measured with various choices of hyper-parameters in `TimeAutoDiff`. The experiments are conducted on Traffic dataset with $T = 24$.

$$\text{VAE-Decoder} = \big\{ \text{Dimension of first FC-layer in MLP-block for latent matrix } \mathbf{Z}_0\text{: } \mathbf{F} \times \mathbf{128},$$
$$\text{Dimension of second FC-layer in MLP-block for latent matrix } \mathbf{Z}_0\text{: } \mathbf{128} \times \mathbf{128} \big\}$$

$$\text{DDPM} = \big\{ \text{Output dimensions of encodings of } (\mathbf{Z}_n^{\text{Lat}}, n, \mathbf{t}, \mathbf{ts})\text{: } \mathbf{200},$$
$$\text{Dimension of hidden layer for the Bi-RNNs: } \mathbf{200},$$
$$\text{Number of layers for the Bi-RNNs: } \mathbf{2},$$
$$\text{Dimension of FC-layer of the output of Bi-RNNs: } \mathbf{400} \times \mathbf{F},$$
$$\text{Diffusion Steps: } \mathbf{100} \big\}$$

Training for both the VAE and DDPM models is set to 25,000 epochs. The batch size for VAE training is 100, while the batch size for DDPM training matches the number of diffusion steps. We use the Adam optimizer, with a learning rate of $2 \times 10^{-4}$ decaying to $10^{-6}$ for the VAE, and a learning rate of $10^{-3}$ for the DDPM. For stabilization of diffusion training, we employ Exponential Moving Average (EMA) with decay rate 0.995. We employ linear noise scheduling for $\beta_n := 1 - \alpha_n$, $n \in \{1, 2, \ldots, N\}$ with $\beta_1 = 10^{-4}$ and $\beta_N = 0.2$:

$$\beta_n = \left(1 - \frac{n}{N}\right)\beta_1 + \frac{n}{N}\beta_N.$$

In the following, we investigate the robustness of our models to the various hyper-parameter choices in VAE and DDPM. Specifically, we studied the effects of (1) feature dimension of $Z_0^{\text{Lat}}$ ($F/2, F/4$), (2) number of diffusion steps $(75, 50, 25)$, (3) training epochs of VAE and DDPM $(20000, 15000, 10000)$, (4) dimension of hidden layers of two RNNs (for $\mu$ and $\sigma$) in VAE $(150, 100, 50)$, (5) dimension of hidden layers of Bi-RNNs in DDPM $(150, 100, 50)$, (6) the number of layers of two RNNs (for $\mu$ and $\sigma$) in VAE $(1)$, (7) the number of layers of Bi-RNNs in DDPM $(1)$. (8) the quadratic noise scheduler used in Song et al. (2020a); Tashiro et al. (2021):

$$\beta_n = \left(\left(1 - \frac{n}{N}\right)\sqrt{\beta_1} + \frac{n}{N}\sqrt{\beta_N}\right)^2.$$

with the minimum noise level $\beta_1 = 0.0001$, and the maximum noise level $\beta_N = 0.5$.

The experiments are conducted over the varying parameters (in the paranthesis), while the remaining parameters in the model are being fixed as in the above settings. The first 2000 rows of **Traffic** data are used for the experiments with sequence length 24. (i.e., the dimensions of tensors used in the experiments are [B, T, F ] = $[1977, 24, 8]$)

**Results Interpretations:** Table 7 presents the performance of the models across four metrics, with variations in hyperparameter settings. Overall, larger models yield better results. Reducing the diffusion steps, dimensions, and the number of hidden layers in RNNs within the VAE and Bi-RNN components of DDPM significantly degrades model performance. Longer training of both VAE and DDPM consistently enhances results. The linear noise scheduler outperforms the quadratic noise scheduler. While reducing the feature dimension to $F/2$ slightly improves discriminative and temporal discriminative scores, further compression to $F/4$ leads to information loss during signal reconstruction, resulting in poorer performance.

## G   Computational Complexity of `TimeAutoDiff`

Let $T$ denote the sequence length, $F$ the number of original features, $L$ the latent feature dimension after VAE encoding, $H$ the hidden size of the encoder RNN, $h$ the hidden size of the decoder MLP, $H_{\text{diff}}$ the hidden size of the diffusion denoiser, and $N$ the number of diffusion steps. All Big-$O$ expressions are computed per sequence $(\mathbf{x}_1, \ldots, \mathbf{x}_T)$ and scale linearly with batch size.

**1. Encoder.** The encoder consists of an embedding layer, an MLP, two RNNs (for $\mu$ and $\log \sigma^2$), and a projection layer. A single RNN forward pass costs $O(T(H^2 + HF))$, while the projection adds $O(THL)$, which is typically negligible. Thus,

$$\text{Cost}_{\text{enc}} = O(T(H^2 + HF)).$$

The overall encoder cost grows linearly with $T$ and $F$, dominated by the recurrent term $O(TH^2)$.

**2. Decoder.** The decoder reconstructs heterogeneous outputs from the latent sequence $Z_0^{\text{Lat}} \in \mathbb{R}^{T \times L}$ through a two-layer MLP followed by separate output heads (binary, categorical, and numerical):

$$\text{Cost}_{\text{dec}} = O(T(Lh + h^2 + hF)).$$

This expression accounts for latent-to-hidden mapping, internal MLP propagation, and multi-head output projections.

**3. Diffusion Model.** The diffusion denoiser operates in the latent space with input dimension $L$. Each step involves RNN or MLP blocks with cost

$$O(T(H_{\text{diff}}^2 + H_{\text{diff}}L)).$$

During training, only one random timestep is used per iteration:

$$\text{Cost}_{\text{diff,train}} = O(T(H_{\text{diff}}^2 + H_{\text{diff}}L)).$$

During inference, $N$ iterative denoising steps are performed:

$$\text{Cost}_{\text{diff,infer}} = O(NT(H_{\text{diff}}^2 + H_{\text{diff}}L)).$$

After denoising, the decoder is applied once to recover full-resolution sequences.

**4. Overall Training and Inference.** Combining all components, the total training cost per sequence is

$$\text{Cost}_{\text{train}} = O(T(H^2 + HF)) + O(T(Lh + h^2 + hF)) + O(T(H_{\text{diff}}^2 + H_{\text{diff}}L)).$$

The inference cost includes the $N$-step denoising and one decoder call:

$$\text{Cost}_{\text{infer}} = O(NT(H_{\text{diff}}^2 + H_{\text{diff}}L)) + O(T(Lh + h^2 + hF)).$$

**5. Discussion.** `TimeAutoDiff` achieves efficient generation by: (1) performing diffusion in a compressed latent space ($L \ll F$), and (2) adopting a moderate number of diffusion steps ($N \approx 50$–$100$). These design choices reduce inference time by approximately 10–100× compared to data-space diffusion models ($N_{\mathrm{d}} \geq 1000$), while offering improved fidelity and flexibility relative to single-step GANs.

## H  Volatility and Moving Average: comparison between real and synthetic under stock data

We provide the performance of our model in terms of volatility and moving average. We first provide the brief descriptions on Simple Moving Average, Exponential Moving Average, and Volatility.

**Simple Moving Average (SMA)**: The Simple Moving Average (SMA) is computed as the arithmetic mean of values over a sliding window of size $w = 5$. For a given time step $t$, the SMA is given by:

$$\mathrm{SMA}_t = \frac{1}{5} \sum_{i=t-4}^{t} \mathrm{Value}_i$$

where $\mathrm{Value}_i$ represents the value of the time series at time $i$. This metric smooths short-term fluctuations and highlights the overall trend by averaging values in the specified window.

**Exponential Moving Average (EMA)** The Exponential Moving Average (EMA) is a weighted average of values where recent data points have exponentially greater weight. For a window size of $w = 5$, the smoothing factor $\alpha$ is computed as: $\alpha = \frac{2}{w+1} = \frac{2}{5+1} = \frac{1}{3}$. The EMA at time $t$ is then computed recursively as:

$$\mathrm{EMA}_t = \alpha \cdot \mathrm{Value}_t + (1 - \alpha) \cdot \mathrm{EMA}_{t-1}$$

where $\mathrm{Value}_t$ is the current value of the time series, and $\mathrm{EMA}_{t-1}$ is the EMA from the previous time step. This method emphasizes recent changes while retaining some information from the historical trend.

**Volatility** Volatility measures the degree of variation in the time series over a sliding window of size $w = 5$. It is calculated as the rolling standard deviation of the percentage changes (returns). First, the percentage change (return) between consecutive values is computed as:

$$\mathrm{Return}_i = \frac{\mathrm{Value}_i - \mathrm{Value}_{i-1}}{\mathrm{Value}_{i-1}}$$

For a given time step $t$, the volatility over the window $w = 5$ is given by: $\mathrm{Volatility}_t = \sqrt{\frac{1}{5} \sum_{i=t-4}^{t} \left( \mathrm{Return}_i - \overline{\mathrm{Return}} \right)^2}$ where $\overline{\mathrm{Return}}$ is the mean of the returns within the window.

**Results:** We work on the stock data. The figure 14 provide a clear side-by-side comparison between the synthetic and real data, with the left column displaying the synthetic data and the right column showcasing the corresponding real data for two selected features (Open & Close prices) over 200 timestamps (i.e.,T=200). Each row focuses on one feature, allowing for a detailed examination of the behavior across key metrics: Simple Moving Average (SMA), Exponential Moving Average (EMA), and Volatility. The SMA and EMA curves, plotted alongside the raw time series data, highlight the ability of the synthetic data to replicate the long-term trends (SMA) and short-term responsiveness (EMA) observed in the real data. Volatility, overlaid as a secondary y-axis in each plot, demonstrates the synthetic data's capacity to reproduce the temporal variability, including periods of high and low uncertainty, as reflected in the real data. The remarkable alignment across all metrics suggests that the synthetic data closely mirrors the real data's dynamics, effectively capturing both the overall patterns and nuanced fluctuations. This visual comparison underscores the robustness and reliability of the synthetic data generation process.

## I  Maximum Mean Discrepancy & Entropy

We used two metrics proposed by TSGM (Nikitin et al., 2023): Maximum Mean Discrepancy (MMD) and Entropy. MMD measures the similarity (or fidelity) between synthetic and real time series data, while Entropy

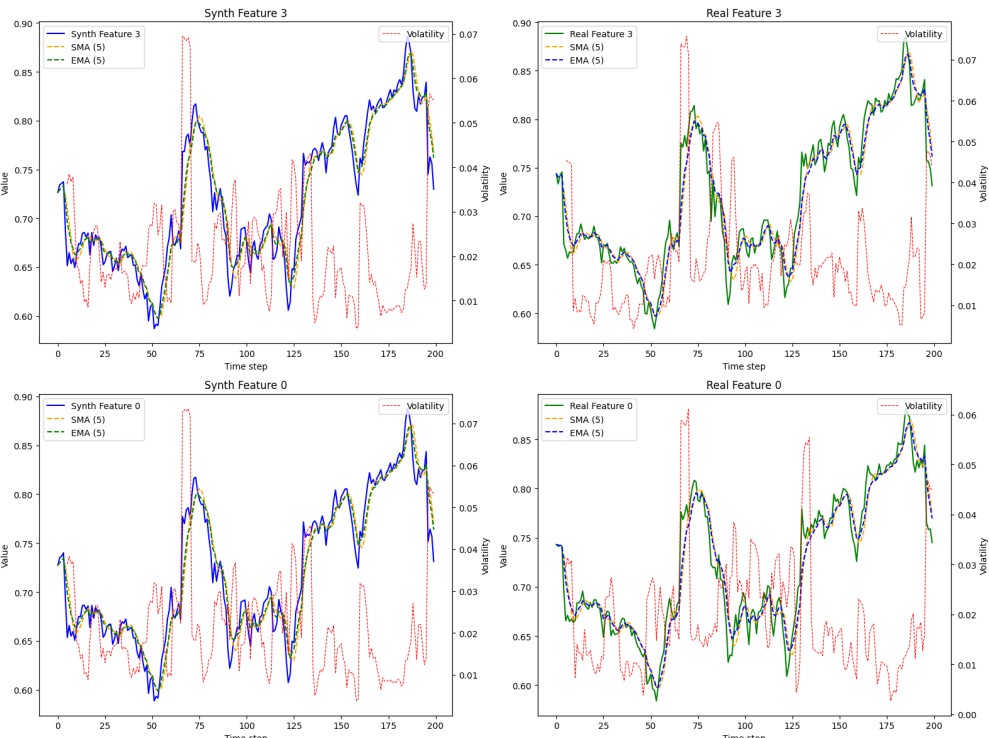

Figure 14: Comparison of synthetic (left) and real (right) data across two features, illustrating alignment in trends (SMA, EMA) and variability (Volatility: secondary y-axis).

assesses the diversity of the synthetic data. The results are summarized in Table 8 and are consistent with those in Table 2. `TimeAutoDiff` achieves the lowest MMD scores across all four datasets, aligning with the discriminative scores reported in Table 2. This indicates that `TimeAutoDiff` effectively generates synthetic data that closely resembles real data. For diversity, higher Entropy values indicate a dataset with more diverse samples. However, as noted in (Nikitin et al., 2023), Entropy should be considered alongside other metrics, as random noise can also result in high Entropy values. `TimeAutoDiff` produces synthetic data with higher Entropy values than the real data, though not as excessively as other baseline models. This suggests that our model generates synthetic data that preserves the statistical properties of the original data, maintaining diversity without introducing excessive deviation.

## J   Applicability to High-Dimensional Data

To demonstrate the effectiveness of feature compression in the latent space, we evaluate our model on high-dimensional synthetic datasets.

***Sine Waves.*** We generate multivariate sinusoidal sequences with varying frequencies $\eta$ and phases $\theta$, providing continuous, periodic, and mutually independent features. For each dimension $i \in \{1, ..., F\}$, the signal is defined as $x_i(t) = \sin(2\pi\eta t + \theta)$, where $\eta \sim \text{Unif}[0, 1]$ and $\theta \sim \text{Unif}[-\pi, \pi]$. The model is trained on data with shape [Batch Size $\times$ Seq Len $\times$ Feature Dim], and samples are drawn with the same configuration. We record the training time (for both VAE and diffusion models), sampling time, GPU memory usage during sampling (in MB), discriminative score, and temporal discriminative score. Under the configurations described in Appendix F, `TimeAutoDiff` successfully generates sequences of length 900 with 5 features, achieving high fidelity (see Table 9). However, when the number of features increases (e.g., from 30 to 50) while keeping the sequence length fixed at 200, performance noticeably degrades if the latent feature dimension remains the same as the original input. To mitigate this, we reduce the latent feature dimension to $F/2$, which significantly improves generation quality in high-dimensional settings.

| Metric | Method | Traffic | Pollution | Hurricane | AirQuality |
|---|---|---|---|---|---|
| MMD Score

(The lower, the better) | `TimeAutoDiff` | 0.000629 | 0.000895 | 0.000891 | 0.001531 |
| | TimeGAN | 0.001738 | 0.009791 | 0.002775 | 0.042986 |
| | DoppelGANer | 0.000644 | 0.000960 | 0.005489 | 0.017038 |
| | Diffusion-TS | 0.005099 | 0.037102 | 0.078387 | 0.004144 |
| | TSGM | 0.001484 | 0.006322 | 0.031971 | 0.013777 |
| | real vs. real | 0.000000 | 0.000000 | 0.000000 | 0.000000 |
| Entropy Score

(Needs to be considered
with other metrics) | `TimeAutoDiff` | 6419.404 | 8472.642 | 7129.152 | 16570.016 |
| | TimeGAN | 6714.156 | 11021.597 | 7804.343 | 15343.967 |
| | DoppelGANer | 3941.083 | 8656.403 | 6946.678 | 8708.616 |
| | Diffusion-TS | 9763.042 | 7372.591 | 9861.151 | 15934.365 |
| | TSGM | 11899.225 | 11854.764 | 6535.306 | 15766.673 |
| | Real | 5983.576 | 6976.253 | 6613.284 | 14952.996 |

Table 8: Maximum Mean Discrepancy (MMD) and Entropy of `TimeAutoDiff`, `TimeGAN`, `DoppelGANer`, `Diffusion-TS`, `TSGM` and Real data.

| Batch Size | Seq Len | VAE | Diff | Sampling | GPU Mem | Disc Scr | Temp Disc Scr |
|---|---|---|---|---|---|---|---|
| 500 | 100 | 187.23 | 94.32 | 1.294 | 910.47 | 0.067 (0.034) | 0.143 (0.114) |
| 400 | 300 | 420.23 | 201.47 | 3.585 | 1991.75 | 0.040 (0.023) | 0.064 (0.059) |
| 300 | 500 | 665.69 | 315.92 | 5.511 | 2572.63 | 0.032 (0.016) | 0.078 (0.078) |
| 200 | 700 | 928.83 | 415.36 | 7.303 | 2466.91 | 0.048 (0.016) | 0.193 (0.122) |
| 100 | 900 | 1209.34 | 530.36 | 8.499 | 1670.75 | 0.16 (0.094) | 0.13 (0.143) |

Table 9: The number of feature is fixed as 5. The sequence length increases up to 900.

| Batch Size | Feat Dim | VAE | Diff | Sampling | GPU Mem | Disc Scr | Temp Disc Scr |
|---|---|---|---|---|---|---|---|
| 800 | 10 | 128.11 | 355.03 | 5.36 | 4696.21 | 0.24 (0.08) | 0.26 (0.09) |
| 800 | 20 | 132.48 | 359.32 | 4.12 | 5080.84 | 0.26 (0.05) | 0.38 (0.08) |
| 800 | 30 | 134.02 | 371.38 | 3.99 | 5540.96 | 0.31 (0.08) | 0.33 (0.17) |
| 800 | 40 | 134.72 | 364.85 | 3.97 | 6003.00 | 0.39 (0.14) | 0.41 (0.14) |
| 800 | 50 | 135.95 | 374.61 | 5.35 | 6464.41 | 0.48 (0.02) | 0.49 (0.00) |

Table 10: The sequence length is fixed as 200. The feature dimension increases up to 50.

| Batch Size | Feat Dim | VAE | Diff | Sampling | GPU Mem | Disc Scr | Temp Disc Scr |
|---|---|---|---|---|---|---|---|
| 800 | 10 | 131.65 | 365.81 | 4.63 | 3288.57 | 0.20 (0.12) | 0.23 (0.14) |
| 800 | 20 | 128.34 | 344.86 | 4.53 | 3947.75 | 0.25 (0.13) | 0.29 (0.09) |
| 800 | 30 | 130.92 | 363.41 | 4.61 | 4358.76 | 0.17 (0.11) | 0.34 (0.14) |
| 800 | 40 | 132.03 | 359.15 | 4.58 | 4771.51 | 0.24 (0.19) | 0.38 (0.09) |
| 800 | 50 | 134.96 | 367.05 | 4.70 | 5185.07 | 0.32 (0.18) | 0.41 (0.10) |

Table 11: Same setting with Table 10, but the dimension of latent matrix is set as $200 \times F/2$.

# K   Experimental details of Imputation and Forecasting Tasks

| Model | Metric | Imputation (MCAR) | | | | Forecasting ($w = 24$) | | | |
|---|---|---|---|---|---|---|---|---|---|
| | | Bike Sharing | Traffic | Pollution | AirQuality | Bike Sharing | Traffic | Pollution | AirQuality |
| TimeAutoDiff | MAE | 0.4620 | **0.0481** | **0.0665** | 0.0442 | **0.0186** | 0.5202 | 0.1457 | **0.0541** |
| | MSE | 0.0996 | **0.0118** | **0.0110** | 0.0007 | 0.052 | **1.7045** | **0.1249** | **0.0097** |
| TimeLLM | MAE | 0.3080 | 0.3034 | 0.1184 | 0.0524 | 0.1554 | **0.5196** | 0.1528 | 0.0661 |
| | MSE | 0.0413 | 0.0721 | 0.0125 | 0.0008 | 0.0800 | 2.1470 | 0.1765 | 0.0221 |
| Moment | MAE | 0.6015 | 4.7193 | 1.2633 | 0.7330 | N.A. | N.A | N.A. | N.A. |
| | MSE | 0.1150 | 2.8289 | 0.2468 | 0.0609 | N.A. | N.A. | N.A. | N.A. |
| TEFN | MAE | 0.3764 | 0.3131 | 0.1238 | 0.0225 | 0.1493 | 0.5333 | 0.1466 | 0.0735 |
| | MSE | 2.0219 | 10.6461 | 0.8955 | 1.1066 | 0.0641 | 1.9177 | 0.1485 | 0.0194 |
| TimeMixer | MAE | 0.2052 | 0.2421 | 0.1128 | 0.0451 | 0.0809 | 0.5455 | 0.1576 | 0.0887 |
| | MSE | 0.0117 | 0.0598 | 0.0122 | 0.0008 | **0.0171** | 1.9208 | 0.1483 | 0.0243 |
| TimesNet | MAE | 0.2184 | 0.2786 | 0.2263 | 0.0494 | 0.1352 | 0.5473 | **0.1446** | 0.0766 |
| | MSE | 0.0137 | 0.0592 | 0.0341 | 0.0007 | 0.0519 | 1.9260 | 0.1451 | 0.0189 |
| MICN | MAE | 0.1829 | 0.1813 | 0.0968 | 0.0407 | 0.1249 | 0.5593 | 0.1642 | 0.0700 |
| | MSE | **0.0086** | 0.0434 | 0.0116 | 0.0006 | 0.0538 | 2.2501 | 0.2036 | 0.0251 |
| DLinear | MAE | 0.2897 | 0.2945 | 0.1428 | 0.1096 | 0.1358 | 0.5215 | 0.1449 | 0.0695 |
| | MSE | 0.0317 | 0.0731 | 0.0121 | 0.0022 | 0.0659 | 1.9290 | 0.1549 | 0.0188 |
| FiLM | MAE | 0.3944 | 0.3043 | 0.1233 | 0.0497 | 0.3135 | 0.5382 | 0.1451 | 0.0800 |
| | MSE | 0.0652 | 0.0736 | 0.0126 | 0.0008 | 0.4062 | 1.9271 | 0.1495 | 0.0213 |
| CSDI | MAE | 8.5081 | 0.0899 | 0.0907 | **0.0186** | 0.1339 | 0.5799 | 0.1603 | 0.0750 |
| | MSE | 59.2891 | 0.0262 | 0.0174 | 0.0004 | 0.1008 | 3.6405 | 0.2764 | 0.0302 |
| PatchTST | MAE | 0.4857 | 0.1175 | 0.0988 | 0.0200 | 0.5217 | 1.6593 | 0.2428 | 0.2850 |
| | MSE | 0.0825 | 0.0251 | 0.0113 | **0.0003** | 0.5706 | 8.1854 | 0.2552 | 0.0501 |
| iTransformer | MAE | **0.1634** | 0.2672 | 0.1022 | 0.0437 | 0.8169 | 2.5419 | 0.4112 | 0.6202 |
| | MSE | **0.0086** | 0.0609 | 0.0127 | 0.0007 | 0.9061 | 12.4538 | 0.3874 | 0.2580 |
| TimeMixerPP | MAE | 0.6939 | 0.7246 | 0.2469 | 0.1749 | 0.5968 | 2.3055 | 0.3815 | 0.5198 |
| | MSE | 0.1680 | 0.2450 | 0.0243 | 0.0067 | 0.4632 | 11.2804 | 0.3801 | 0.1865 |

Table 12: **Imputation (MCAR) and mid-horizon forecasting** ($w$=24). TimeAutoDiff delivers strong overall performance, achieving the best MAE/MSE on multiple datasets in both tasks, particularly on *Traffic* and *Pollution* for imputation and on *Bike Sharing*, *Traffic*, *Pollution*, and *AirQuality* for forecasting. Bold indicates the best score.

| Model | Metric | Forecasting (w=12) | | | | Forecasting (w=36) | | | |
|---|---|---|---|---|---|---|---|---|---|
| | | Bike Sharing | Traffic | Pollution | AirQuality | Bike Sharing | Traffic | Pollution | AirQuality |
| TimeAutoDiff | MAE | **0.0103** | 0.3942 | 0.1458 | **0.0689** | 0.0159 | **0.2544** | **0.1317** | 0.0156 |
| | MSE | **0.0027** | 1.3371 | 0.1610 | 0.0221 | **0.0014** | **0.2223** | **0.0558** | 0.0004 |
| TimeLLM | MAE | NaN | NaN | NaN | NaN | 0.1546 | 0.2785 | 0.1429 | 0.0655 |
| | MSE | NaN | NaN | NaN | NaN | 0.0823 | 0.6198 | NaN | 0.0190 |
| TEFN | MAE | 0.1537 | **0.3725** | 0.1377 | 0.0795 | 0.1330 | 0.2776 | 0.1381 | 0.0652 |
| | MSE | 0.0724 | 0.9662 | 0.1186 | 0.0207 | 0.0640 | 0.5744 | 0.1390 | 0.0167 |
| TimeMixer | MAE | 0.1430 | 0.3961 | **0.1372** | 0.0958 | 0.0574 | 0.3271 | 0.1394 | **0.0056** |
| | MSE | 0.0564 | 0.9710 | **0.1179** | 0.0268 | 0.0080 | 0.6007 | 0.1343 | **0.0001** |
| TimesNet | MAE | 0.1271 | 0.3785 | 0.1375 | 0.0744 | **0.0137** | 0.2820 | 0.1425 | 0.0234 |
| | MSE | 0.0569 | 0.9659 | 0.1187 | **0.0204** | 0.0016 | 0.5722 | 0.1363 | 0.0027 |
| DLinear | MAE | NaN | NaN | NaN | NaN | 0.1308 | 0.2687 | 0.1318 | 0.0616 |
| | MSE | NaN | NaN | NaN | NaN | 0.0681 | 0.5816 | 0.1438 | 0.0158 |
| FiLM | MAE | 0.3238 | 0.3736 | 0.1377 | 0.0823 | 0.2954 | 0.2712 | 0.1345 | 0.0692 |
| | MSE | 0.4193 | **0.9620** | 0.1242 | 0.0219 | 0.3851 | 0.5615 | 0.1391 | 0.0179 |
| CSDI | MAE | 0.1385 | 0.4616 | 0.1484 | 0.0718 | 0.1380 | 0.3025 | 0.1406 | 0.0476 |
| | MSE | 0.1176 | 2.0394 | 0.2048 | 0.0269 | 0.0990 | 1.0445 | 0.1593 | 0.0176 |
| PatchTST | MAE | 0.5193 | 1.3809 | 0.2749 | 0.2107 | 0.5535 | 1.2894 | 0.2327 | 0.1173 |
| | MSE | 0.7533 | 11.7008 | 0.3984 | 0.0471 | 0.2821 | 3.0525 | 0.1208 | 0.0114 |
| iTransformer | MAE | 0.8852 | 2.1533 | 0.4912 | 0.6517 | 0.8038 | 2.1488 | 0.4141 | 0.5340 |
| | MSE | 1.3915 | 18.0809 | 0.7799 | 0.4296 | 0.4269 | 4.7132 | 0.1799 | 0.1036 |
| TimeMixerPP | MAE | 0.5719 | 1.6961 | 0.3926 | 0.4514 | 0.6389 | 1.6801 | 0.3468 | 0.4651 |
| | MSE | 0.8420 | 13.3529 | 0.6534 | 0.2208 | 0.2537 | 3.5940 | 0.1693 | 0.0730 |

Table 13: **Long- and short-horizon forecasting.** `TimeAutoDiff` achieves competitive or state-of-the-art accuracy across both horizons, showing particularly strong performance at $w = 36$ (lowest MAE on *Traffic/Pollution* and lowest MSE on *Bike Sharing/Traffic/Pollution*). At $w = 12$, it attains the best MAE on *Bike Sharing/AirQuality* and best MSE on *Bike Sharing*, while other baselines lead in selected datasets. Bold denotes the best score.

| Model | Metric | Imputation (block_len = 3, block_width=2) | | | | Imputation (seq_len= 12) | | | |
|---|---|---|---|---|---|---|---|---|---|
| | | Bike Sharing | Traffic | Pollution | AirQuality | Bike Sharing | Traffic | Pollution | AirQuality |
| TimeAutoDiff | MAE | 0.6210 | **0.0626** | **0.1018** | 0.0416 | 0.6275 | **0.0619** | **0.1047** | 0.0442 |
| | MSE | 0.1777 | **0.0216** | 0.0167 | 0.0006 | 0.1938 | **0.0169** | 0.0163 | 0.0006 |
| TimeLLM | MAE | 0.3377 | 0.7765 | 0.1509 | 0.0749 | 0.2986 | 0.3892 | 0.1798 | 0.0959 |
| | MSE | 0.0348 | 0.2630 | 0.0131 | 0.0011 | 0.0238 | 0.0821 | 0.0168 | 0.0016 |
| MOMENT | MAE | 1.1657 | 3.3265 | 0.1520 | 0.1644 | 1.0849 | 2.8521 | 0.1810 | 0.5845 |
| | MSE | 0.3263 | 1.6206 | 0.0128 | 0.0042 | 0.2814 | 1.9517 | 0.0142 | 0.0477 |
| TEFN | MAE | 0.3445 | 0.4689 | 0.1300 | 0.0260 | 0.3887 | 0.3169 | 0.1350 | 0.0259 |
| | MSE | 2.0100 | 7.1343 | 0.8980 | 1.1044 | 2.0158 | 14.8765 | 0.8609 | 1.1044 |
| TimeMixer | MAE | 0.4839 | 0.5438 | 0.1389 | 0.0527 | 0.5316 | 0.3438 | 0.1463 | 0.0518 |
| | MSE | 0.0783 | 0.1969 | 0.0134 | 0.0008 | 0.0861 | 0.0877 | 0.0140 | 0.0008 |
| TimesNet | MAE | 0.2351 | 0.7233 | 0.3009 | 0.0660 | 0.2791 | 0.4186 | 0.2650 | 0.0579 |
| | MSE | 0.0144 | 0.4586 | 0.0370 | 0.0010 | 0.0197 | 0.1282 | 0.0349 | 0.0008 |
| MICN | MAE | 0.4087 | 0.7839 | 0.2100 | 0.3505 | 0.4412 | 0.9407 | 0.2151 | 0.3621 |
| | MSE | 0.0522 | 0.3624 | 0.0208 | 0.0168 | 0.0592 | 0.4592 | 0.0232 | 0.0177 |
| DLinear | MAE | 0.2600 | 0.6439 | 0.1532 | 0.0841 | 0.2672 | 0.5241 | 0.1623 | 0.0779 |
| | MSE | 0.0255 | 0.2428 | 0.0141 | 0.0014 | 0.0242 | 0.1237 | 0.0179 | 0.0012 |
| FiLM | MAE | 0.4382 | 0.5119 | 0.1312 | 0.0516 | 0.4938 | 0.3197 | 0.1637 | 0.0509 |
| | MSE | 0.0676 | 0.1988 | 0.0130 | 0.0007 | 0.0776 | 0.0783 | 0.0144 | 0.0007 |
| CSDI | MAE | 2.0009 | 0.1184 | 0.1208 | **0.0238** | 0.7977 | 0.1043 | 0.1254 | **0.0229** |
| | MSE | 11.7243 | 0.0348 | 0.0198 | 0.0005 | 0.4599 | 0.0287 | 0.0203 | **0.0004** |
| PatchTST | MAE | **0.1899** | 0.1983 | 0.1238 | 0.0337 | **0.1878** | 0.1754 | 0.1118 | 0.0332 |
| | MSE | **0.0095** | 0.0348 | **0.0127** | **0.0004** | **0.0099** | 0.0291 | **0.0111** | **0.0004** |
| iTransformer | MAE | 0.2161 | 0.5851 | 0.1535 | 0.0501 | 0.2202 | 0.3511 | 0.1713 | 0.0508 |
| | MSE | 0.0156 | 0.2033 | 0.0129 | 0.0007 | 0.0150 | 0.0832 | 0.0154 | 0.0007 |
| TimeMixerPP | MAE | 0.6323 | 1.3245 | 0.3046 | 0.2598 | 0.6785 | 1.2450 | 0.3373 | 0.3353 |
| | MSE | 0.1517 | 0.6668 | 0.0332 | 0.0116 | 0.1654 | 0.7089 | 0.0364 | 0.0191 |

Table 14: **Imputation with structured masks (block vs. sequential).** Block (`block_len`=3, `block_width`=2) and Sequential (`seq_len`=12). `PatchTST` shows strong performance, leading MAE/MSE on *Bike Sharing* (both masks) and *Pollution* (MSE). `TimeAutoDiff` consistently leads *Traffic* (MAE/MSE) and *Pollution* (MAE). `CSDI` and `TEFN` perform best on *AirQuality*. Bold = best.

# L   Additional Plots: Auto-Correlation / periodic, cyclic patterns

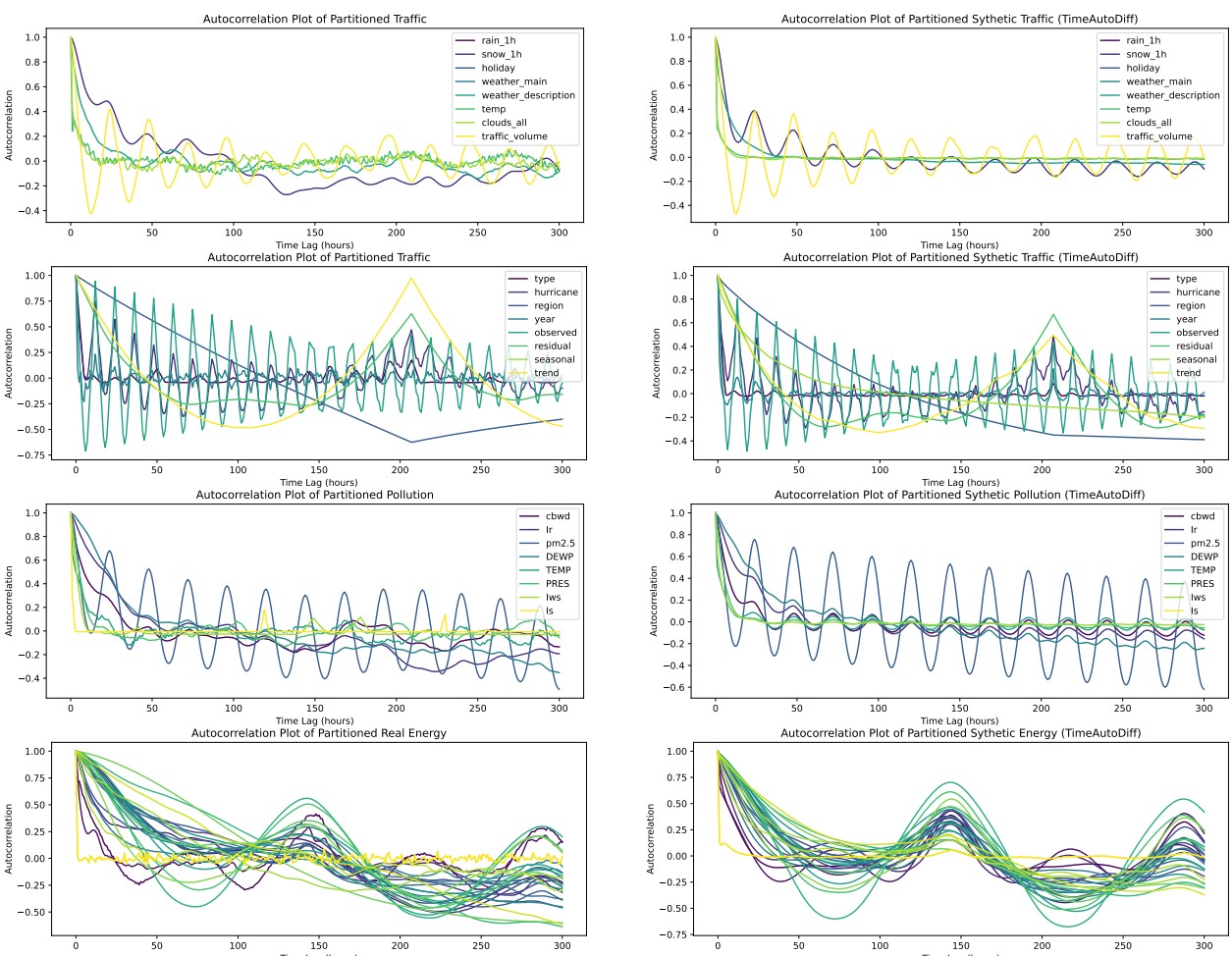

Figure 15: The four plots from the top are auto-correlation plots of lag 300 for real (left) and synthetic (right) of 'Traffic','Hurricane','Pollution', and 'Energy'.

## M    Plots of Synthethic v.s. Real on TV-MCG from `TimeAutoDiff`

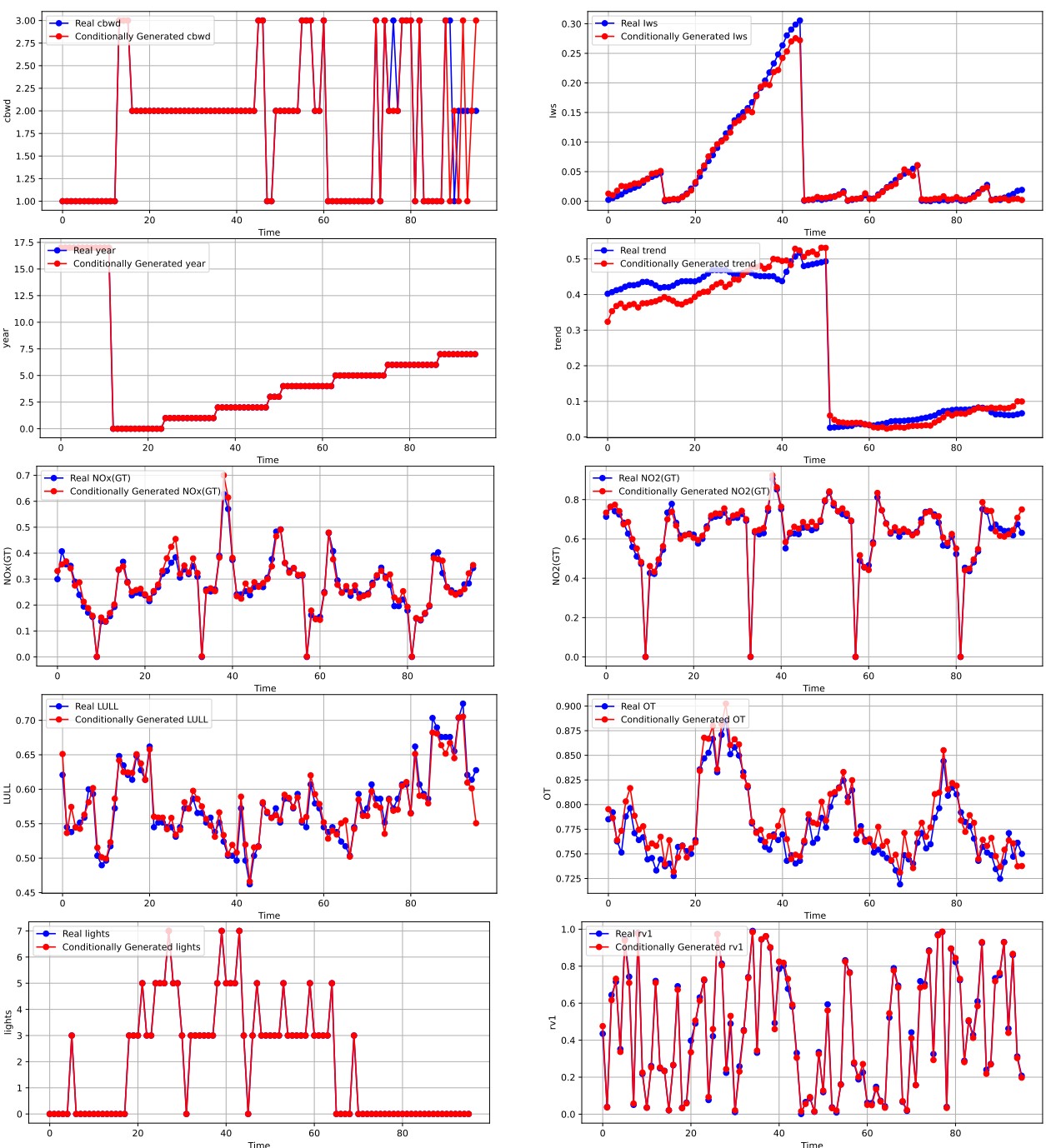

Figure 16: Datasets: (output variables) from top to bottom: **Pollution**: ('cbwd', 'Iws'), **Hurricane**: ('year', 'trend'), **AirQuality**: ('NOx(GT)', 'NO2(GT)'), **ETTh1**: ('LULL', 'OT'), **Energy**: ('lights', 'rv1'). The output is chosen to be heterogeneous (except AirQuality & ETTh1) both having discrete and continuous variables. Conditional variables **c** are set as remaining variables from the entire features. See the list of entire features of each dataset through the link in Appendix B.

