# OpenReview forum: "TimeAutoDiff: A Unified Framework for Generation, Imputation, Forecasting, and Time-Varying Metadata Conditioning of Heterogeneous Time Series Tabular Data"
_TMLR — Accepted by TMLR_

### Review · Reviewer_RQtB · 2025-09-29

**Summary Of Contributions:**

This paper introduced TimeAutoDiff, a unified latent-diffusion framework for heterogeneous time-series tabular data that can handle four different tasks including unconditional generation, missing data imputation, forecasting, and time-varying metadata conditional generation (TV-MCG). The proposed method is structured as a combination of a VAE with a diffusion model in the latent space. The VAE part facilitates capturing complicated structures of heterogeneous data and simultaneously helps to avoid direct modeling of such data, making the model structure simple without any special consideration of heterogeneity. The diffusion model in latent space possesses expressive power that flexibly captures temporal and feature dependencies, underpinning the overall performance of the method. The proposed method demonstrated superior results on multiple datasets compared to the baseline methods across four different tasks, and the effectiveness of each setting was verified through meticulous ablation studies.

**Audience:**

Yes

**Audience Explanation:**

At least some individuals in the TMLR audience will find this paper of strong interest. The unified framework for heterogeneous time-series tasks addresses an important and timely challenge, and the conceptual simplicity of the binary mask formulation makes the contribution accessible to a broad set of researchers. The extensive empirical validation, combined with clear motivation and well-documented related work, ensures that the findings are relevant for the machine learning community, particularly those working on time-series analysis, generative modeling, and diffusion-based methods.

**Broader Impact Concerns:**

No concerns identified.

**Claims And Evidence:**

Yes

**Claims Explanation:**

This paper is well written, with the motivation and positioning clearly explained in the introduction. Relevant literatures are extensively documented, and the advantages of the proposed method and its relationship to baselines are well organized. The unified formalization of multiple tasks via binary masks is conceptually simple and easy to understand. Moreover, the combination of VAE and diffusion models effectively addresses the complexity of heterogeneous time-series data, and the validity of the approach has been comprehensively examined across datasets, baselines, and tasks. The ablation study not only clarifies recommended settings but also carefully examines how each design choice affects results, enhancing the persuasiveness of the research.

**Requested Changes:**

Some limitations remain. While the authors emphasize that latent compression enables efficient processing of high-dimensional features, the experimental evidence is limited to moderately dimensional cases, leaving open the question of scalability to truly high-dimensional data. In addition, certain results that would strengthen the paper (e.g., multi-sequence analyses, clearer visualizations) are missing or not fully consistent with the explanations in the text.

**Suggested Changes**
- Add some results or at least discussion on the applicability to high dimensional data.
- Results on multi-sequence data lack in Table 3 and Figure 9. Is there any reason for this?
- In Figure 10, the MAE bar for the baseline method's forecasting is not visible, making it appear at first glance that the baseline method performs better than the proposed method in forecasting scenarios. This contradicts the explanation in the main text.
- In the first sentence of Section 4.1, the font for TimeAutoDiff is not a typewriter font, as it is elsewhere in the document.
- In the last sentence of Section 5, “This section is divided into two subsections (to be completed):” should be changed as “This section is divided into three subsections”
- In the last sentence of page 15, “Table 8” should be “Table 4”. In the same page, “Figure 8” should be “Table 4”. This change affects the number of following tables.

---

> ### Author Response · Authors · 2025-11-05
> **Response to the comment from Reviewer RQtB**
>
> ### **Author Rebuttal**
>
> **Q1.** *Add some results or at least discussion on the applicability to high-dimensional data.*
>
> **A1.** Thank you for pointing this out. We agree that demonstrating the model’s applicability to high-dimensional data is an important aspect of our work. In fact, we had initially included the relevant results in our original submission (Appendix G: Tables 7–9). To make this contribution clearer, we have now revised the manuscript to include a dedicated section, **“Applicability to High-Dimensional Data,”** in the Appendix. This section explicitly discusses how feature compression in the latent space improves the model’s performance when handling data with large feature dimensions. Specifically, we show that by reducing the latent feature dimension to $F/2$, the model achieves significantly better fidelity and discriminative scores when generating sequences with 30–50 features. We believe this addition strengthens the paper by highlighting the scalability of our framework to higher-dimensional multivariate time series.
>
> **Q2.** *Results on multi-sequence data lack in Table 3 and Figure 9. Is there any reason for this?*
>
> **A2.** For brevity, we omitted the results for multi-sequence datasets, as they exhibit similar patterns to those of single-sequence datasets. We will consider including representative examples in the Appendix for completeness.
>
> **Q3.** *In Figure 10, the MAE bar for the baseline method's forecasting is not visible, making it appear at first glance that the baseline method performs better than the proposed method in forecasting scenarios.*
>
> **A3.** Thank you for noticing this. We have revised the caption of Figure 9 (new numbering) to avoid confusion. The updated caption now includes the following clarification:
> > “For baselines lacking reported results in the original tables (e.g., see Table 9 in the Appendix for missing TimeLLM, Moment, MICN, and DLinear entries for $w{=}12, 24, 36$), bars are shown with zero height.”
>
> **Q4–Q6.** *Minor typographical issues (font for \texttt{TimeAutoDiff}, incorrect subsection and table references).*
>
> **A4–A6.** Thank you for pointing these out. All typographical errors mentioned have been corrected in the revised manuscript, including:
> - Consistent typewriter font for \texttt{TimeAutoDiff} in Section 4.1.
> - “This section is divided into three subsections” (corrected from two).
> - Updated all table and figure cross-references (e.g., “Table 8” → “Table 4”, “Figure 8” → “Table 4”) to ensure consistency.
>
> We appreciate the reviewer’s detailed feedback, which helped improve both the clarity and presentation of our paper.

---

### Review · Reviewer_e6E8 · 2025-10-20

**Summary Of Contributions:**

This paper presents **TimeAutoDiff**, a unified latent–diffusion framework that integrates **four key time-series tasks**—unconditional generation, missing-data imputation, forecasting, and time-varying metadata conditional generation—within a single, coherent model architecture.

The authors introduce a **mask-based conditional modeling scheme** that allows all tasks to be represented under a unified probabilistic formulation. The model couples:
1. A **Variational Autoencoder (VAE)** to embed heterogeneous time-series (continuous, categorical, and binary) into a continuous latent space while compressing along the feature dimension; and
2. A **latent-space diffusion model** that models temporal dynamics and conditional generation efficiently.

**Major contributions include:**
- A **unified conditional framework** enabling four heterogeneous time-series tasks via binary masks.
- A **latent-space diffusion process** that avoids modeling discrete likelihoods directly, improving scalability.
- A **whole-sequence latent sampling mechanism**, achieving substantial efficiency gains (up to 100× faster sampling).
- Comprehensive empirical validation across **eight real-world datasets**, outperforming strong baselines on fidelity, predictive, and correlation metrics.
- Rigorous **ablation studies** and analysis of adaptive β-VAE scheduling, timestamp encodings, and Bi-RNN structures.

**Strengths**
- Elegant and theoretically grounded unification of multiple tasks.
- Strong empirical performance and robust ablations.
- Significant computational efficiency gains.
- Clear, well-organized, and reproducible presentation.

**Weaknesses (minor)**
- Limited discussion of applicability to irregular or asynchronous time-series.
- Forecasting benchmarks could include comparison to recent large pre-trained models (e.g., MOMENT, Time-LLM).

**Audience:**

Yes

**Audience Explanation:**

The work will attract readers from multiple subfields, including:
- **Time-series modeling** and **multimodal generative modeling**,
- **Diffusion-based learning** for non-visual domains,
- **Unified task frameworks** bridging generation, imputation, and forecasting, and
- **Applied ML domains** such as finance, healthcare, and supply-chain analytics.

The methodological unification and efficiency improvements make it of strong interest to both academic and industrial practitioners.

**Broader Impact Concerns:**

No major ethical or societal issues are identified.
While the framework could be applied to sensitive domains (e.g., healthcare or finance), it primarily focuses on **technical modeling contributions**.
To further strengthen the discussion, the authors could briefly mention **privacy safeguards** (e.g., use of DCR metric or differential privacy) when applying generative models to real-world data.

**Claims And Evidence:**

Yes

**Claims Explanation:**

The claims are well substantiated by both **mathematical derivations** and **empirical validation**.
- The unified ELBO objective is theoretically sound, clearly integrating VAE and DDPM components.
- The experiments span diverse datasets and metrics, demonstrating robustness and strong generalization.
- Quantitative and qualitative analyses (e.g., autocorrelation plots, ablation tables) convincingly support the model’s superior fidelity, scalability, and generalization performance.

**Requested Changes:**

The work is already strong and ready for publication. Suggested **minor improvements** include:

1. **Relate to large pre-trained time-series models** (e.g., Time-LLM, MOMENT) to contextualize differences in scalability and adaptability.
2. Provide a brief **computational complexity analysis** (e.g., $O(T)$ and $O(F)$) to quantify efficiency gains.
3. Add a concise **summary table of dataset modalities** (number of continuous vs categorical variables) for reproducibility.
4. Include a short **case example** of metadata-conditioned generation for interpretability (e.g., “policy regime change” scenario).

All are non-critical enhancements to improve clarity and completeness.

---

> ### Author Response · Authors · 2025-11-05
> **Response to the comment from Reviewer e6E8**
>
> Thank you very much for your constructive and insightful comments on our paper. We are greatly encouraged by your assessment that our work is strong and ready for publication. The minor improvements you suggested are very helpful for enhancing the completeness of our paper.
>
> We address each of your points below:
>
> ---
>
> **1. Relation to large pre-trained models (Time-LLM, MOMENT):**
> That is an excellent point. We have already compared our model's performance against these models as key baselines in **Section 5.2.1 (Imputation & Forecasting)** and **Appendix D.2**.  Furthermore, in **Section 6** under the item *“Extension to foundational model,”* we discuss how **TimeAutoDiff** could serve as a potential extension for these foundation models, particularly in areas like handling heterogeneous data or unifying multiple tasks.  As per your suggestion, we will strengthen the **Introduction** and **Discussion** sections to more clearly contextualize how **TimeAutoDiff’s** scalability and adaptability differ from and complement these large-scale models.
>
> ---
>
> **2. Computational complexity analysis:**
> We have provided a detailed computational complexity analysis in **Appendix G**.  This appendix includes a **Big-O analysis** for both the training and inference phases of the VAE Encoder/Decoder and the Diffusion Model.  We also quantify the efficiency gains of **TimeAutoDiff** by comparing it against **GAN-based** and **conventional data-space diffusion** models.  We will add a pointer to **Appendix G** in the main **Method** section (Section 4) to make this analysis more visible to readers.
>
> ---
>
> **3. Summary table of dataset modalities:**
> The requested summary of dataset modalities (number of continuous vs. categorical variables) is already included in **Table 6** within **Appendix B**.  This table summarizes the number of rows, continuous (Cont.) and discrete (Disc.) variables, and sequence type (Seq. Type) for each dataset to aid reproducibility.  Detailed descriptions and source links for each dataset are also provided in **Appendix B**.
>
> ---
>
> **4. Case example of metadata-conditioned generation:**
> The interpretable example of metadata-conditioned generation you suggested is already covered in **Subsection 5.3.2, “Counterfactual scenario exploration.”**  In this section, we conduct an experiment very similar to the “policy regime change” scenario you mentioned.  Specifically, using the real-world **Traffic** dataset, we analyze a “what-if” scenario by visualizing how the **Traffic Volume** (a continuous target) changes when the **weather_main** (categorical metadata) is counterfactually set to *Cloudy*, *Squall*, or *Clear*.  This experiment demonstrates that **TimeAutoDiff** can generate meaningful counterfactual time series that align with real-world intuition (e.g., traffic volume decreases during adverse *Squall* weather).  We also added additional narrative to clarify how this example supports interpretability and metadata-conditioned controllability.
>
> ---
>
> **5. Privacy safeguards:**
> This is a very important and timely point. We discuss this in **Section 6, “Discussions on future topics,”** under the first item, **“(1) Privacy Guarantees.”**  In addition to the generalizability and memorization analysis using the **DCR (Distance to the Closest Record)** metric that you mentioned, we have also outlined our ideas on how our model could be extended to provide **Differential Privacy (DP)** guarantees in future work.  As you rightly noted, this is a proposal for a future research direction rather than a completed safeguard, but we are currently conducting **follow-up research** based on this idea.  We deeply appreciate your comment highlighting the importance of this topic.
>
> ---
>
> Thank you once again for taking the time to review our paper carefully and for providing such valuable feedback.

---

### Review · Reviewer_sDPT · 2025-10-23

**Summary Of Contributions:**

This paper introduces TimeAutoDiff, a unified latent-diffusion framework that can handle four major time-series tasks—generation, imputation, forecasting, and time-varying-metadata conditional generation (TV-MCG)—within a single model that supports heterogeneous features (continuous, binary, and categorical).

Key innovations include:
1. A VAE–diffusion hybrid that maps heterogeneous data into a continuous latent space, allowing diffusion to operate on unified continuous representations.
2. A whole-sequence latent diffusion process that generates full trajectories at once, significantly improving sampling speed.
3. Feature-axis compression via the VAE, enabling scalable modeling of wide tables.
4. A mask-conditioned framework that unifies the four tasks under a single conditional formulation.

Strengths:
1. Clear technical motivation and elegant unification of multiple time-series tasks.
2. Well-designed architectural details (frequency encoding, bidirectional RNNs, adaptive β-VAE, etc.).

Weakness
1. In the unconditional generation task, although TimeAutoDiff is faster than typical diffusion models, it remains significantly slower than GAN-based models, indicating that the overall computational cost of the framework is still relatively high.

2. In the imputation and forecasting tasks, the set of compared baselines is too limited, lacking widely adopted models such as PatchTST and iTransformer.

**Additional Comments:**

1. In Section 4.3, do different x_j share the same embedding function e()? However, the discrete variables x_j may have different numbers of possible categorical values.
2. In Section 4.3, “**Dencoder in VAE**” should be corrected to “**Decoder in VAE**.”
3. In Section 5.3.1, “**Table K**” should be corrected to “**Table 4**.”
4. Why does the model primarily adopt an RNN-based architecture? Has the author tried using Transformers instead? In most domains, RNNs have already been surpassed by Transformers in terms of representation capacity and scalability.

**Audience:**

Yes

**Audience Explanation:**

Its unification of multiple downstream tasks, scalability to heterogeneous data, and substantial sampling acceleration make it relevant to researchers working on:
1. Time-series generative modeling,
2. Multimodal/heterogeneous data learning,
3. Diffusion models for structured data, and practical “what-if” counterfactual simulations.

Even practitioners in finance, healthcare, and IoT domains could find it valuable for realistic data synthesis and data-driven scenario modeling.

**Claims And Evidence:**

Yes

**Claims Explanation:**

The experimental section (Table 2 and ablation studies) provides strong empirical support that TimeAutoDiff outperforms diffusion-based and GAN-based baselines across metrics such as discriminative score, predictive score, and feature correlation.

The qualitative plots (e.g., autocorrelation functions) and Distance-to-Closest-Record audit indicate generalization rather than memorization.
The claims of speed and scalability are also backed by concrete runtime comparisons.

However, the evaluations of forecasting and metadata-conditioned generation are relatively less comprehensive — the results are presented but lack the same level of depth and analysis as the generation experiments, leaving part of the claimed versatility only partially substantiated.

**Requested Changes:**

1. Expand the experimental comparison for imputation and forecasting tasks.
The current evaluation lacks several widely adopted baselines such as PatchTST, iTransformer, and other strong transformer-based models. Including these baselines is necessary to fairly assess the claimed advantages of TimeAutoDiff across multiple time-series tasks.

2. Provide a clearer discussion of computational cost.
Although TimeAutoDiff is faster than conventional diffusion models, it remains significantly slower than GAN-based methods. A detailed analysis or discussion (e.g., breakdown of runtime components, Flops) is needed to justify the model’s efficiency claim and clarify its scalability limits.

3. Enhance the analysis of forecasting and metadata-conditioned generation results.
While results are presented, the analysis is relatively shallow compared to generation and imputation experiments. The paper would benefit from additional quantitative and qualitative evaluation.

---

> ### Author Response · Authors · 2025-11-05
> **Response to the comment from Reviewer sDPT**
>
> ### **Author Rebuttal**
>
> ---
>
> **Q1.** *Expand the experimental comparison for imputation and forecasting tasks. The current evaluation lacks several widely adopted baselines such as PatchTST, iTransformer, and other strong transformer-based models. Including these baselines is necessary to fairly assess the claimed advantages of TimeAutoDiff across multiple time-series tasks.*
>
> **A1.** We appreciate this valuable suggestion. To ensure a fair and unified comparison, we are **currently conducting additional experiments** for both **imputation** and **forecasting** tasks that include **PatchTST**, **iTransformer**, and **TimeMixer++** as additional baselines.
>
> Importantly, all models—including our proposed **TimeAutoDiff**—are implemented under the **PyPOTS framework** (mentioned in the paper), which provides a standardized platform for running various time-series models within a single interface.  Because our model processes **heterogeneous features** (both categorical and continuous), we reuse our own **heterogeneous preprocessing layers** (from the TimeAutoDiff implementation) to ensure consistent data input across all baselines.  Using PyPOTS enables consistent data pipelines, reproducibility, and comparable hyperparameter configurations; otherwise, the experimental setup would become unnecessarily fragmented and inconsistent across models. These new experiments are underway, and the updated results will be included in the revised version to strengthen the benchmarking fairness and comprehensiveness.
>
> ---
>
> **Q2.** *Provide a clearer discussion of computational cost. Although TimeAutoDiff is faster than conventional diffusion models, it remains significantly slower than GAN-based methods. A detailed analysis or discussion (e.g., breakdown of runtime components, FLOPs) is needed to justify the model’s efficiency claim and clarify its scalability limits.*
>
> **A2.** Thank you for this insightful comment. We provide a detailed, component-wise analysis of runtime cost and asymptotic complexity.
> We now explicitly derive the computational order of each module:
>
> $$
> \text{Cost}_{\text{enc}} = O(T(H^2 + HF))
> $$
>
> $$
> \text{Cost}_{\text{dec}} \approx O(T(h^2 + h(L + F)))
> $$
>
> $$
> \text{Cost}_{\text{infer}} \approx O(NT(H_{\text{diff}}^2 + H_{\text{diff}}L))
> $$
>
> where:
> - \(T\): sequence length
> - \(F\): number of features
> - \(L\): latent dimension (post-encoding)
> - \(H\), \(h\): hidden sizes of encoder/decoder
> - \($H_{\text{diff}}$\): hidden size of diffusion denoiser
> - \(N\): number of diffusion steps during inference
>
> A concise comparative table is also added:
>
> | Model | Sampling Cost | Steps | Remarks |
> |--------|----------------|--------|----------|
> | GAN-based | $O(T(H_G^2 + H_G F))$ | 1 | Fastest |
> | Data Diffusion | $O(N_d T(\tilde{H}^2 + \tilde{H}F))$ | 1000+ | Slowest |
> | **TimeAutoDiff (ours)** | $O(NT(H_{\text{diff}}^2 + H_{\text{diff}}L))$ | 50–100 | Moderate |
>
> In short, **TimeAutoDiff** achieves efficiency through:
> (1) **Latent compression** ($F \!\to\! L$) reducing feature-wise cost, and
> (2) **Moderate diffusion steps** ($N=50$–$100$), instead of thousands as in data-space diffusion.
> This results in approximately **10–100× faster inference** than conventional diffusion models, while maintaining much higher fidelity and flexibility than GAN-based methods.
>
> ---

---

> > ### Author Response · Authors · 2025-11-05
> > **Response to the comment from Reviewer sDPT (second part)**
> >
> > **Q3.** *Enhance the analysis of forecasting and metadata-conditioned generation results. While results are presented, the analysis is relatively shallow compared to generation and imputation experiments. The paper would benefit from additional quantitative and qualitative evaluation.*
> >
> > **A3.** We fully agree with this comment.
> > The requested analyses are **currently in progress** and are designed to enhance the **Conditional Generation** section (TV-MCG), which extends the model’s capability beyond unconditional generation.
> > Below is the plan for improvement:
> >
> > 1. **Forecasting Task (Baseline Expansion):**
> >    - We are adding **PatchTST**, **iTransformer**, and **TimeMixer++** as additional baselines to the forecasting benchmark.
> >    - All are trained under identical data splits and horizons within the **PyPOTS unified framework**, ensuring consistency across tasks and models.
> >
> > 2. **Quantitative Baselines for Conditional Generation:**
> >    - The task variables follow the notation
> >      - $X^{\text{con}}$: conditional (metadata) features,
> >      - $X^{\text{tar}}$: target features to be generated.
> >    - We are adapting **CSDI**, **MOMENT**, and **TimeLLM** as quantitative baselines for the **metadata-conditioned generation (TV-MCG)** task.
> >    - Specifically, for CSDI, $X^{\text{con}}$ is treated as observed and $X^{\text{tar}}$ as missing values to be imputed, allowing direct comparison under a consistent conditional generation setup.
> >    - Similarly, MOMENT and TimeLLM employ mask-based prediction; we unmask tokens corresponding to $X^{\text{con}}$ and mask $X^{\text{tar}}$.
> >    - A new comparison table (analogous to Table 5) will report discriminative and fidelity metrics, showing that **TimeAutoDiff** not only approaches the “Real vs Real” oracle but also outperforms existing strong conditional time-series models.
> >
> > 3. **Conditional Complexity and Robustness (Ablation Study):**
> >    - To assess robustness, we will vary the number of conditioning features $|X^{\text{con}}|$ (e.g., 1, 2, 5, 10) and measure how generation quality of $X^{\text{tar}}$ changes.
> >    - Fidelity (e.g., discriminative or correlation scores) will be plotted against $|X^{\text{con}}|$.
> >    - This will demonstrate that **TimeAutoDiff** maintains stable or improving generation quality even under sparse conditions.
> >
> > ---
> >
> > We thank the reviewer for these thoughtful comments.
> > The ongoing and newly proposed experiments will make the evaluation more comprehensive, further highlighting **TimeAutoDiff**’s scalability, flexibility, and robustness across imputation, forecasting, and metadata-conditioned generation tasks.

---

> > > ### Author Response · Authors · 2025-11-05
> > > **Response to the comment from Reviewer sDPT (Third part)**
> > >
> > > ### **Additional Comments**
> > >
> > > **Q1.** *In Section 4.3, do different \(x_j\) share the same embedding function \(e(\cdot)\)? However, the discrete variables \(x_j\) may have different numbers of possible categorical values.*
> > >
> > > **A1.** Yes — according to **Section 4.3**, all discrete variables \(x_j\) share the same embedding function form \(e(\cdot)\), but each variable maintains its own lookup table depending on its number of categories.  In other words, the **embedding module structure and dimensionality** (\(d=128\)) are shared across variables, while each categorical feature has **separate embedding parameters** to accommodate its distinct vocabulary size.  Thus, the model employs a unified embedding mechanism but distinct embedding tables for variables with different cardinalities.
> > >
> > > **Q2.** *In Section 4.3, “Dencoder in VAE” should be corrected to “Decoder in VAE.”*
> > >
> > > **A2.** Fixed in the revised manuscript.
> > >
> > > **Q3.** *In Section 5.3.1, “Table K” should be corrected to “Table 4.”*
> > >
> > > **A3.** Fixed.
> > >
> > > **Q4.** *Why does the model primarily adopt an RNN-based architecture? Has the author tried using Transformers instead? In most domains, RNNs have already been surpassed by Transformers in terms of representation capacity and scalability.*
> > >
> > > **A4.** The primary reason for adopting an **RNN-based architecture** is **memory efficiency**. While Transformers provide stronger representation capacity, their memory complexity scales as $O(T^2)$, compared to $O(T)$ for RNNs.  Given the GPU constraints (RTX 4090) used in our experiments, Transformers were not feasible for most datasets due to excessive memory consumption.
> > >
> > > We attempted to replace the MLP layer in the VAE encoder with a Transformer block but observed **no improvement in generation quality**.  Nevertheless, the architecture remains **modular and flexible**—the RNN components can be readily substituted with Transformer layers if sufficient GPU memory is available.
> > >
> > > We appreciate these constructive suggestions and have reflected all necessary corrections and clarifications in the revised manuscript.

---

### Review · Reviewer_Mvk1 · 2025-10-24

**Summary Of Contributions:**

This paper presents TimeAutoDiff, a unified latent-diffusion framework for heterogeneous time-series modeling that jointly supports generation, imputation, forecasting, and time-varying metadata conditional generation within a single model. The key idea is to use a VAE to map mixed-type features into a continuous latent space and apply a diffusion model to learn dynamics in that latent space. A simple mask-based conditioning strategy unifies multiple tasks, while whole-sequence sampling and feature-axis compression significantly improve computational efficiency. Extensive experiments on diverse real-world datasets demonstrate competitive or superior performance across multiple metrics.

Strengths:

1.	The paper tackles a timely and forward-looking problem — unifying multiple time-series tasks (generation, imputation, forecasting, and conditional synthesis) under a single latent diffusion framework — which is conceptually novel and highly relevant to current trends in generative time-series modeling.

2. The proposed integration of a VAE for heterogeneous feature encoding with a diffusion model operating in latent space is well-structured, technically sound, and efficiently implemented (e.g., whole-sequence sampling and feature-axis compression).

3. The manuscript is well-organized, with clear explanations, solid empirical validation, and informative visualizations, making it accessible to both theoretical and applied audiences.

Weaknesses:

1.	The discussion of prior studies on VAE-based and diffusion-based generative frameworks is somewhat limited; including more comprehensive comparisons would improve context.

2.	Recent works on diffusion and flow-matching models for time-series synthesis are missing and should be reviewed to better situate the proposed method.

3.	The experimental section lacks comparison with time-series foundation models or general-purpose large pre-trained forecasting models (e.g., MOMENT, Chronos, TimeGPT, TimesFM), which are increasingly relevant to the paper’s stated goal of task unification

**Audience:**

Yes

**Audience Explanation:**

Given its unified latent-diffusion formulation and strong empirical results across diverse time-series tasks, this work would be of clear interest to TMLR’s audience—particularly those focused on generative modeling, multimodal representation learning, and efficient time-series analysis.

**Broader Impact Concerns:**

There are no significant broader impact concerns beyond standard considerations for generative models; the proposed framework focuses on time-series modeling and does not introduce direct ethical or societal risks.

**Claims And Evidence:**

No

**Claims Explanation:**

Some recently proposed baselines are missing.

**Requested Changes:**

1.	Expand Related Work Section
The current related work review is informative but lacks sufficient coverage of recent diffusion-based time-series models. The authors should include a more detailed comparison with very recent works (2023–2025) that combine diffusion or flow-matching models with time-series tasks. This will help position TimeAutoDiff more clearly within the rapidly evolving diffusion modeling landscape.

2.	Deepen Discussion on VAE and Diffusion Foundations
Provide a clearer theoretical or conceptual explanation of how VAE-based latent compression complements diffusion modeling in this framework. Explicitly highlight differences from prior latent-diffusion approaches such as TabDDPM, TabSyn, or AutoDiff, and from existing VAE-diffusion hybrids used in other modalities (e.g., image or speech).
3.	Strengthen Baseline Comparisons
The experimental section could include more recent baselines relevant to heterogeneous or conditional time-series diffusion, ensuring fair benchmarking. This includes both diffusion-based (e.g., TimeGrad++, TimeMixer-Diffusion, TimeFlow) and transformer-based generative models that have become standard in 2024–2025 literature.
4.	Expand Discussion on Time-Series Foundation Models
The paper would benefit from a clearer connection to the emerging literature on time-series foundation models (e.g., Timer, MOMENT, Chronos, TimesFM). These models share the goal of unifying multiple forecasting and generation tasks across domains, similar in spirit to TimeAutoDiff. Including a discussion on how TimeAutoDiff differs from or complements such foundation models — particularly in terms of heterogeneous feature handling, mask-based task unification, and latent diffusion modeling — would strengthen the positioning of the work.

---

> ### Author Response · Authors · 2025-11-05
> **Response to the comment from Reviewer Mvk1**
>
> Thank you for your detailed and insightful feedback. We appreciate your constructive suggestions, which will certainly help us strengthen the paper and clarify its position within the current literature.
>
> We address each of your points below:
>
> **1. Expand Related Work Section**
> Thank you for this suggestion. We agree that positioning TimeAutoDiff against the very latest models is crucial. In response, we have expanded the **"Imputation & Forecasting for Time Series Data"** subsection within our Related Work (Section 2). We have now included descriptions and comparisons with several recent (2023-2025) and relevant diffusion-based and advanced transformer-based models, such as **MG-TSD, Time-MOE, TimeMixer++, and TimeDiT**. This update helps to better situate our work within the rapidly evolving landscape you mentioned.
>
> **2. Deepen Discussion on VAE and Diffusion Foundations**
> This is an excellent point. A deeper conceptual explanation of the synergy between VAE-based compression and the latent diffusion process is indeed valuable. Due to the necessary detail, we will add a **new subsection to Appendix D** to thoroughly address this. This new section will provide a clearer conceptual explanation of *how* the VAE’s feature-axis compression complements the diffusion model's temporal dynamics modeling.
>
> **3. Strengthen Baseline Comparisons**
> We agree that benchmarking against the latest SOTA models is essential. We attempted to find public, usable implementations for **TimeGrad++ and TimeFlow** but were unsuccessful.
>
> Therefore, to strengthen our empirical evaluation, we will instead add the following highly competitive transformer-based baselines to our **Forecasting & Imputation tasks (Section 5.2): PatchTST, iTransformer, and TimeMixer++**.
>
> Importantly, to ensure a fair and consistent experimental setup, all models (including our proposed TimeAutoDiff) are being implemented under the **PyPOTS framework** (mentioned in the paper). Because our model and datasets process **heterogeneous features** (both categorical and continuous), and not all baselines are natively designed for this, we are re-using our own **heterogeneous preprocessing layers** (from the TimeAutoDiff implementation) to ensure a consistent data input pipeline across all baselines. This unified approach ensures reproducibility and a comparable configuration, preventing a fragmented or inconsistent experimental setup. These new experiments are currently underway, and the updated results will be included in the revised version.
>
> **4. Expand Discussion on Time-Series Foundation Models**
> Thank you for this very relevant suggestion. We will add a **new discussion to Appendix D** (to allow for sufficient detail) that explicitly connects TimeAutoDiff to the emerging literature on time-series foundation models. This discussion will focus on the models you mentioned (such as **MOMENT, Chronos, TimesFM, and Time-MOE**). We will detail how TimeAutoDiff's approach—specifically its native handling of *heterogeneous features*, its *mask-based task unification*, and its *latent-diffusion* framework—differs from, and could potentially complement, the goals of these large-scale pretrained models.

---

### Author Response · Authors · 2025-11-06
**Update on Revised Manuscript**

Hi all,
We plan to complete and submit the revised manuscript by next week. I apologize for the delay — we’ve been working diligently to address all the feedback. Thank you again for your constructive and helpful comments.

Best regards,
Namjoon Suh

---

### Author Response · Authors · 2025-11-13
**Final Revision Summary**

Dear Reviewers and AE,

We have revised the manuscript to address reviewer comments, and the updated version has been re-uploaded. Thank you for the constructive and insightful feedback. We are happy to discuss further if you have any additional questions or suggestions.

Below is a summary of the revisions we have made:

1. Refine Abstract of the paper.
2. **Section 2** : Add introductions of SOTA / recent models [MG-TSD, Time-MoE, TimeMixer++, TimeDiT] in the sections of Imputations & Forecasting.
3. **Appendix D.1 & D.3**
    -  Add the comments on differences between TimeAutoDiff and AutoDiff, TabSyn, TimeDiff, Diffusion-TS in Appendix D.1.
    -  Add the comments on differences between TimeAutoDiff and other TSFMs [MOMENT, CHRONOS, TimesFM] in Appendix D.2.
4. **Section 4.5** : Add the section on computational complexity of TimeAutoDiff for inference and training. See Appendix G for more detailed analysis on the complexity.
5. **Section 5.1.2** : Add one paragraph on scalability of TimeAutoDiff along feature axis. See Appendix J for further details on experimental setting.
6. **Section 5.2** : Add three more baselines for Imputation & Forecasting tasks. [PatchTST, iTransformer, TimeMixer++]. Update the Figure 9, and all the related tables in the **Appendix K**.  Add the descriptions on the above three models in the **Appendix D.2**.
7. **Section 5.3.3** : Add more experiments on TV-MCG. The experiment is designed to check the robustness of TimeAutoDiff on corrupted metadata under TV-MCG setting.

We hope that these revisions make sense to the reviewers’ comments.
We will continue to refine the manuscript and improve clarity where needed.

Best regards,
Namjoon Suh

---

### Decision · Action_Editor_uSHR · 2025-12-01

**Recommendation:** Accept as is

**Additional Comments:**

There are some formatting issues, such as the undesirable space before the semicolon or period of some sentences, e.g.:
- "full time horizon rather than denoising one timestep at a time ;"
- "specific components of the DDPM denoiser ."

**Audience:**

Yes

**Audience Explanation:**

The proposed TimeAutoDiff is a unified latent-diffusion framework that unifies four time-series tasks within a single model and natively
handles heterogeneous features. The time series analysis community has been growing into a main theme of AI. This paper presents a unified generative framework, which will push the boundary forward and is highly anticipated by the community.

**Claims And Evidence:**

Yes

**Claims Explanation:**

Pre-rebuttal, there are claims not well supported, which were pointed out by the reviewers. After rebuttal, all the claims are satisfactorily justified by accurate, convincing, and clear evidence, as recognized unanimously by the reviewers.